# Missing Pattern Recognized Diffusion Imputation Model for Missing Not At Random

## Abstract

Missing data frequently arises across diverse domains, including time-series and image domains. In the real world, missing occurrences often depend on the unobservable values themselves, which are referred to as Missing Not at Random (MNAR). To address this, numerous generative models have been proposed, with diffusion models in particular demonstrating strong capabilities in out-of-sample imputation. However, most existing diffusion-based imputation approaches overlook the MNAR setting and instead rely on restrictive assumptions about the missing process, thereby limiting their applicability to practical scenarios. In this work, we introduce the Missing Pattern Recognized Diffusion Imputation Model (PRDIM), a novel framework that explicitly captures the missing pattern and precisely imputes unobserved values. PRDIM iteratively maximizes the likelihood of the joint distribution for observed values and missing mask under an Expectation-Maximization (EM) algorithm. In this sense, we first employ a pattern recognizer, which approximates the underlying missing pattern and provides guidance during every inference toward more plausible imputations with respect to the missing information. In various experimental settings, we demonstrate that PRDIM achieves the state-of-the-art performance compared to previous diffusion imputation approaches under MNAR setting.

## 1 Introduction

Missing data imputation aims to recover missing values from partially observed incomplete datasets, and the imputation algorithms serve as a fundamental component in many domains, including healthcare (Goldberger et al., 2000), traffic (Li et al., 2017), and image domain (Xiao et al., 2017). Formally, the imputation goal is to accurately estimate missing values conditioned on observed values. Many recent imputation models assume the missing to be random or to depend on observed values, which are referred to as Missing Completely at Random (MCAR) and Missing at Random (MAR) respectively (Little & Rubin, 1987; Schafer, 1997). However, in real-world scenarios, we have missing values because of some underlying causes such as health deterioration or mortality (Carreras et al., 2021); so the missing tends to have patterns in its occurrences. Therefore, such patterned missing cases are referred to as Missing Not at Random (MNAR), which considered to be more realistic, and which have not been approached by diffusion-based imputation models in the previous works. Consequently, this paper proposes a diffusion-based imputation model to estimate the *missing pattern* to better recover missing under the general MNAR setting.

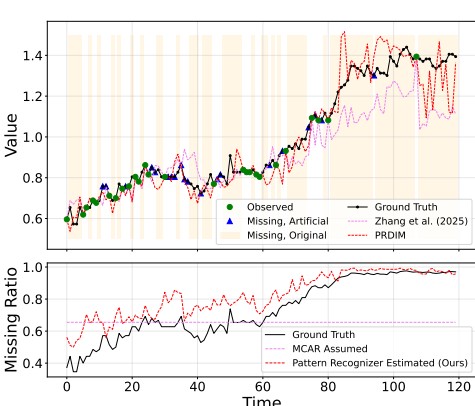

Figure 1: (Top) Comparison of imputation performance on original missing entries (orange) versus artificial missing entries (blue) under observed values (green). (Bottom) Estimated missing ratio by pattern recognizer (red) regard to true missing ratio.

Table 1: Numerical results; imputing original missing presents a more challenging task.

| Method | Artificial missing entries | | | Original missing entries | | |
|---|---|---|---|---|---|---|
| | RMSE | MAE | MRE | RMSE | MAE | MRE |
| Zhang et al. (2025) | 0.230 | 0.146 | 21.572 | 1.209 | 0.782 | 46.188 |
| PRDIM | **0.201** | **0.124** | **18.310** | **1.057** | **0.663** | **39.156** |

To impute high-dimensional data, recent studies have increasingly employed diffusion models (Ho et al., 2020; Song et al., 2021), which provide a powerful framework for capturing complex data distributions under incomplete settings. Existing methods are generally built upon conditional diffusion frameworks (Dhariwal & Nichol, 2021; Ho & Salimans, 2022), in which target entries are artificially masked and treated as missing values for training and evaluation. These methods could be trained on incomplete data; however, direct evaluation on the original missing values is infeasible. Consequently, their performance is typically assessed using artificially masked values. We illustrate this critical distinction between the artificial and original missing entries in the upper part of Figure 1, and argue that the evaluation of the latter is essential for measuring the practical imputation capability. Furthermore, Table 1 demonstrates that imputing original missing entries is a more challenging task. Additional imputation results and underlying missing probability estimations shown in Figure 1 are provided in Appendix C.2.4 and Appendix C.2.5. A detailed description of *original* versus *artificial* missing entries is further discussed in Section 2.3.

In this work, we target the more realistic and challenging setting of imputing original missing values under MNAR. We introduce Missing **P**attern **R**ecognized **D**iffusion **I**mputation **M**odel (**PRDIM**), a diffusion-based imputation model augmented with a *pattern recognizer* that explicitly models the missing pattern with EM framework. By learning the missing pattern, PRDIM provides additional guidance during denoising and provides more precise imputation results than existing methods. Through extensive experiments, we demonstrate that PRDIM achieves state-of-the-art performance on multiple benchmarks, significantly improving imputation accuracy under MNAR conditions.

In summary, our contributions are as follows:

- We propose PRDIM, an EM-based algorithm that maximizes the joint likelihood of observed values and mask information within a diffusion framework. This design enables the model to infer latent missing patterns in incomplete data.

- We theoretically demonstrate that the missing model, termed the pattern recognizer, can provide additional approximate guidance for imputing missing values during the expectation step and inference when the mask information is available.

- We empirically validate PRDIM on incomplete datasets under MNAR setting, showing that it consistently outperforms existing approaches across multiple evaluation metrics.

## 2 PRELIMINARIES

### 2.1 MISSING DATA IMPUTATION FOR MNAR

As discussed earlier, accurate estimation of the missing values requires conditioning on both the observed variables $X^{\text{obs}}$ and the missing mask $M$. Let $X = [X_d] \in \mathbb{R}^D$ be a complete instance with $D$ dimensions, and $M \in \{0, 1\}^D$ the missing indicator where $M_d = 1$ if $X_d$ is observed, otherwise 0. We denote the observed and missing subsets by $X^{\text{obs}} = X \odot M$ and $X^{\text{mis}} = X \odot (1 - M)$ under $X = (X^{\text{obs}}, X^{\text{mis}})$, respectively. Given access to the underlying complete data $X$ and its mask $M$, the ultimate goal is to recover the joint distribution of the observed data, missing mask, and missed data:

$$\max_{\theta,\phi} \mathbb{E}\big[ \log p_{\theta,\phi}(X^{\text{obs}}, X^{\text{mis}}, M) \big], \quad \text{with } (X^{\text{obs}}, X^{\text{mis}}, M) \sim p_{\text{data}}(X, M).$$

Here, $\theta$ and $\phi$ are the parameters of the conditional distribution, which will be discussed in their individual roles on describing distributions of $X$ and $M|X$, respectively.

In the scenario of missing value imputation on incomplete data, the inference becomes maximizing the joint distribution of only two random variables $X^{obs}$ and $M$ because of an unobservable property of $X^{\text{mis}}$. This likelihood maximization is formulated with the expectation on $X^{mis}$, which eventually turns the problem into the Expectation-Maximization framework.

$$p_{\theta,\phi}(X^{\text{obs}}, M) = \int_{X^{\text{mis}}} p_{\theta,\phi}(X^{\text{obs}}, X^{\text{mis}}, M) dX^{\text{mis}} = \int_{X^{\text{mis}}} p_\theta(X) \, p_\phi(M|X) dX^{\text{mis}}$$

Consequently, principled inference requires joint modeling of $p_\theta(X)$ and $p_\phi(M|X)$ and optimizing a suitable lower bound of the EM algorithm. Now, the focus becomes how to infer the two distributions: $p_\theta(X)$ and $p_\phi(M|X)$. Particularly, the inference requirement on $p_\phi(M|X)$ becomes different depending on the assumed missing mechanism across different scenarios.

**Missing Mechanisms** Since the distribution $p_{\theta,\phi}(X, M)$ can be decomposed as $p_\theta(X)p_\phi(M|X)$, it becomes necessary to explicitly model the generation process of the mask variable $M$. The following three missing processes (Little & Rubin, 1987) can be modeled as a conditional distribution $p_\phi(M|X)$ following the standard taxonomy:

$$\text{Missing Completely At Random (\textbf{MCAR}): } p_\phi(M|X) = p_\phi(M),$$

$$\text{Missing At Random (\textbf{MAR}): } p_\phi(M|X) = p_\phi(M|X^{\text{obs}}),$$

$$\text{Missing Not At Random (\textbf{MNAR}): } p_\phi(M|X) = p_\phi(M|X^{\text{obs}}, X^{\text{mis}}).$$

Under MCAR/MAR, the likelihood $p_{\theta,\phi}(X^{\text{obs}}, M)$ which is proportional to $p_\theta(X^{\text{obs}})$ can be learned while ignoring missing process (Mattei & Frellsen, 2019) with respect to the missing variable $X^{\text{mis}}$. In contrast, the mask also depends on unobserved values under MNAR; makes the missing process non-ignorable (Ipsen et al., 2020). While more realistic scenario comes from MNAR, this new requirement of inferring $p_\phi(M|X)$ renders the imputation models under MAR and MCAR to be ineffective and needs to be overhauled significantly.

**Missing Model** Some previous works have incorporated the missing mask $M$ as a supervised learning target. Originated from Generative Adversarial Network (Goodfellow et al., 2014), GAIN (Yoon et al., 2018a) first proposed that a discriminator $D_\phi(X)$ can be trained to approximate $p(M|X)$, with the objective of optimal discriminator $D_{\phi^*}$ towards the specific missing ratio regardless of data distribution. This design ensures that, when the missing mechanism is independent of the data, the discriminator converges to a uniform prediction over missing and observed variables.

Modified from Variational Autoencoder (Kingma & Welling, 2013), not-MIWAE (Ipsen et al., 2020) extended the discriminator framework to MNAR by directly modeling $p(M|X^{\text{obs}}, X^{\text{mis}})$. Their approach demonstrated that the discriminator loss can be integrated into a variational objective, allowing optimization via minimization of an additional term in the ELBO.

Under the shared assumption adopted by GAIN and not-MIWAE, each missing value indicator $M_d \in \{0, 1\}$ follows Bernoulli distribution conditioned on the entire data (*i.e.* $\log p(M|X) = \sum_{d=1}^{D} \log p(M_d|X)$); the loss for the missing model $D_\phi$ can be formulated as a binary cross-entropy (BCE) objective:

$$\mathcal{L}(M, X, D_\phi) = -M^\top \log D_\phi(X) - (1 - M)^\top \log\left(1 - D_\phi(X)\right), \tag{1}$$

where optimal $D_{\phi^*}(X)$ predicts the probability whether each entry would be observed or not (*i.e.* $D_{\phi^*}(X) = \left[p(M_d = 1|X)\right] \in \mathbb{R}^D$). This formulation provides a flexible way to incorporate the missing mechanism into generative imputation models. In this sense, we hereafter refer to the discriminator $D_\phi$ as the *pattern recognizer*. Correspondingly, the loss $\mathcal{L}$ will be denoted as $\mathcal{L}_{\text{PR}}$.

## 2.2 DIFFUSION MODELS

Diffusion models have achieved state-of-the-art generation performances in multiple domains, including vision (Esser et al., 2024), audio (Kong et al., 2020), graphs (Jo et al., 2022), and time-series (Coletta et al., 2023). They learn a data distribution by inverting a Markovian noising process. The forward process gradually corrupts a clean sample $X_0 \sim q(X_0)$ with Gaussian noise according to a prescribed variance schedule $\{\beta_t\}_{t=1}^T$:

$$q(X_{1:T}|X_0) \coloneqq \Pi_{t=1}^T q(X_t|X_{t-1}) \text{ where } q(X_t|X_{t-1}) = \mathcal{N}(\sqrt{\alpha_t}, (1 - \alpha_t)\mathbf{I}). \tag{2}$$

Here, $\alpha_t \coloneqq 1 - \beta_t$ and $\bar{\alpha}_t \coloneqq \Pi_{s=1}^t \alpha_s$. This yields the closed form to sample $X_t$ at time $t$ given $X_0$:

$$q(X_t|X_0) = \mathcal{N}(\sqrt{\bar{\alpha}_t}X_0, (1 - \bar{\alpha}_t\mathbf{I}) \tag{3}$$

The reverse process is modeled as a learned Markov chain that gradually removes noise:

$$p_\theta(X_{0:T}) \coloneqq \Pi_{t=1}^T p_\theta(X_T)p_\theta(X_{t-1}|X_t) \tag{4}$$

where $p_\theta(X_T)$ is a standard Gaussian prior, $\theta$ is the learnable parameter of a diffusion model.

Diffusion models are trained by maximizing a variational lower bound on the log-likelihood:

$$\mathbb{E}_{X_0 \sim q(X_0)}\left[\log p_\theta(X_0)\right] \geq \mathbb{E}_{X_0 \sim q(X_0), X_{1:T} \sim q(X_{1:T}|X_0)}\left[\log \frac{p_\theta(X_{0:T})}{q(X_{1:T}|X_0)}\right] \tag{5}$$

This objective could be reduced as the data reconstruction (Karras et al., 2022), the noise prediction (Ho et al., 2020), or the score matching objective (Song et al., 2021). At inference time, sampling proceeds by drawing $X_T \sim p_T(X_T)$ and iteratively applying the learned reverse transitions $p_\theta(X_{t-1}|X_t)$ to obtain a clean sample $X_0$.

We present the entire methodology of this work from the perspective of data reconstruction (*i.e.* $X_0$ prediction). Under this view, the diffusion model is trained to reconstruct the clean data $X_0$ from noisy samples $X_t$ across timesteps. Accordingly, the diffusion loss function for a single sample $X_0$ is defined as

$$\mathcal{L}_{\text{diff}}(X_0, X_t, t, f_\theta) = \lambda(t)\|f_\theta(X_t, t) - X_0\|_2^2 \tag{6}$$

where $t \sim \mathcal{U}(0, T)$, $\lambda(t) = 1$ in our experiment, $f_\theta$ is the $X_0$ prediction network parameterized by $\theta$, and $X_t$ follows Equation 3.

### 2.3 REVISITING THE OBJECTIVE OF DIFFUSION IMPUTATION MODELS

In this section, we clarify how the target objective adopted in our work differs in perspective from those used in existing diffusion imputation models, and we justify the validity of our proposed objective. Throughout this paper, the term *original missing* values refers to two situations: (i) naturally occurring missing values in real-world datasets such as PhysioNet (Goldberger et al., 2000) or AirQuality (Zhang et al., 2017) which the underlying values are not accessible for evaluation, and (ii) simulated missing values generated by applying a missing mechanism (under MCAR, MAR, or MNAR) to a complete dataset. To evaluate imputation performance on such original missing values, we construct various incomplete datasets by applying different missing mechanisms to complete benchmarks, and then apply imputation methods to these resulting incomplete data distributions.

For an originally incomplete dataset $(X_0^{\text{obs}}, X_0^{\text{mis}})$, $M_O$ be the corresponding original missing mask. In many prior works (Tashiro et al., 2021; Zhou et al., 2024; Liu et al., 2024), it is common practice to impose an additional mask on $X_0^{\text{obs}}$ during training. Let $M_A$ denote this *artificial missing* mask, and $X_0^{\text{obs},A}$ indicates the subset of observed entries remaining after the artificial masking is applied (i.e., $X_0^{\text{obs},A} \equiv X_0^{\text{obs}} \odot M_A$). These artificially masked entries supply the supervision required for both training and quantitative evaluation. We can formulate these works aim to obtain $p_\theta(X_0 \mid X_0^{\text{obs},A}, M_A)$ and evaluation follows the same protocol by inserting an artificial mask into the test data. From this perspective, our research question emerges from a fundamentally different viewpoint:

**If the distribution of the original missing mask $M_O$ differs substantially from that of the artificial mask $M_A$, can a model trained under $p_\theta(X_0 \mid X_0^{\text{obs},A}, M_A)$ be expected to perform well under $p_\theta(X_0 \mid X_0^{\text{obs}}, M_O)$?**

This distinction is crucial to the motivation of our work. Our methodology newly introduces a pattern recognizer, $D_\phi$, which explicitly learns missing distribution, and it enables to model the distribution $p_{\theta,\phi}(X_0 \mid X_0^{\text{obs}}, M_O)$ regard to underlying missing mechanism.

## 3 METHODOLOGY

We decompose PRDIM into two major components: (i) diffusion backbone pre-training and (ii) missing model training with joint distribution fine-tuning. Each component contains practical implementation details that stabilize training and enhance imputation performance. We begin by presenting an overview of PRDIM in Section 3.1, followed by theoretical analysis in subsequent sections.

### 3.1 PRE-IMPUTATION AND EM ALGORITHM

Figure 2 illustrates the graphical model and overall training procedure of PRDIM. The framework consists of two complementary phases: a diffusion-based pre-imputation stage (Phase 1) and an EM iteration stage (Phase 2). The combination of these two phases enables PRDIM to learn both the data distribution and the missing pattern in a principled manner.

**Phase 1: Diffusion model Pre-training and Pre-imputation.** Direct optimization of the joint distribution $p_\theta(X_0^{\text{obs}}, X_0^{\text{mis}})$ often suffers from instability and overfitting when missing entries are

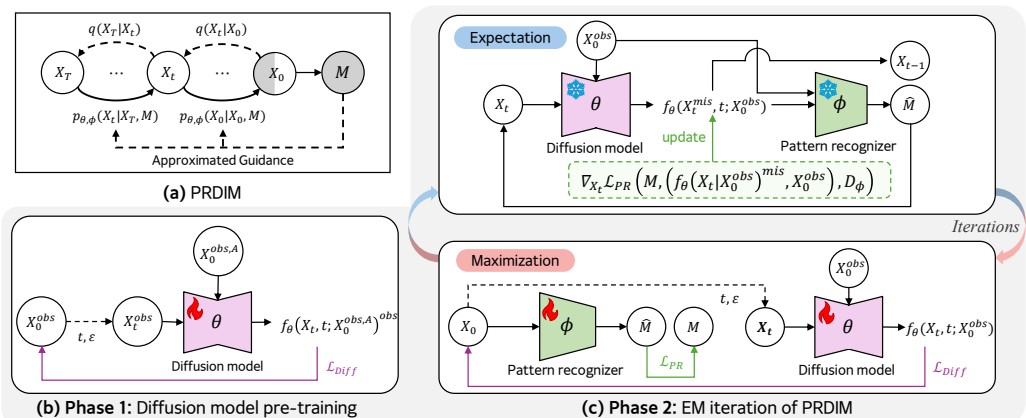

Figure 2: Overall training procedure of PRDIM; (a) Graphical model of PRDIM, (b) Diffusion model pre-training, (c) EM iteration for diffusion model fine-tuning and pattern recognizer training. In expectation step, PRDIM progressively denoising the sample with additional guidance of pattern recognizer (Top). In maximization step, diffusion model and pattern recognizer are trained independently based on $X_0^{\text{obs}}, X_0^{\text{mis}}$ which generated in the previous expectation step (Bottom).

simply set to zero. Phase 1 of PRDIM pre-trains a diffusion backbone under a Tashiro et al. (2021)'s framework enabling the model to capture plausible data distributions from observed variables. For generic imputation tasks, we extend the diffusion target up to conditional input which is proposed at Du et al. (2023a) as Observed Reconstruction Task, which demonstrates joint distribution learning under a conditional modeling. Therefore, to train the diffusion model for pre-imputation, we need to artificially select missing entries. Unlike CSDI, which adopts artificial masking under the MCAR assumption, we avoid this strategy in our approach. Instead, we introduce an adjacent target masking scheme, where artificial missing entries are placed near original missing values to exploit potential correlations. In multivariate time-series data, the artificial missing values are chosen only across the temporal axis. In image data, they are selected as the top, bottom, left, or right neighboring missing pixels. All subsequent experimental results are conducted under this adjacent target masking setup. We denote this artificial missing mask as $A$ and define the corresponding subset $X^{\text{obs},A} := X^{\text{obs}} \odot A$. Through this strategy, the diffusion backbone learns to reconstruct plausible imputations while being robust to any missing pattern. Then, the diffusion loss objective can be rewritten as follows:

$$\mathcal{L}_{\text{diff}}(X_t, t, f_\theta) = \left\| f_\theta(X_t, t; X_0^{\text{obs},A})^{\text{obs}} - X_0^{\text{obs}} \right\|_2^2 \tag{7}$$

where $f_\theta(X_0, X_t, t; X_0^{\text{obs},A})^{\text{obs}}$ is the subset of $X_0$ prediction corresponds to observed entries at timestep $t$. Recent studies have shown that plausible data distributions can be effectively estimated from artificially masked data, as evidenced by He et al. (2022); Peebles & Xie (2023). Furthermore, we demonstrate the performance gain across adjacent target masking and different ratio of MCAR masking schemes in Table 3.

**Phase 2: EM Iteration of PRDIM.** After diffusion pre-training, PRDIM enters the EM algorithm phase, which is inspired by Zhang et al. (2025). Whereas the previous work trains only joint diffusion model, our approach trains both diffusion model $\theta$ and missing pattern recognizer $\phi$ simultaneously to refine the imputation performance.

In the maximization step, the diffusion model $\theta$ is updated to capture the full joint distribution $p_\theta(X_0^{\text{obs}}, X_0^{\text{mis}})$, while the pattern recognizer $D_\phi(X_0) = \hat{M}$ is trained to discriminate the mask variable $M$ as a supervised learning target.

$$\mathcal{L}_{\text{diff}}(X_0, X_t, t, f_\theta) = \left\| f_\theta(X_t, t; X_0^{\text{obs}}) - X_0 \right\|_2^2 \tag{8}$$

$$\mathcal{L}_{\text{PR}}(M, X_0, D_\phi) = -M^\top \log D_\phi(X_0) - (1 - M)^\top \log \left( 1 - D_\phi(X_0) \right) \tag{9}$$

This enables the model to explicitly incorporate the missing pattern. Section 3.2 provides the details of maximization objective.

In the expectation step, the diffusion model generates $X^{\mathrm{mis}}$ conditioned on $M$ and $X^{\mathrm{obs}}$, while the pattern recognizer provides an additional approximated guidance signal that biases the generation toward imputations consistent with the estimated missing patterns. Importantly, during the early iterations, it is acceptable to use a randomly initialized pattern recognizer as guidance. Since such a recognizer has no discriminative ability, the guidance provides a degenerated signal toward a near-zero vector, yielding a neutral effect on the generation process (Kim et al., 2022). Section 3.3 illustrates the guided generation of $X^{\mathrm{mis}}$ for the expectation step.

Furthermore, while Zhang et al. (2025) adopts a soft EM strategy, our framework employs a hard EM variant to enhance the exploration ability to generate $X_0^{\mathrm{mis}}$ distribution, which has been theoretically justified in the context of expectation maximization (Samdani et al., 2012). Details of the maximization and expectation steps that are defined in $X_0$ prediction are summarized in Algorithm 1 and Algorithm 2.

### 3.2 ELBO OBJECTIVE OF MNAR ON DIFFUSION MODEL FRAMEWORK

To address the Missing Not at Random (MNAR) mechanism, we formulate the evidence lower bound (ELBO) of diffusion imputation model within a diffusion framework. Let $X := X_0 = (X_0^{\mathrm{obs}}, X_0^{\mathrm{mis}})$ denote the true data which can be divided into observed and missing variables, and $M$ the missing mask variable. We derive the ELBO of $\log p_{\theta,\phi}(X_0^{\mathrm{obs}}, M)$ as follows:

**Proposition 3.1.** *Suppose that the variational distribution and missing process satisfy the following assumptions: (i) Forward process satisfies conditional independence to mask variable given entire data $q(X_{1:T}|X_0, M) = q(X_{1:T}|X_0)$, (ii) Missing process satisfies conditional independence to noisy data $p_\phi(M|X_0, X_{1:T}) = p_\phi(M|X_0)$.*

*Then, the ELBO of joint log-likelihood objective of the observed data and mask can be expressed as*

$$\log p_{\theta,\phi}(X_0^{obs}, M) \geq \mathbb{E}_{X_{1:T}, X_0^{mis}}\Big[ \log p_\phi(M|X_0) + \log \frac{p_\theta(X_T) \prod_{t=1}^{T} p_\theta(X_{t-1}|X_t)}{\prod_{t=1}^{T} q(X_t|X_{t-1})} \Big] \quad (10)$$

$$+ \mathbb{E}_{X_{1:T}}\Big[ \mathbb{H}(q(X_0^{mis}|X_0^{obs}, M)) \Big] \quad (11)$$

The proof is given in Appendix B.1. This is the first ELBO derivation of missing patterns under the diffusion setting. Unlike Ipsen et al. (2020), the iterative forward and reverse processes inherent to diffusion models render direct ELBO optimization intractable.

Thus, we utilize the above ELBO in the EM algorithm perspective, so that the generation of $X^{\mathrm{mis}}$ becomes an expectation step. In the maximization step, we simultaneously train the missing model $\phi$ for the missing mask $M$ and the diffusion model $\theta$ that estimates the joint probability of $X^{\mathrm{obs}}, X^{\mathrm{mis}}$ by the Equation 10. Subsequently, the expectation step requires conditional generation of $X^{\mathrm{mis}}$ given $M$, which completes the EM loop by requiring another iteration of conditional generation.

In the expectation step, when $q(X_0^{\mathrm{mis}}|X_0^{\mathrm{obs}}, M)$ is replaced with $p_{\theta,\phi}(X_0^{\mathrm{mis}}|X_0^{\mathrm{obs}}, M)$, an additional guidance term with respect to $M$ becomes necessary during the diffusion sampling process. In addition, while the diffusion model is designed to learn the joint distribution, Theorem 1 in Zhang et al. (2025), which is also provided in Appendix B.3, provides a theoretical justification that $X^{\mathrm{mis}}$ can be estimated via the joint score function. The guidance mechanism will be discussed in detail in the following subsection.

### 3.3 FIND BEST $X^{\mathrm{MIS}}$ WITH APPROXIMATED GUIDANCE OF PATTERN RECOGNIZER

After training both the missing model and the diffusion model, the expectation step allows us to replace $q(X_0^{\mathrm{mis}}|X_0^{\mathrm{obs}}, M)$ with the parameters $\theta, \phi$ that were optimized in the preceding maximization step. Since the underlying generative process is score-based, the gradient term $\nabla_{X_t} \log p_{\theta,\phi}(X_0|X_0^{\mathrm{obs}}, M)$ can be decomposed into two components: the score function term $\nabla_{X_t} p_\theta(X_t|X_0^{\mathrm{obs}})$ corresponding to the joint distribution, and the mask guidance term $\nabla_{X_t} p_\phi(M|X_0^{\mathrm{obs}}, X_t^{\mathrm{mis}})$ reflecting the missing pattern. We further show that the mask guidance term can be approximated with the pattern recognizer $D_\phi$ according to Proposition 3.2.

**Proposition 3.2.** *Suppose that the pattern recognizer $D_\phi$ is optimal which satisfies $D_{\phi^*}(X_0) = \Big[ p_\phi(M_d|X_0) \Big] \in \mathbb{R}^D$, the score function of the joint log-likelihood with respect to the missing mask*

**Algorithm 1** E step - $X_0$ prediction

1: Diffusion model $\theta$, pattern recognizer $\phi$, observed data $X^{\mathrm{obs}}$, and mask data $M$.
2: Sample $X_T \sim \mathcal{N}(0, \mathbf{I})$
3: **for** $t = T, \ldots, 1$ **do**
4:     Get $\hat{X}_0 = f_\theta(X_t, t; X_0^{\mathrm{obs}})$
5:     $\tilde{X}_0 = \hat{X}_0 \odot M + X_0^{\mathrm{obs}} \odot (1 - M)$
6:     $\hat{X}_0 = \hat{X}_0 - \frac{1 - \bar{\alpha}_t}{\sqrt{\bar{\alpha}_t}} \nabla_{X_t} \mathcal{L}_{\mathrm{PR}}(M, \tilde{X}_0, D_\phi)$
7:     Sample $\varepsilon \sim \mathcal{N}(0, \mathbf{I})$
8:     $X_{t-1} = \sqrt{\bar{\alpha}_{t-1}} \hat{X}_0 + \sqrt{1 - \bar{\alpha}_{t-1}} \varepsilon$
9: **end for**
10: **return** $X_0$

**Algorithm 2** M step - $X_0$ prediction

1: Diffusion model $\theta$, pattern recognizer $\phi$, corresponding learning rate $\eta_\theta, \eta_\phi$, imputed data $X_0$, mask data $M$, and maximization epoch $N_m$.
2: **for** $i = 1, \ldots, N_m$ **do**
3:     Sample $t, \varepsilon \sim \mathcal{U}(0, T), \mathcal{N}(0, \mathbf{I})$
4:     $X_t = \sqrt{\bar{\alpha}_t} X + \sqrt{1 - \bar{\alpha}_t} \varepsilon$
5:     $\theta \leftarrow \theta - \eta_\theta \nabla_\theta \mathcal{L}_{\mathrm{diff}}(X_0, X_t, t, f_\theta)$
6:     $\phi \leftarrow \phi - \eta_\phi \nabla_\phi \mathcal{L}_{\mathrm{PR}}(M, X_0, D_\phi)$
7: **end for**
8: **return** $\theta, \phi$

*can be approximated as*

$$\nabla_{X_t} \log p_{\theta,\phi}(X_t | X_0^{obs}, M) \simeq \nabla_{X_t} \log p_\theta(X_t | X_0^{obs}) - \nabla_{X_t} \mathcal{L}_{PR}\Big(M, \hat{X}_0, D_{\phi^*}\Big) \tag{12}$$

*where* $\hat{X}_0 := (f_\theta(X_t, t; X_0^{obs})^{mis}, X_0^{obs}) = f_\theta(X_t, t; X_0^{obs}) \odot (1 - M) + X_0^{obs} \odot M$.

Here, $f_\theta(X_t, t; X_0^{\mathrm{obs}})$ is $X_0$ prediction at timestep $t$, and its superscript means the subset which indicates missing entries (*i.e.* $f_\theta^{\mathrm{mis}} = f_\theta \odot (1 - M)$). The detailed proof is given in Appendix B.2. This proposition is noteworthy because it steers the more convincing gradient of the intermediate sample $X_t$ with respect to the estimated missing pattern $D_\phi$; it provides a meaningful signal according to the negative missing probability $\mathcal{L}_{\mathrm{PR}}$.

To summarize, within the DDPM framework, given $X_t$ at time step $t$, the variable $X_{t-1}$ can be obtained through the following three steps. First, using the diffusion model together with Tweedie's formula (Carlin et al., 2000; Efron, 2011), we can compute the posterior mean and an intermediate estimate $\tilde{X}_{t-1}^{\mathrm{mis}}$ (Ho et al., 2020) only with the diffusion parameter $\theta$:

$$\tilde{X}_{t-1} = \frac{1}{\sqrt{\alpha_t}}(X_t + (1 - \alpha_t)\nabla_{X_t} \log p_\theta(X_t | X_0^{\mathrm{obs}})) + \sigma_t Z \quad \text{where} \quad Z \sim \mathcal{N}(0, \mathbf{I}) \tag{13}$$

$$\hat{X}_0 = \frac{1}{\sqrt{\bar{\alpha}_t}}(X_t + (1 - \bar{\alpha}_t)\nabla_{X_t} \log p_\theta(X_t | X_0^{\mathrm{obs}})) = f_\theta(X_t, t; X_0^{\mathrm{obs}}) \tag{14}$$

Second, the pattern recognizer evaluates the missing probability for each entry based on the estimated missing values $\hat{X}_0^{\mathrm{mis}}$ (Equation 14) and the observed component $X_0^{\mathrm{obs}}$. Finally, according to Proposition 3.2, the denoised sample $X_{t-1} = (X_{t-1}^{\mathrm{obs}}, X_{t-1}^{\mathrm{mis}})$ at time $t-1$ is updated by incorporating the approximated guidance.

$$X_{t-1} = \tilde{X}_{t-1} - \frac{1 - \alpha_t}{\sqrt{\alpha_t}} \nabla_{X_t} \mathcal{L}_{\mathrm{PR}}\Big(M, (f_\theta(X_t, t; X_0^{\mathrm{obs}})^{\mathrm{mis}}, X_0^{\mathrm{obs}}), D_\phi\Big) \tag{15}$$

## 4 EXPERIMENTS

In this section, we evaluate PRDIM on multiple benchmark datasets under the MNAR setting. The results show that PRDIM achieves consistent improvements over prior methods, confirming the effectiveness of incorporating the missing model into the diffusion framework. The detailed experimental settings are provided in the Appendix C.

### 4.1 EXPERIMENTAL SETTING

**Synthetic MNAR Datasets** We evaluate our method on three widely used multivariate time-series datasets and one image dataset: (1) **ETT** (Zhou et al., 2021), which records load and temperature of

Table 2: Overall **MAE** performance on three benchmark datasets. We report mean $\pm$ std over 5 runs according to each methodology. Best results are in **bold**, and second best results are in underline.

| Method | Original / Out-of-Sample | | | Original / In-Sample | | |
|---|---|---|---|---|---|---|
| | ETT | STOCK | PEMS-Bay | ETT | STOCK | PEMS-Bay |
| Mean | $2.034_{\pm0.000}$ | $1.949_{\pm0.000}$ | $0.813_{\pm0.000}$ | $1.486_{\pm0.000}$ | $2.039_{\pm0.000}$ | $0.789_{\pm0.000}$ |
| *Discriminative models* | | | | | | |
| TimesNet | $1.044_{\pm0.065}$ | $1.111_{\pm0.073}$ | $0.291_{\pm0.007}$ | $1.154_{\pm0.068}$ | $1.221_{\pm0.077}$ | $0.225_{\pm0.001}$ |
| TimeMixer++ | $1.642_{\pm0.025}$ | $1.287_{\pm0.239}$ | $0.579_{\pm0.018}$ | $1.100_{\pm0.032}$ | $1.369_{\pm0.260}$ | $0.557_{\pm0.020}$ |
| BRITS | $0.992_{\pm0.037}$ | $0.627_{\pm0.010}$ | $0.278_{\pm0.006}$ | $0.491_{\pm0.008}$ | $0.701_{\pm0.010}$ | $0.182_{\pm0.003}$ |
| SAITS | $0.814_{\pm0.046}$ | $0.442_{\pm0.022}$ | $0.302_{\pm0.009}$ | $0.366_{\pm0.014}$ | $0.498_{\pm0.025}$ | $0.212_{\pm0.003}$ |
| *Generative models* | | | | | | |
| GP-VAE | $1.511_{\pm0.011}$ | $0.902_{\pm0.109}$ | $0.345_{\pm0.001}$ | $0.896_{\pm0.018}$ | $1.010_{\pm0.118}$ | $0.292_{\pm0.002}$ |
| not-MIWAE | $1.311_{\pm0.016}$ | $0.681_{\pm0.045}$ | $0.396_{\pm0.005}$ | $0.637_{\pm0.011}$ | $0.759_{\pm0.039}$ | $0.352_{\pm0.005}$ |
| *Diffusion-based models* | | | | | | |
| CSDI | $1.071_{\pm0.001}$ | $0.641_{\pm0.000}$ | $\underline{0.177}_{\pm0.000}$ | $0.522_{\pm0.001}$ | $0.710_{\pm0.000}$ | $\underline{0.158}_{\pm0.000}$ |
| MTSCI | $0.957_{\pm0.001}$ | $0.736_{\pm0.001}$ | $0.193_{\pm0.000}$ | $0.500_{\pm0.000}$ | $0.809_{\pm0.001}$ | $0.179_{\pm0.000}$ |
| cDiffPuter | $\underline{0.782}_{\pm0.000}$ | $\underline{0.406}_{\pm0.000}$ | $0.182_{\pm0.000}$ | $\underline{0.362}_{\pm0.000}$ | $\underline{0.450}_{\pm0.000}$ | $0.168_{\pm0.000}$ |
| **PRDIM** | $\mathbf{0.663}_{\pm0.000}$ | $\mathbf{0.254}_{\pm0.000}$ | $\mathbf{0.170}_{\pm0.000}$ | $\mathbf{0.303}_{\pm0.000}$ | $\mathbf{0.275}_{\pm0.000}$ | $\mathbf{0.154}_{\pm0.000}$ |

electricity transformers from 2016 to 2018 and has been a standard benchmark in time-series forecasting and imputation tasks; (2) **STOCK**, which contains historical daily Google stock prices from 2004 to 2019 and reflects complex temporal dynamics with strong non-stationarities; (3) **PEMS-Bay** (Li et al., 2017), which consists of road occupancy rates collected from highway sensors in the Bay Area, exhibiting highly correlated spatial-temporal patterns, and (4) **Fashion-MNIST** (FM-NIST) (Xiao et al., 2017), which consists of gray-scale images of clothing items across 10 categories, widely used as a benchmark for evaluating imputation models.

These datasets are originally complete, and we arbitrarily mask missing values using a MNAR mechanism. As discussed in the introduction, all baseline models are trained on the resulting incomplete data, while evaluation is performed on the imputation of unobserved ground-truth values. The missing mechanism in our main experiment is proposed in not-MIWAE (Ipsen et al., 2020). A detailed description of the employed missing mechanism is provided in the Appendix C.

**Baselines** We compare our approach PRDIM against 12 representative imputation methods. **Mean** serves as a traditional statistical baseline. **TimesNet** (Wu et al., 2022), **TimeMixer++** (Wang et al., 2024), **BRITS** (Cao et al., 2018), and **SAITS** (Du et al., 2023a) are discriminative models, known to achieve strong performance for time-series imputations. **GP-VAE** (Fortuin et al., 2020) and **not-MIWAE** (Ipsen et al., 2020) are VAE-based imputation model that could be implemented on incomplete data. Among diffusion-based methods, we reproduce **CSDI** (Tashiro et al., 2021), **MTSCI** (Zhou et al., 2024), and **DiffPuter** (Zhang et al., 2025) which shows robust performance on various datasets. Specifically, We modified DiffPuter into conditional diffusion framework, which denote as **cDiffPuter**. In image domain, we reproduce **misGAN** (Li et al., 2019) and **MCFlow** (Richardson et al., 2020) which are GAN, Flow based generative imputation model respectively. The details of baselines can be found in Appendix A.

**Evaluation Metrics** We consider two types of evaluation: (i) *original in-sample* imputation and *original out-of-sample* imputation, both targeting the recovery of true missing values. As highlighted in the introduction, our main goal is to impute original missing entries, as opposed to artificially masked ones. We report the three error-based metrics (i) RMSE, (ii) MAE, and (iii) MRE which are defined in Appendix C, throughout this section.

## 4.2 OVERALL PERFORMANCE

We first evaluate PRDIM against representative discriminative, generative, and diffusion-based baselines on three multivariate time-series datasets (ETT, STOCK, PEMS-Bay) and FMNIST dataset. As reported in Table 2 and detailed in the Appendix C.1, PRDIM consistently achieves the best performance across all metrics and datasets. In particular, the gains are most pronounced on out-of-sample imputation tasks, while in-sample results remain competitive, suggesting that PRDIM generalizes well to unseen missing values.

Figure 3 shows qualitative comparisons on FMNIST, further illustrating that PRDIM generates semantically more consistent reconstructions than other approaches. In addition, the corresponding FID scores demonstrate that our model achieves outstanding generative quality, corroborating the visual improvements with quantitative evidence. These results highlight the importance of explicitly modeling the missing mechanism in the diffusion process.

### 4.3 CONTROLLED PARAMETER ANALYSIS

To better understand the dynamics of PRDIM, we analyze the training behavior of the EM procedure. Figure 4 shows the convergence of the pattern recognizer's loss, where red curves indicate the ability to distinguish miss-

Figure 3: Imputation on FMNIST. Second row shows observed inputs, where red pixels indicate missing.

ing values and blue curves correspond to observed values. The results on both the ETT and STOCK datasets demonstrate that the pattern recognizer effectively captures the missing pattern, thereby providing informative guidance during generation. Furthermore, Figure 5 illustrates the evolution of MRE across EM iterations on ETT and STOCK datasets, revealing a consistent improvement in imputation accuracy as the number of EM epochs increases.

Moreover, according to the findings on a previous work (Ho & Salimans, 2022), increasing the weight of guidance generally leads to the better generation. As shown in Figure 6, the imputation performance consistently improves as the guidance scale increases, demonstrating the effectiveness of guidance weighting. These findings confirm that EM refinement is a critical component of PRDIM, substantially enhancing its capacity to model the joint distribution of the data and the missing pattern. Despite the strong performance of our proposed PRDIM framework, it should be noted that introducing an additional guidance term in diffusion models inherently incurs extra computational cost (Kim et al., 2022; Chung et al., 2023). To clarify this trade-off, we provide supplementary results in Table 8 of Appendix C, where training and inference times are compared across different diffusion-based imputation models.

### 4.4 ABLATION STUDIES

We further conduct ablation experiments to quantify the contribution of each component in PRDIM. Table 3 reports results when either the pattern recognizer is removed or hard EM is replaced with soft EM. Both modifications lead to a significant performance drop, indicating that explicit missing modeling and iterative EM updates are indispensable for exploring the missing data distribution. Furthermore, we investigate the impact of the artificial missing mask $A$. While previous works typically adopt $A$ under the MCAR, we report performance for missing rates of $10\%$, $50\%$, and $90\%$ of $A$. Overall, the missing rate of $A$ has limited influence on imputation accuracy. Specifically, extremely high missing rate degrades the quality of pre-imputation in Phase 1, thereby requiring more EM iterations for convergence.

Tables 4 and 5 investigate robustness under different missing mechanisms by applying MNAR and MCAR masks to the ETT dataset. Implementation details for the corresponding mechanisms are

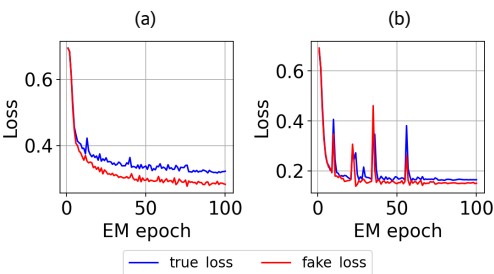

Figure 4: Convergence of the pattern recognizer's loss during EM training. (a) ETT (b) STOCK.

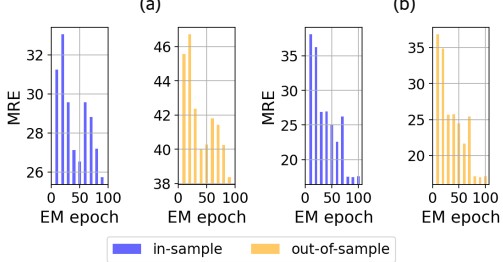

Figure 5: Evolution of MRE across EM epochs. (a) ETT (b) STOCK.

Table 3: Ablation study results on the **STOCK** dataset. We report the average over 5 runs. $n\%$ missing means utilizing artificial missing masking $A$ under MCAR mechanism with $n\%$ instead of adjacent target masking defined in Section 3.1.

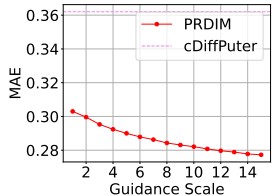

Figure 6: Effect of guidance scale on STOCK.

| Method | original out-of-sample | | | original in-sample | | |
|---|---|---|---|---|---|---|
| | RMSE | MAE | MRE | RMSE | MAE | MRE |
| PRDIM | 0.599 | **0.254** | **16.794** | 0.633 | **0.275** | **17.150** |
| 10% missing of $A$ | **0.590** | 0.258 | 17.057 | **0.630** | 0.282 | 17.632 |
| 50% missing of $A$ | 0.624 | 0.259 | 17.118 | 0.667 | 0.281 | 17.563 |
| 90% missing of $A$ | 0.631 | 0.293 | 19.396 | 0.677 | 0.323 | 20.146 |
| w/o PR | 0.650 | 0.306 | 20.233 | 0.691 | 0.339 | 21.171 |
| w/o PR and hard EM | 0.734 | 0.406 | 26.878 | 0.778 | 0.450 | 28.064 |

described in Appendix C. PRDIM consistently outperforms baselines under MNAR, whereas under MCAR the advantage diminishes, as the pattern recognizer effectively learns randomly in this scenario. Together, these ablation studies confirm the necessity of PRDIM's design choices and its robustness across varying missing conditions.

Table 4: Imputation performance on the **ETT** dataset in different **MNAR** pattern. We report the average RMSE / MAE / MRE over 5 runs.

| Method | out-of-sample | in-sample |
|---|---|---|
| CSDI | 0.345 / 0.218 / 13.825 | 0.233 / 0.159 / 13.259 |
| cDiffPuter | **0.264** / 0.177 / 11.168 | 0.281 / 0.178 / 14.836 |
| PRDIM | 0.282 / **0.171** / **10.837** | **0.197** / **0.130** / **10.816** |

Table 5: Imputation performance on the **ETT** dataset in **MCAR** mechanism. We report the average RMSE / MAE / MRE over 5 runs.

| Method | out-of-sample | in-sample |
|---|---|---|
| CSDI | 0.237 / 0.160 / 15.550 | 0.182 / 0.128 / 16.752 |
| cDiffPuter | 0.251 / 0.162 / 15.737 | 0.202 / 0.136 / 17.795 |
| PRDIM | **0.225 / 0.147 / 14.267** | **0.172 / 0.118 / 15.485** |

### 4.5 Application of PRDIM

We conducted two additional experiments to investigate the importance of Phase 1 initialization and the generalization capability of PRDIM. First, because the main objective of PRDIM can operate even without an explicit Phase 1, we evaluated several initialization strategies (MEAN, BRITS, SAITS) under identical conditions to examine how different pre-imputation methods influence PRDIM's performance. As shown in Table 6, initialization with CSDI yields the best results. However, PRDIM consistently outperforms the baseline regardless of the chosen initialization method, demonstrating that it does not heavily rely on a particular Phase 1 strategy.

Second, we assessed post-imputation classification accuracy on the FMNIST dataset to evaluate how well the imputed data support downstream tasks. This experiment examines whether higher-quality imputations translate into improved task performance. As reported in Table 7, PRDIM surpasses Flow-based, GAN-based, and prior diffusion-based approaches, indicating that PRDIM is capable of restoring semantically meaningful information even in the image domain.

Table 6: Out-of-sample MAE under different Phase 1 methods.

| Initialization Method | ETT | STOCK | PEMS-Bay |
|---|---|---|---|
| MEAN (close to w/o phase 1) | 0.766 | 0.326 | 0.207 |
| BRITS | 0.824 | 0.300 | 0.188 |
| SAITS | 0.774 | 0.307 | 0.180 |
| **CSDI (PRDIM)** | **0.663** | **0.254** | **0.170** |

Table 7: Post-imputation classification results on FMNIST.

| Method | Accuracy (%) |
|---|---|
| Clean Data | 92.59 |
| PRDIM | **91.14** |
| MCFlow | 90.49 |
| CSDI | 87.41 |
| misGAN | 84.34 |

## 5 Conclusion

We presented PRDIM, a diffusion-based imputation framework that incorporates an additional discriminator denoted pattern recognizer under an EM algorithm to explicitly estimate missing patterns. We shows that the guidance which understands missing pattern could be helpful for generating missing values precisely in diffusion imputation model. Our theoretical derivation and extensive experiments demonstrate that PRDIM consistently improves imputation compared to existing methods.

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

## A    RELATED WORKS

Traditional imputation approaches typically rely on simple statistical heuristics such as mean, median, or last observation carried forward to fill in missing entries of multivariate time-series data (Kantardzic, 2011). These strategies often fail to capture the complex temporal dynamics and cross-feature dependencies in them. To improve upon these methods, more sophisticated methods such as MICE (Van Buuren & Groothuis-Oudshoorn, 2011), which iteratively applies the EM algorithm, and MissForest (Stekhoven & Bühlmann, 2012), which leverages random forests for iterative refinement, have been proposed. Although these techniques provide more better imputation performances than naive statistical rules, their capacity remains limited when handling high-dimensional time-series data with intricate dependencies.

**Imputation with Deep Learning**    Early attempts to exploit deep learning for time-series imputation include mRNN (Yoon et al., 2018b), which leverages recurrent neural networks to capture temporal dependencies and model complex patterns in partially observed sequences. BRITS (Cao et al., 2018) further improves upon RNN-based imputers by introducing a bidirectional structure, allowing information to flow forward and backward across time to enhance estimation accuracy. Then NAOMI (Liu et al., 2019) combines multi-resolution RNNs with adversarial training strategies to refine imputations at different time scales. SAITS (Du et al., 2023a) introduces a self-attention mechanism to better capture long-range temporal dependencies. Most of them could be reproduced with the released Python toolkit (Du et al., 2023b).

Beyond time-series-specific architectures, several general-purpose imputation frameworks have also shaped the development of recent methods. GAIN (Yoon et al., 2018a), although not tailored to time-series data, was the first to introduce an adversarial discriminator to imputation, providing a novel mechanism to distinguish observed values and missing values. Successively, misGAN (Li et al., 2019) provided multiple generators and discriminators system for stable training across varied missing patterns. Flow-based approaches such as MCFlow (Richardson et al., 2020) further demonstrated that the EM algorithm can be combined with invertible generative models to jointly optimize flow parameters and missing entries. not-MIWAE (Ipsen et al., 2020) addressed the MNAR scenario by explicitly optimizing a missing model within the ELBO objective, which can inference missing values by missing model weighted importance sampling. Although originally proposed for general imputation tasks, these frameworks have significantly influenced subsequent advances in time-series imputation by highlighting the value of probabilistic, adversarial, and likelihood-based modeling.

**Diffusion-based Approaches for Imputation**    In the time-series domain, TimeGrad (Rasul et al., 2021) applies diffusion to probabilistic forecasting, though its design mainly focuses on forecasting task. For the imputation task, CSDI (Tashiro et al., 2021) introduces a conditional diffusion framework with masking to handle arbitrary missing. Building on this line of research, methods such as SSSD (Alcaraz & Strodthoff, 2022) and Diffusion-TS (Yuan & Qiao, 2024) incorporate additional regularization losses tailored to time-series characteristics, thereby enhancing the interpretability of the imputed sequences. On the other hand, MTSCI (Zhou et al., 2024) integrates a contrastive loss to maximize mutual information between observed variable and missed variable which improves generated sample consistency. More recently, DiffPuter (Zhang et al., 2025) further improves probabilistic imputation with the EM algorithm, which progressively refines the missing values.

## B    PROOFS

### B.1    PROOF OF PROPOSITION 3.1

**Proposition 3.1.** *Suppose that the variational distribution and missing process satisfy the following assumptions: (i) Forward process satisfies conditional independence to mask variable given entire data* $q(X_{1:T}|X_0, M) = q(X_{1:T}|X_0)$, *(ii) Missing process satisfies conditional independence to noisy data* $p_\phi(M|X_0, X_{1:T}) = p_\phi(M|X_0)$.

*Then, the ELBO of joint log-likelihood objective of the observed data and mask can be expressed as*

$$\log p_{\theta,\phi}(X_0^{obs}, M) \geq \mathbb{E}_{X_{1:T}, X_0^{mis}}\Big[ \log p_\phi(M|X_0) + \log \frac{p_\theta(X_T)\prod_{t=1}^T p_\theta(X_{t-1}|X_t)}{\prod_{t=1}^T q(X_t|X_{t-1})}\Big] \quad (10)$$

$$+ \mathbb{E}_{X_{1:T}}\Big[\mathbb{H}(q(X_0^{mis}|X_0^{obs}, M))\Big] \quad (11)$$

*Proof.*

$$\log p_{\theta,\phi}(X^{obs}, M) = \log \int_Z \int_{X^{mis}} p_{\theta,\phi}(X^{obs}, X^{mis}, M, Z) dX^{mis} dZ \quad (16)$$

$$= \log \int_Z \int_{X^{mis}} \frac{p_{\theta,\phi}(X^{obs}, X^{mis}, M, Z)}{q(X^{mis}, Z|X^{obs}, M)} q(X^{mis}, Z|X^{obs}, M) dX^{mis} dZ \quad (17)$$

$$= \log \int_Z \int_{X^{mis}} \frac{p_\phi(M|X)p_\theta(X|Z)p_\theta(Z)}{q(Z|X)q(X^{mis}|X^{obs}, M)} q(Z|X)q(X^{mis}|X^{obs}, M) dX^{mis} dZ \quad (18)$$

$$\geq \mathbb{E}_{Z\sim q(Z|X), X^{mis}\sim q(X^{mis}|X^{obs}, M)}[\log p_\phi(M|X) + \log \frac{p_\theta(X|Z)p_\theta(Z)}{q(Z|X)}] \quad (19)$$

$$+ \mathbb{E}_{Z\sim q(Z|X)}[\mathbb{H}(q(X^{mis}|X^{obs}, M))] \quad (20)$$

Equation 19 defines the loss objective between the true parameters and the corresponding variational distribution. Replacing the latent variable $Z$ with the diffusion latents $X_{1:T}$, can be formulated as equation 21 under the Markov property of the diffusion process.

$$\mathbb{E}_{X_{1:T}\sim q(X_{1:T}|X), X^{mis}\sim q(X^{mis}|X^{obs}, M)}[\log p_\phi(M|X) + \log \frac{p_\theta(X|X_1)p_\theta(X_{1:T})}{q(X_{1:T}|X)}] \quad (21)$$

$$\square$$

## B.2 PROOF OF PROPOSITION 3.2

**Proposition 3.2.** *Suppose that the pattern recognizer $D_\phi$ is optimal which satisfies $D_{\phi^*}(X_0) = \Big[p_\phi(M_d|X_0)\Big] \in \mathbb{R}^D$, the score function of the joint log-likelihood with respect to the missing mask can be approximated as*

$$\nabla_{X_t} \log p_{\theta,\phi}(X_t|X_0^{obs}, M) \simeq \nabla_{X_t} \log p_\theta(X_t|X_0^{obs}) - \nabla_{X_t}\mathcal{L}_{PR}\Big(M, \hat{X}_0, D_{\phi^*}\Big) \quad (12)$$

*where $\hat{X}_0 := (f_\theta(X_t, t; X_0^{obs})^{mis}, X_0^{obs}) = f_\theta(X_t, t; X_0^{obs}) \odot (1 - M) + X_0^{obs} \odot M$.*

*Proof.* From the graphical model of Figure 2 (a), conditional independence for missing process satisfies $p_\phi(M|X_0, X_t) = p_\phi(M|X_0)$ at Equation 23. The approximation 24 came from the Thm 1 of (Chung et al., 2023).

$$\nabla_{X_t} \log p_{\theta,\phi}(X_t|X_0^{obs}, M) = \nabla_{X_t} \log p_\theta(X_t|X_0^{obs}) + \nabla_{X_t} \log p_{\theta,\phi}(M|X_t, X_0^{obs}) \quad (22)$$

$$= \nabla_{X_t} \log p_\theta(X_t|X_0^{obs}) + \nabla_{X_t} \log \int p_\phi(M|X_0)p_\theta(X_0^{mis}|X_t, X_0^{obs}) dX_0^{mis} \quad (23)$$

$$\simeq \nabla_{X_t} \log p_\theta(X_t|X_0^{obs}) + \nabla_{X_t} \log p_\phi(M|f_\theta(X_t, t; X_0^{obs})^{mis}, X_0^{obs}) \quad (24)$$

$$= \nabla_{X_t} \log p_\theta(X_t|X_0^{obs}) + \nabla_{X_t} M \log D_{\phi^*}\Big(f_\theta(X_t, t; X_0^{obs})^{mis}, X_0^{obs}\Big) \quad (25)$$

$$+ \nabla_{X_t}(1 - M) \log \Big\{1 - D_{\phi^*}\Big(f_\theta(X_t, t; X_0^{obs})^{mis}, X_0^{obs}\Big)\Big\} \quad (26)$$

$$= \nabla_{X_t} \log p_\theta(X_t|X_0^{obs}) - \nabla_{X_t}\mathcal{L}_{PR}\Big(M, (f_\theta(X_t, t; X_0^{obs})^{mis}, X_0^{obs}), D_{\phi^*}\Big) \quad (27)$$

Since $d$-th element of $D_{\phi^*}(X_0)$ converges to $p_\phi(M_d = 1|X_0)$ (Ipsen et al., 2020; Ma & Zhang, 2021), entire probability of mask variable $M$ follows:

$$\log p_\phi(M|f_\theta(X_t, t; X_0^{\text{obs}})^{\text{mis}}, X_0^{\text{obs}}) = \sum_{i=1}^{D} \log p_\phi(M_d|f_\theta(X_t, t; X_0^{\text{obs}})^{\text{mis}}, X_0^{\text{obs}}) \tag{28}$$

$$= M \log D_{\phi^*}\left(f_\theta(X_t, t; X_0^{\text{obs}})^{\text{mis}}, X_0^{\text{obs}}\right) + (1 - M) \log \left\{1 - D_{\phi^*}\left(f_\theta(X_t, t; X_0^{\text{obs}})^{\text{mis}}, X_0^{\text{obs}}\right)\right\} \tag{29}$$

$\square$

### B.3 REWRITTEN THEOREM 1 OF ZHANG ET AL. (2025)

**Theorem 1.** *Let $X_T$ be a sample from the prior distribution $p_\theta(X_T) = \mathcal{N}(0, \mathbf{I})$, $X$ be the data to impute, and the known entries of $X$ are denoted by $X^{\text{obs}} = X_0^{\text{obs}}$. The score function $\nabla_{X_t} \log p(X_t)$ could be parameterized by neural network $f_\theta(X_t, t; X_0^{obs})$. Applying forward and reverse process of the diffusion model iteratively from $t = T \gg 0$ until $t = 0$ with $\Delta t \to 0$, then $\hat{X}_0$ is a sample from $p_\theta(X)$, under the condition that its observed entries $\hat{X}_0^{obs} = X_0^{obs}$. Formally,*

$$\hat{X}_0 \sim p_\theta(X|X^{obs} = X_0^{obs}) \tag{30}$$

This section presents a rewritten version of Theorem 1 from Zhang et al. (2025), and we refer the reader to the original paper for the detailed proof. The theorem establishes that when learning the joint probability of $X^{\text{obs}}$ and $X^{\text{mis}}$, the missing values can be inferred by conditioning on the observed values of a given sample. We adopt the same line of reasoning in Section 3.3 to support our theoretical development.

## C EXPERIMENT DETAILS

**Model Configuration** In our implementation, the pattern recognizer is designed as a lightweight multi-layer perceptron (MLP) to minimize additional model complexity. As summarized in Table 8, the overall model size remains comparable to other diffusion-based approaches. The higher inference time of PRDIM, relative to competing methods, arises from the use of autogradient of the input with respect to the outputs of the pattern recognizer during the inference process. This design choice, while incurring additional computational cost, enables the model to provide more informative guidance for imputing missing values.

Table 8: Detailed model configuration. Model Size indicates the number of parameter of each diffusion-based model, and $+n$ in PRDIM shows the number of pattern recognizer. The inference time (s) is measured based on the a single inference required to impute the entire out-of-sample data.

| | ETT | | | STOCK | | |
|---|---|---|---|---|---|---|
| **Method** | Model Size | Training Time (s) | Inference Time (s) | Model Size | Training Time (s) | Inference Time (s) |
| CSDI | 164025 | 366 | 39 | 164017 | 138 | 11 |
| MTSCI | 162321 | 585 | 21 | 146969 | 354 | 12 |
| cDiffPuter | 163769 | 981 | 28 | 163761 | 516 | 11 |
| PRDIM | 163769+17376 | 1812 | 47 | 163761+17359 | 1052 | 28 |

**EM Configuration** Since PRDIM is built upon the EM framework, both training time and performance are affected by the design of EM configuration. In this section, we analyze the impact of different EM settings on the STOCK dataset and show that the PRDIM can overcome its inherent training time consumption with respect to imputation performance. We denote **1E** $N_m$**M** as the number of training epochs for the maximization step per expectation step, and $N$ as the total number of EM iterations. In the main

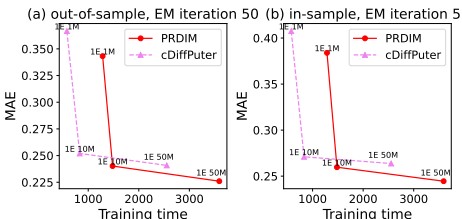

Figure 7: Training time and imputation performance (MAE) under different EM configurations on the **STOCK** dataset.

experiment, both cDiffPuter and PRDIM were trained with $N = 100$ EM iterations and **1E 1M** configuration, i.e., one epoch for the maximization step per expectation step.

When varying the number of training epochs in the maximization step, we observe that the training time increases proportionally to the epoch size. As the number of training epochs in each maximization step increases, the relative computational cost of the expectation step becomes negligible. Thus, the training time of PRDIM converges to that of cDiffPuter as the maximization step grows. Nevertheless, as shown in Figures 7 (a) and 7 (b), PRDIM with the configuration of **1E 10M** and a total of 50 EM iterations achieves superior performance and requires less training time compared to cDiffPuter whose configuration is **1E 50M** and 50 EM iterations. This observation highlights the importance of appropriate hyperparameter tuning of the EM configuration, and further suggests that PRDIM provides strong scalability and practical utility for imputation tasks on real-world datasets.

**Dataset Configuration**   Table 9 summarizes the dataset configurations employed in our experiments across (train / test / valid) set. For the three time-series datasets (ETT, STOCK, and PEMS-Bay), the data size are represented as *time length $\times$ feature dimension*, while for FMNIST, the dimensions are denoted as *width $\times$ height*. The reported missing ratios correspond to the proportion of original missing values observed after applying the MNAR mechanism to generate incomplete data for training, thereby reflecting the intrinsic difficulty of the imputation task.

Table 9: Dataset configuration used in the main experiments. For time-series datasets (ETT, STOCK, PEMS-Bay), the data size is denoted as *time length $\times$ feature dimension*, whereas for FMNIST, it is expressed as *width $\times$ height*. The missing ratios represent the proportion of original missing values after applying the MNAR mechanism to construct incomplete data for training.

| Dataset | ETTm1 | STOCK | PEMS-Bay | FMNIST |
|---|---|---|---|---|
| Data size | $24 \times 7$ | $24 \times 6$ | $12 \times 325$ | $28 \times 28$ |
| # of Samples | 3861 / 983 / 959 | 2418 / 622 / 622 | 5788 / 1448 / 1448 | 60000 / 5000 / 5000 |
| Missing ratio (%) | 21.4 / 43.9 / 14.0 | 21.2 / 20.0 / 20.9 | 13.5 / 13.0 / 14.1 | 25.8 / 25.8 / 25.8 |

**Missing Mechanisms**   Table 2 reports results obtained under the following MNAR mechanism. Inspired by the MNAR mechanism of not-MIWAE (Ipsen et al., 2020), we design the MNAR mechanism such that the probability of a missing entry increases exponentially with its value:

$$p(M_d = 1 | X_d) = \frac{1}{1 + e^{-\text{logits}}}, \quad \text{logits} = W(X_d - b), \tag{31}$$

where $M_d \in \{0, 1\}$ denotes the mask variable of entry $X_d$, $W$ controls the slope, and $b$ is a bias term. This mechanism ensures that entries with values larger than the mean are more likely to be missing, thus faithfully mimicking MNAR conditions. In main experiments on section 4.2, we set $W = 5$ and $b = 0.8$ for all time-series dataset while $W = 7$ and $b = 0.6$ for FMNIST.

To verify coherent results under MCAR and MNAR mechanisms, we follow the missingness simulation procedures of Hyperimpute (Jarrett et al., 2022) and MissingOT (Muzellec et al., 2020), as reported in Table 5 and Table 4. In MCAR setting, each value is excluded following the Bernoulli random variable with a fixed parameter. In our implementation, we randomly assign 10% of the entries as missing to maintain MCAR property. Otherwise, to construct the MNAR setting, we employ a quantile-based mechanism distinct from the previous logistic approach. A subset of variables is randomly selected, and missing values are generated within the $q$-quantile. In our implementation, a missing rate of around 30% was created using a bilateral 25% quantile.

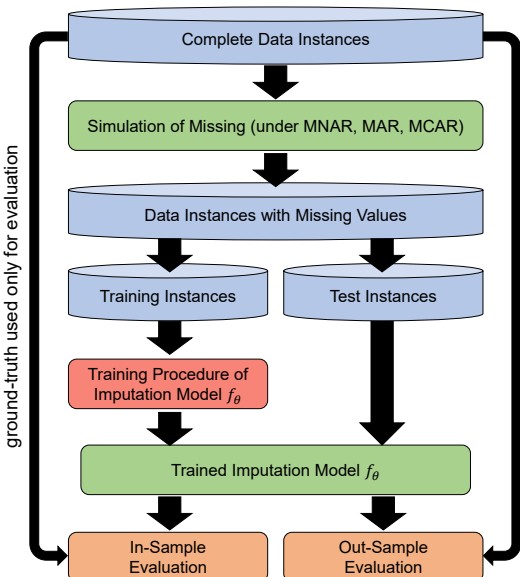

Figure 8: Overall information flow between the in-sample imputation and the out-of-sample imputation.

**Evaluation Metrics**  We report three error-based metrics:

$$\text{RMSE} = \sqrt{\frac{\sum_{n=1,d=1}^{N,D}(X_d^n - \hat{X}_d^n)^2 \times M_d^n}{\sum_{n=1,d=1}^{N,D} M_d^n}}, \tag{32}$$

$$\text{MAE} = \frac{\sum_{n=1,d=1}^{N,D}|X_d^n - \hat{X}_d^n| \times M_d^n}{\sum_{n=1,d=1}^{N,D} M_d^n}, \tag{33}$$

$$\text{MRE} = \frac{\sum_{n=1,d=1}^{N,D}|X_d^n - \hat{X}_d^n| \times M_d^n}{\sum_{n=1,d=1}^{N,D}|X_d^n| \times M_d^n} \times 100(\%), \tag{34}$$

where $X_d^n$ denotes the ground-truth value of dimension $d$ of $n$th sample and $\hat{X}_d^n$ its imputed counterpart. These complementary measures assess squared error, absolute error, and relative error, providing a comprehensive evaluation of imputation performance.

**In-sample and Out-of-sample imputation**  In our experiments, In-sample imputation refers to evaluating imputation performance on the same dataset used for training, whereas out-of-sample imputation evaluates the model on held-out test splits containing unseen data. The simulated MNAR pattern distributions are generated consistently for each split to ensure complete evaluation. Figure 8 summarizes the distinction between the in-sample imputation process and the out-of-sample imputation process.

## C.1 OVERALL PERFORMANCE

In this section, we provide additional results on the overall imputation performance across all time-series datasets in Table 10, Table 11, and Table 12. These results complement the main findings reported in the paper and further validate the effectiveness of our approach.

Table 10: Overall performance on the **ETT** dataset. We report $mean \pm std$ over 5 runs according to each methodology. Best results are in **bold**. Second best results are in underline.

| Method | Original / Out-of-Sample | | | Original / In-Sample | | |
|---|---|---|---|---|---|---|
| | RMSE | MAE | MRE | RMSE | MAE | MRE |
| Mean | $2.307_{\pm0.000}$ | $2.034_{\pm0.000}$ | $120.233_{\pm0.000}$ | $1.618_{\pm0.000}$ | $1.486_{\pm0.000}$ | $127.379_{\pm0.000}$ |
| *Discriminative models* | | | | | | |
| TimesNet | $1.393_{\pm0.038}$ | $1.044_{\pm0.065}$ | $69.040_{\pm4.305}$ | $1.485_{\pm0.044}$ | $1.154_{\pm0.068}$ | $72.065_{\pm4.222}$ |
| TimeMixer++ | $1.965_{\pm0.012}$ | $1.642_{\pm0.025}$ | $97.093_{\pm0.015}$ | $1.283_{\pm0.015}$ | $1.100_{\pm0.032}$ | $94.319_{\pm0.028}$ |
| BRITS | $1.461_{\pm0.048}$ | $0.992_{\pm0.037}$ | $58.600_{\pm0.022}$ | $0.850_{\pm0.020}$ | $0.491_{\pm0.008}$ | $42.067_{\pm0.007}$ |
| SAITS | $1.247_{\pm0.069}$ | $0.814_{\pm0.046}$ | $48.119_{\pm0.027}$ | $0.626_{\pm0.018}$ | $0.366_{\pm0.014}$ | $31.417_{\pm0.012}$ |
| *Generative models* | | | | | | |
| GP-VAE | $1.915_{\pm0.006}$ | $1.511_{\pm0.011}$ | $89.315_{\pm0.638}$ | $1.147_{\pm0.008}$ | $0.896_{\pm0.018}$ | $76.809_{\pm1.507}$ |
| not-MIWAE | $1.781_{\pm0.012}$ | $1.311_{\pm0.016}$ | $77.512_{\pm0.972}$ | $0.945_{\pm0.010}$ | $0.637_{\pm0.011}$ | $54.643_{\pm0.918}$ |
| *Diffusion-based models* | | | | | | |
| CSDI | $1.658_{\pm0.001}$ | $1.071_{\pm0.001}$ | $63.254_{\pm0.038}$ | $0.822_{\pm0.001}$ | $0.522_{\pm0.001}$ | $44.733_{\pm0.049}$ |
| MTSCI | $1.335_{\pm0.001}$ | $0.957_{\pm0.001}$ | $56.574_{\pm0.036}$ | $0.730_{\pm0.000}$ | $0.500_{\pm0.000}$ | $42.827_{\pm0.018}$ |
| cDiffPuter | $\underline{1.209}_{\pm0.001}$ | $\underline{0.782}_{\pm0.000}$ | $\underline{46.188}_{\pm0.020}$ | $\underline{0.612}_{\pm0.001}$ | $\underline{0.362}_{\pm0.000}$ | $\underline{31.069}_{\pm0.020}$ |
| **PRDIM** | $\mathbf{1.057}_{\pm0.000}$ | $\mathbf{0.663}_{\pm0.000}$ | $\mathbf{39.156}_{\pm0.009}$ | $\mathbf{0.538}_{\pm0.000}$ | $\mathbf{0.303}_{\pm0.000}$ | $\mathbf{25.986}_{\pm0.015}$ |

Table 11: Overall performance on the **STOCK** dataset. We report $mean \pm std$ over 5 runs according to each methodology. Best results are in **bold**. Second best results are in underline.

| Method | Original / Out-of-Sample | | | Original / In-Sample | | |
|---|---|---|---|---|---|---|
| | RMSE | MAE | MRE | RMSE | MAE | MRE |
| Mean | $2.079_{\pm0.000}$ | $1.949_{\pm0.000}$ | $128.903_{\pm0.000}$ | $2.168_{\pm0.000}$ | $2.039_{\pm0.000}$ | $127.313_{\pm0.000}$ |
| *Discriminative models* | | | | | | |
| TimesNet | $1.415_{\pm0.054}$ | $1.111_{\pm0.073}$ | $73.528_{\pm0.049}$ | $1.509_{\pm0.057}$ | $1.221_{\pm0.077}$ | $76.237_{\pm0.048}$ |
| TimeMixer++ | $1.490_{\pm0.223}$ | $1.287_{\pm0.239}$ | $85.153_{\pm0.158}$ | $1.569_{\pm0.239}$ | $1.369_{\pm0.260}$ | $85.456_{\pm0.162}$ |
| BRITS | $0.953_{\pm0.016}$ | $0.627_{\pm0.010}$ | $41.478_{\pm0.006}$ | $1.020_{\pm0.016}$ | $0.701_{\pm0.010}$ | $43.757_{\pm0.006}$ |
| SAITS | $0.743_{\pm0.021}$ | $0.442_{\pm0.022}$ | $29.115_{\pm0.015}$ | $0.801_{\pm0.023}$ | $0.498_{\pm0.025}$ | $31.071_{\pm0.016}$ |
| *Generative models* | | | | | | |
| GP-VAE | $1.239_{\pm0.118}$ | $0.902_{\pm0.109}$ | $59.684_{\pm7.220}$ | $1.333_{\pm0.123}$ | $1.010_{\pm0.118}$ | $63.046_{\pm7.371}$ |
| not-MIWAE | $1.028_{\pm0.043}$ | $0.681_{\pm0.045}$ | $45.039_{\pm0.296}$ | $1.114_{\pm0.043}$ | $0.759_{\pm0.039}$ | $47.368_{\pm0.243}$ |
| *Diffusion-based models* | | | | | | |
| CSDI | $0.932_{\pm0.000}$ | $0.641_{\pm0.000}$ | $42.393_{\pm0.004}$ | $0.995_{\pm0.000}$ | $0.710_{\pm0.000}$ | $44.330_{\pm0.001}$ |
| MTSCI | $0.988_{\pm0.002}$ | $0.736_{\pm0.001}$ | $48.629_{\pm0.009}$ | $1.056_{\pm0.001}$ | $0.809_{\pm0.001}$ | $50.485_{\pm0.041}$ |
| cDiffPuter | $\underline{0.734}_{\pm0.000}$ | $\underline{0.406}_{\pm0.000}$ | $\underline{26.878}_{\pm0.100}$ | $\underline{0.778}_{\pm0.000}$ | $\underline{0.450}_{\pm0.000}$ | $28.064_{\pm0.008}$ |
| **PRDIM** | $\mathbf{0.599}_{\pm0.001}$ | $\mathbf{0.254}_{\pm0.000}$ | $\mathbf{16.794}_{\pm0.027}$ | $\mathbf{0.633}_{\pm0.000}$ | $\mathbf{0.275}_{\pm0.000}$ | $\mathbf{17.150}_{\pm0.008}$ |

## C.2 ADDITIONAL EXPERIMENTS

### C.2.1 FASHION-MNIST

In the image domain, missing values are not limited to the MNAR mechanism demonstrated in the main experiment. As a representative example in the image dataset, we additionally exhibit imputation results under the *block missing* mechanism. Experiments are conducted on the FMNIST dataset, and we further include comparisons with representative GAN-based and Flow-based imputation approaches, namely misGAN (Li et al., 2019) and MCFlow (Richardson et al., 2020).

As illustrated by the following Figure 9, among the three methods, our proposed PRDIM most effectively captures the underlying object structure and achieves the most faithful reconstructions. These findings demonstrate that PRDIM can generalize beyond MNAR to handle other types of missingness, such as block-MAR, while retaining its ability to generate semantically plausible imputations.

Table 12: Overall performance on the **PEMS-Bay** dataset. We report $mean \pm std$ over 5 runs according to each methodology. Best results are in **bold**. Second best results are in underline.

| Method | Original / Out-of-Sample | | | Original / In-Sample | | |
|---|---|---|---|---|---|---|
| | RMSE | MAE | MRE | RMSE | MAE | MRE |
| Mean | $0.901_{\pm0.000}$ | $0.813_{\pm0.000}$ | $119.064_{\pm0.000}$ | $0.868_{\pm0.000}$ | $0.789_{\pm0.000}$ | $119.066_{\pm0.000}$ |
| *Discriminative models* | | | | | | |
| TimesNet | $0.481_{\pm0.009}$ | $0.291_{\pm0.007}$ | $42.579_{\pm0.010}$ | $0.392_{\pm0.004}$ | $0.225_{\pm0.001}$ | $33.970_{\pm0.002}$ |
| TimeMixer++ | $0.684_{\pm0.013}$ | $0.579_{\pm0.018}$ | $84.816_{\pm0.026}$ | $0.652_{\pm0.015}$ | $0.557_{\pm0.020}$ | $84.113_{\pm0.030}$ |
| BRITS | $0.503_{\pm0.007}$ | $0.278_{\pm0.006}$ | $40.758_{\pm0.008}$ | $0.342_{\pm0.004}$ | $0.182_{\pm0.003}$ | $27.490_{\pm0.004}$ |
| SAITS | $0.481_{\pm0.011}$ | $0.302_{\pm0.009}$ | $44.266_{\pm0.013}$ | $0.356_{\pm0.004}$ | $0.212_{\pm0.003}$ | $31.970_{\pm0.005}$ |
| *Generative models* | | | | | | |
| GP-VAE | $0.537_{\pm0.001}$ | $0.345_{\pm0.001}$ | $50.561_{\pm0.208}$ | $0.470_{\pm0.003}$ | $0.292_{\pm0.002}$ | $44.039_{\pm0.367}$ |
| not-MIWAE | $0.623_{\pm0.006}$ | $0.396_{\pm0.005}$ | $57.510_{\pm0.774}$ | $0.608_{\pm0.004}$ | $0.352_{\pm0.005}$ | $53.181_{\pm0.746}$ |
| *Diffusion-based models* | | | | | | |
| CSDI | $\underline{0.338}_{\pm0.002}$ | $\underline{0.177}_{\pm0.000}$ | $\underline{25.912}_{\pm0.017}$ | $\mathbf{0.302}_{\pm0.000}$ | $\underline{0.158}_{\pm0.000}$ | $\underline{23.910}_{\pm0.005}$ |
| MTSCI | $0.349_{\pm0.000}$ | $0.193_{\pm0.000}$ | $28.289_{\pm0.011}$ | $0.322_{\pm0.000}$ | $0.179_{\pm0.000}$ | $27.017_{\pm0.003}$ |
| cDiffPuter | $0.349_{\pm0.007}$ | $0.182_{\pm0.000}$ | $26.714_{\pm0.011}$ | $0.330_{\pm0.000}$ | $0.168_{\pm0.000}$ | $25.377_{\pm0.005}$ |
| **PRDIM** | $\mathbf{0.334}_{\pm0.002}$ | $\mathbf{0.170}_{\pm0.000}$ | $\mathbf{24.966}_{\pm0.015}$ | $\underline{0.306}_{\pm0.000}$ | $\mathbf{0.154}_{\pm0.000}$ | $\mathbf{23.304}_{\pm0.006}$ |

Furthermore, to highlight the general imputation ability of our model under the MNAR mechanism from the main experiment, we also present additional qualitative results in Figure 10. For both experiments, we evaluate the quality of generated samples using the Fréchet Inception Distance (FID) (Heusel et al., 2017), which is computed with the released Python library (Seitzer, 2020).

### C.2.2 CELEBA-HQ

To verify the scalability of PRDIM on high-dimensional data, we conducted an additional imputation experiment on the RGB image benchmark dataset named CelebA-HQ (Lee et al., 2020). We compared our method with a vanilla diffusion model trained under the CSDI objective. Each image in CelebA-HQ is accompanied by corresponding annotation mask vectors that label facial attributes such as eyes, nose, mouth, and hair. To design an incomplete dataset under an MNAR pattern, we utilized the annotation masks of eyes, nose, and mouth to construct a missing-value mask. Specifically, for each facial attribute, we introduced missing pixels with an 80% probability within the annotated regions, forming an MNAR missing mechanism for the experiment.

To efficiently manage the training time of the EM-based PRDIM model trained from a scratch, both images and their corresponding mask vectors were resized to a resolution of 64×64. The original CelebA-HQ dataset consists of 1024×1024 images and 512×512 annotation masks.

Qualitative results are shown in Figure 11. The vanilla diffusion model trained with the CSDI objective tends to fill in missing areas with averaged color tones around the missing regions, resulting in naive reconstructions and relatively high FID scores despite a moderate missing ratio. In contrast, PRDIM generates more detailed and realistic facial structures, accurately reconstructing attribute boundaries and color variations, which leads to significantly improved perceptual quality and lower FID values.

### C.2.3 TABULAR DATA

To further demonstrate the generalization capability of PRDIM, we reproduced the official implementation of DiffPuter[1] and compared its performance with PRDIM under the MNAR setting.

In this experiment, we followed the practical implementation of DiffPuter with default configuration regardless to dataset, which differs from the main experiments in that incomplete samples were not used as conditional information during imputation.

---

[1] https://github.com/hengruizhang98/DiffPuter

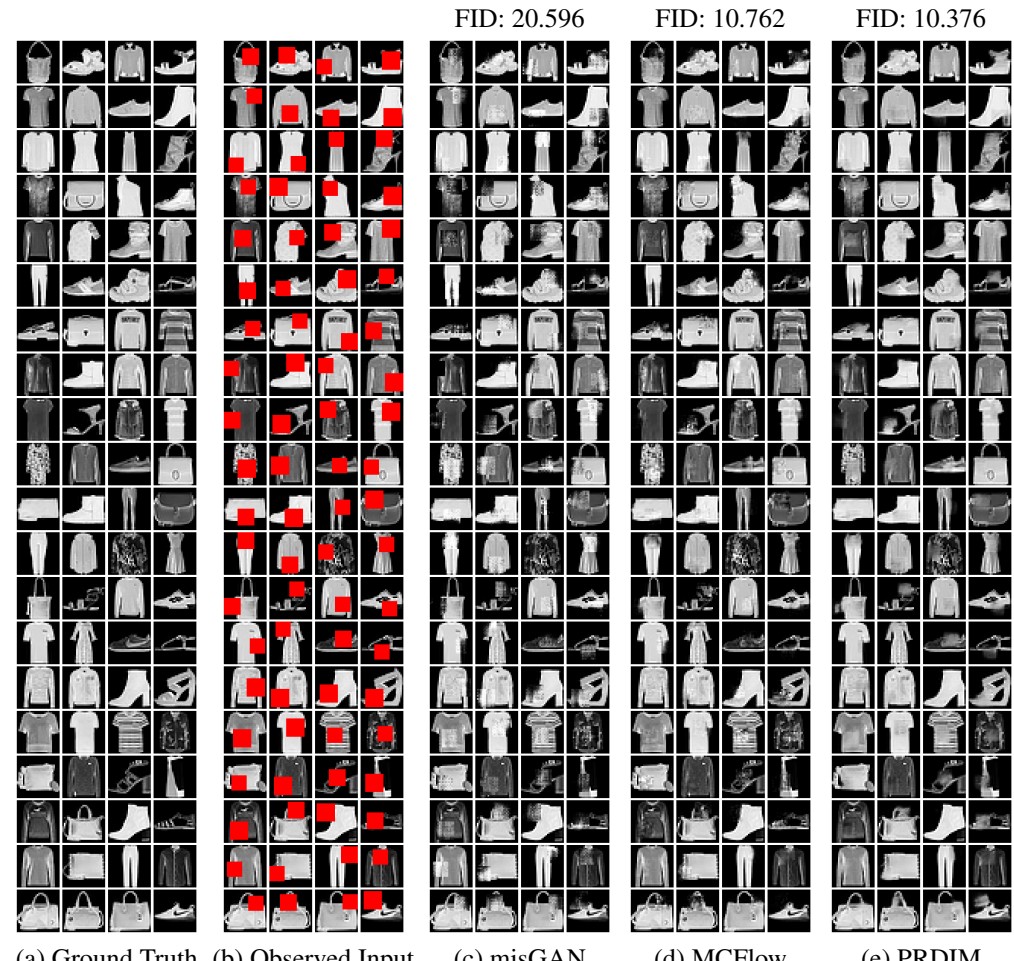

FID: 20.596  FID: 10.762  FID: 10.376

(a) Ground Truth  (b) Observed Input  (c) misGAN  (d) MCFlow  (e) PRDIM

Figure 9: Comparison of out-of-sample imputation results under block missing pattern.

Table 13: In-sample imputation results across tabular datasets. We report the mean ± std over five different training/test split combinations.

|  | adult | bean | default | gesture | magic |
|---|---|---|---|---|---|
| DiffPuter | 0.497±0.013 | 0.240±0.069 | 0.374±0.092 | **0.391±0.023** | 0.539±0.088 |
| PRDIM | **0.474±0.012** | **0.199±0.058** | **0.336±0.074** | 0.394±0.029 | **0.490±0.083** |

We selected 5 different tabular datasets available from the UCI Machine Learning Repository. Among them, bean, gesture, and magic consist solely of continuous features, while adult and default contain both continuous and discrete features. The discrete attributes were label-encoded to preserve the original data dimensionality, and the corresponding mask vectors were designed to match this structure.

Tables 13 and 14 present the in-sample and out-of-sample imputation performance of DiffPuter and PRDIM, respectively. Across most datasets, PRDIM achieves more accurate imputations, validating its robustness and adaptability across different data modalities.

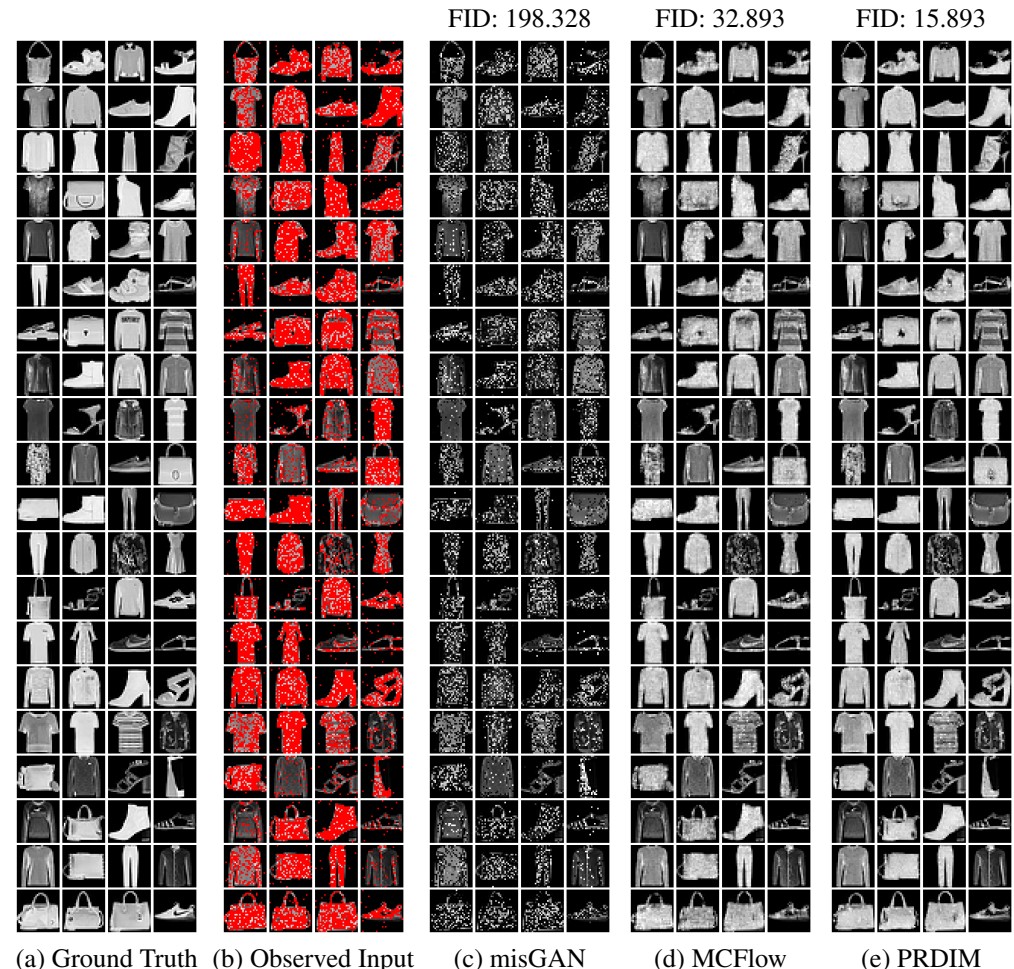

(a) Ground Truth    (b) Observed Input    (c) misGAN    (d) MCFlow    (e) PRDIM

Figure 10: Comparison of out-of-sample imputation results under MNAR missing pattern.

Table 14: Out-of-sample imputation results across tabular datasets. We report the mean $\pm$ std over five different training/test split combinations.

|  | adult | bean | default | gesture | magic |
|---|---|---|---|---|---|
| DiffPuter | 0.504±0.012 | 0.219±0.053 | 0.315±0.040 | **0.353±0.007** | 0.539±0.049 |
| PRDIM | **0.482±0.022** | **0.199±0.053** | **0.279±0.039** | 0.371±0.052 | **0.488±0.047** |

### C.2.4 INTERPRETABILITY OF THE PATTERN RECOGNIZER WITH CASE STUDY

To evaluate whether the pattern recognizer $D_\phi$ trained under the EM iterations has effectively learned the underlying missing mechanism, we conducted a case study on three time-series datasets: ETT, STOCK, and PEMS-Bay. Specifically, we randomly sampled instances from the ETT dataset and plotted both the true missing ratio for each entry and the corresponding output of the trained pattern recognizer, $D_\phi(\hat{X}_0)$, where $\hat{X}_0$ represents the imputed sample generated through approximate guided generation by PRDIM. This visualization allows us to examine whether the learned $D_\phi$ accurately captures and mimics the MNAR missing patterns inherent in high dimensional data.

For each dataset, we sampled time intervals of length 72. The ETT and STOCK datasets contain 7 and 6 features, respectively, and we visualized all features for completeness. In the case of the PEMS-Bay dataset, which has a total of 325 feature dimensions, only the first 10 features were used

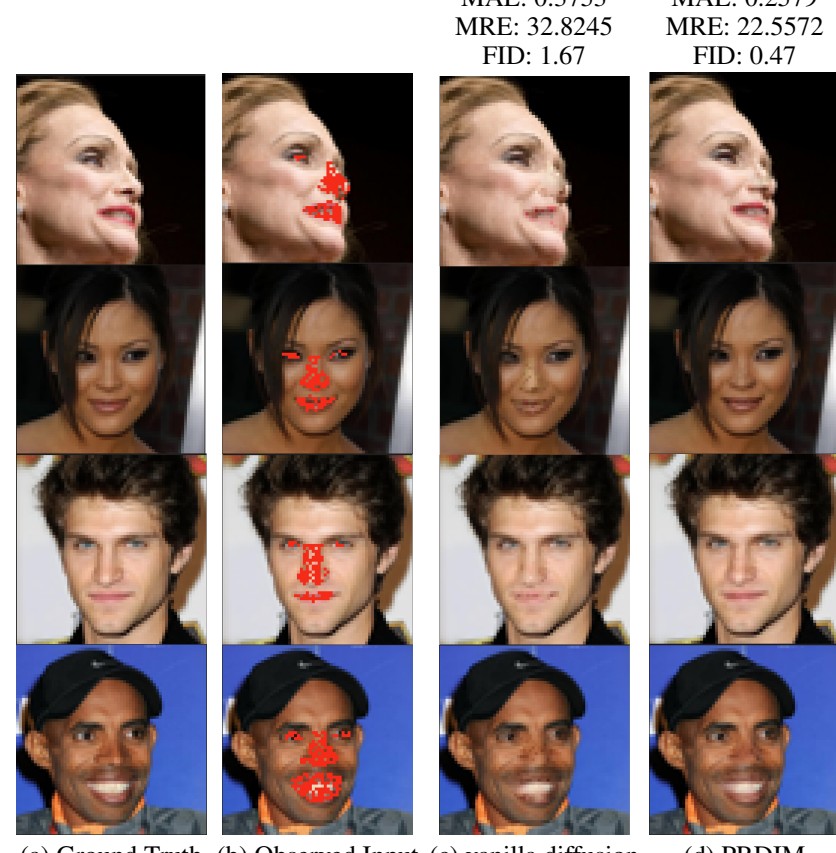

MAE: 0.3753  MAE: 0.2579
MRE: 32.8245  MRE: 22.5572
FID: 1.67  FID: 0.47

(a) Ground Truth (b) Observed Input (c) vanilla diffusion (d) PRDIM

Figure 11: Qualitative results of PRDIM on celebA-HQ (64×64 downsized) test dataset compared to vanilla diffusion model under custom MNAR missing pattern. Red regions in observed inputs denote missing entries under the MNAR setting.

for visualization due to its high dimensionality. The experimental results for ETT, STOCK, and PEMS-Bay are shown in Figure 13, Figrue 14, and Figure 15, 16, respectively.

Overall, the results demonstrate that while the pattern recognizer tends to slightly overestimate missing entries, it nonetheless captures the overall tendency and structure of the true missing pattern remarkably well, indicating its strong capability to model MNAR mechanisms.This interpretability analysis provides empirical evidence that the pattern recognizer contributes meaningful guidance during the generation phase.

### C.2.5 QUALITATIVE RESULTS OF PRDIM IMPUTATION

To further illustrate the behavior of PRDIM under the out-of-sample imputation setting, we provide qualitative visualizations across ETT, STOCK, and PEMS-Bay on Figure 17, 18, and 19 respectively. For each dataset, we randomly sample 4 test instances and display (i) the locations of missing values as yellow points and (ii) the corresponding imputation results produced by CSDI, MTSCI, cDiffPuter, MTSI, and PRDIM respectively. These visualization results allow for a direct visual comparison of reconstruction quality, highlighting the degree to which each model captures temporal structure and recovers unseen missing values.

To examine whether PRDIM can operate on data with naturally occurring missing values where ground-truth values for the missing entries are not available, we additionally conducted experiments on the PhysioNet (Goldberger et al., 2000) dataset. Figure 20 visualizes the out-of-sample imputation results of CSDI, cDiffPuter, and PRDIM on PhysioNet.

One notable observation is that both cDiffPuter and PRDIM involve a joint optimization procedure over the latent missing variables $X_0^{\text{mis}}$ and the observed variables $X_0^{\text{obs}}$ during the EM iterations. As a consequence, when the natural missing rate is extremely high (approaching nearly 80% in the PhysioNet dataset), the imputed values may become biased toward zero (i.e. initial imputed value). This highlights an inherent limitation of EM-based diffusion imputation methods under severe natural missingness.

## D    DETAILED DESCRIPTION OF DATA PROCESSING AND OBJECTIVE

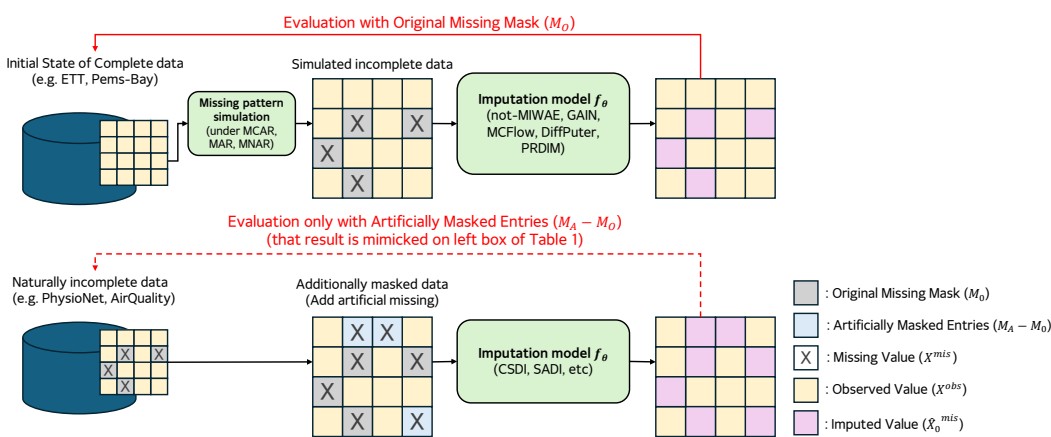

Figure 12: Overview of the data processing pipeline and the distinction between two classes of imputation objectives.

In this section, we aim to clarify the rationale behind our choice of datasets, draw theoretical connections to the EM-based training procedure adopted in PRDIM, and contrast our evaluation protocol with that of prior studies that share a similar experimental framework. Figure 12 provides an overview of two distinct classes of objectives used in existing imputation research, highlighting why directly evaluating models on naturally incomplete datasets such as PhysioNet (Goldberger et al., 2000) or AirQuality (Zhang et al., 2017) can be problematic.

Imputation applicable diffusion models including CSDI (Tashiro et al., 2021), SSSD (Alcaraz & Strodthoff, 2022), and Diffusion-TS (Yuan & Qiao, 2024), generally rely on one of two strategies. (i) introducing artificial missingness into a complete dataset so that ground-truth values are available during training, or (ii) injecting additional artificial missingness into already incomplete datasets, thereby increasing the overall missing ratio and using the resulting data as model input. A key commonality between the two imputation paradigms is that the ground-truth values employed for evaluation are implicitly utilized during model training.

Let $M_O$ denote the original missing mask of the incomplete dataset $X_0^{\text{obs}}$, and let $M_A$ denote the mask obtained after applying additional artificial missingness. The missingness distributions induced by these two masks differ intrinsically, which can be formalized as $p(M_O|X_0) \neq p(M_A|X_0, M_O)$. Consequently, the imputed results generated under these differing mask conditions also diverge $p_\theta(X_0|M_O, X_0^{\text{obs}}) \neq p_\theta(X_0|M_A, X_0^{\text{obs},A})$. Such discrepancies indicate that the imputation task inevitably involves a latent missing variable $X_0^{\text{mis}}$, whose distribution cannot be directly inferred from artificially masked data alone. This observation motivates the necessity of adopting an Expectation–Maximization (EM) training framework, wherein the missing entries are treated as latent variables and iteratively refined during model optimization.

## E    THE USE OF LARGE LANGUAGE MODELS

We've got some help by Large Language Models (LLMs) only in the areas of translation and grammar examination. The core research ideation and all theoretical statements are our own work.

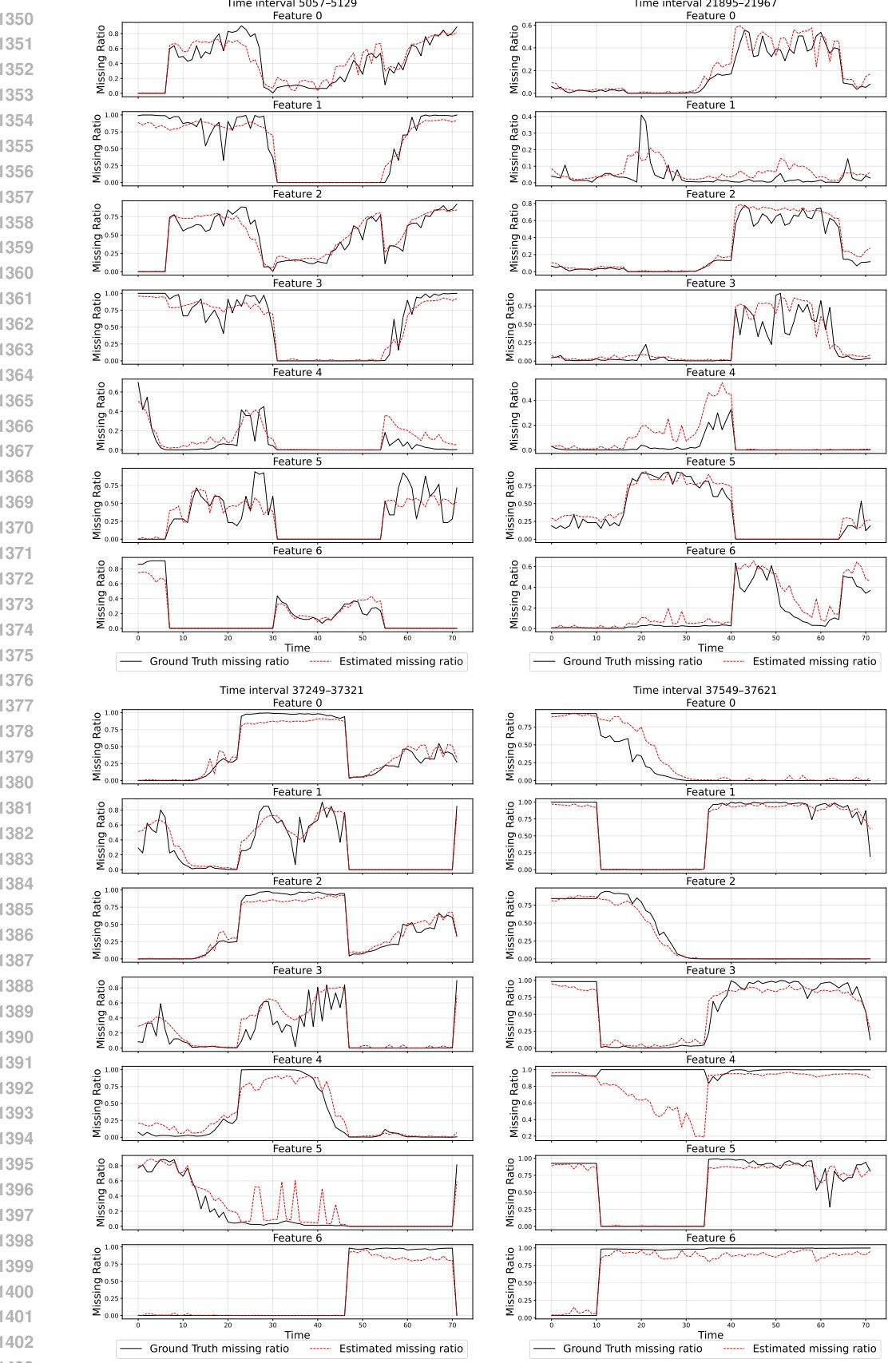

Figure 13: Randomly sampled 4 ETT segments with time length 72: Ground-truth missing ratio (black) versus Pattern Recognizer-estimated missing ratio $D_\phi(\hat{X}_0)$ across 7 features.

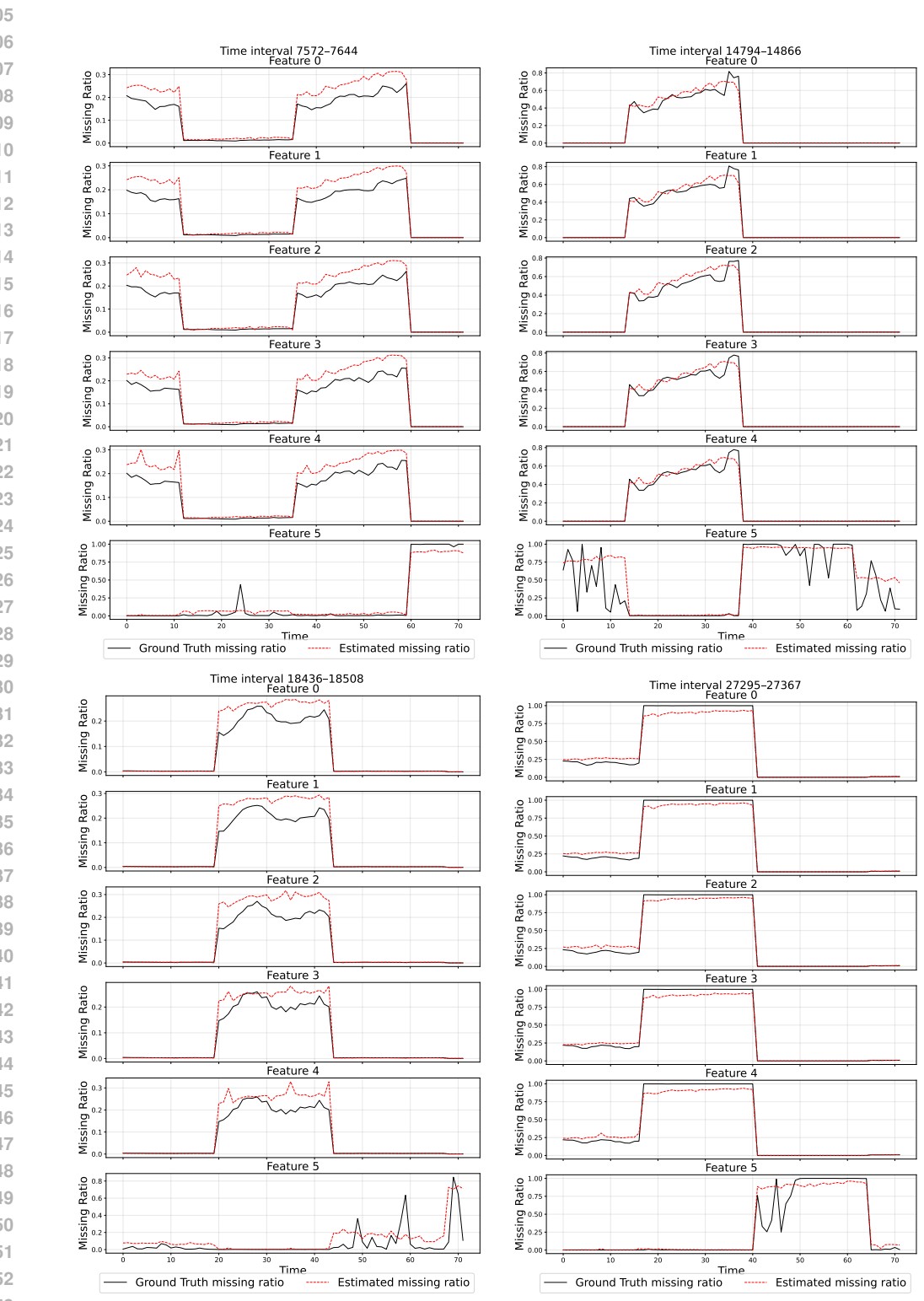

Figure 14: Randomly sampled 4 STOCK segments with time length 72: Ground-truth missing ratio (black) versus Pattern Recognizer-estimated missing ratio $D_\phi(\hat{X}_0)$ across 6 features.

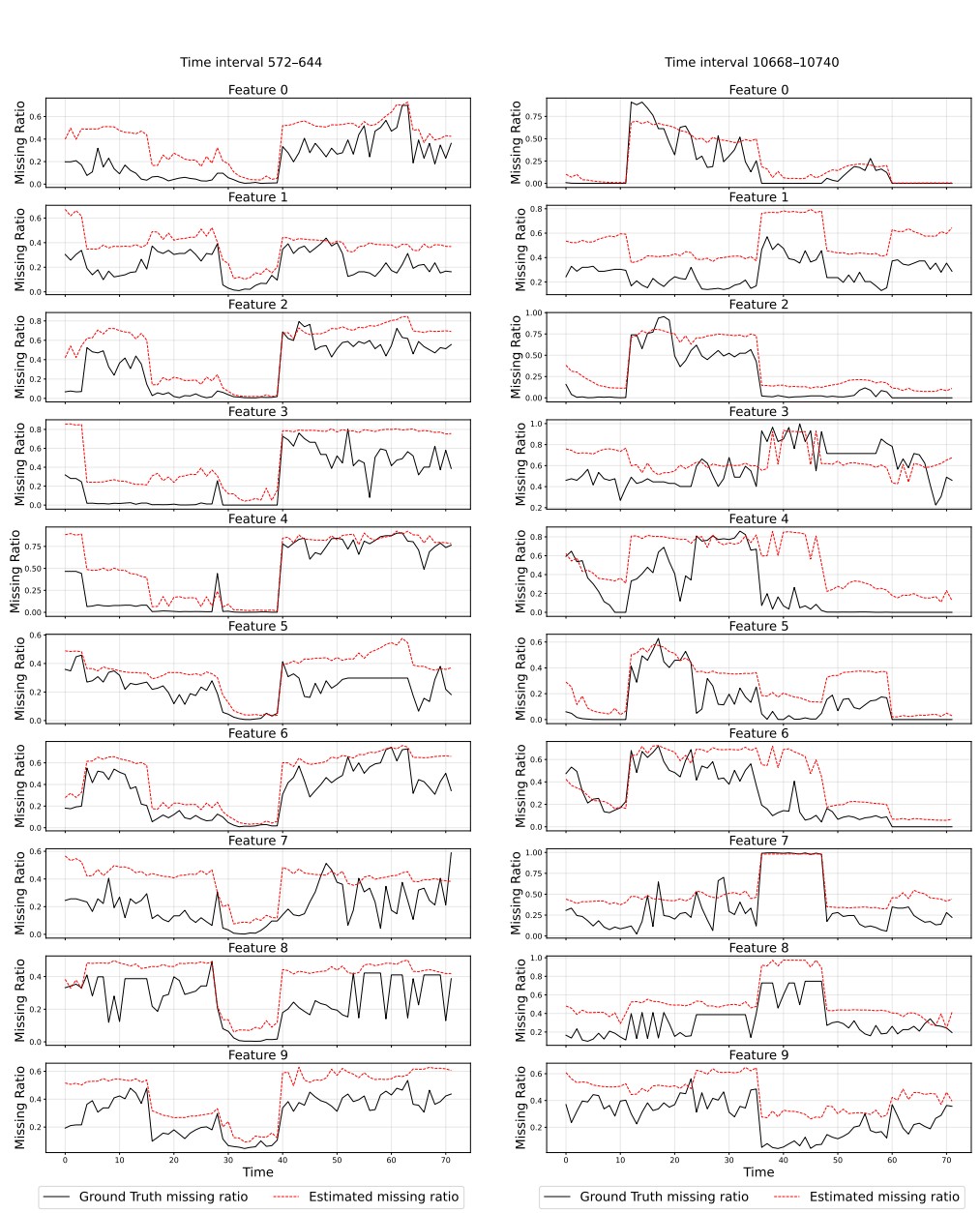

Figure 15: Randomly sampled 2 PEMS-Bay segments with time length 72: Ground-truth missing ratio (black) versus Pattern Recognizer-estimated missing ratio $D_\phi(\hat{X}_0)$ across 10 of 325 features.

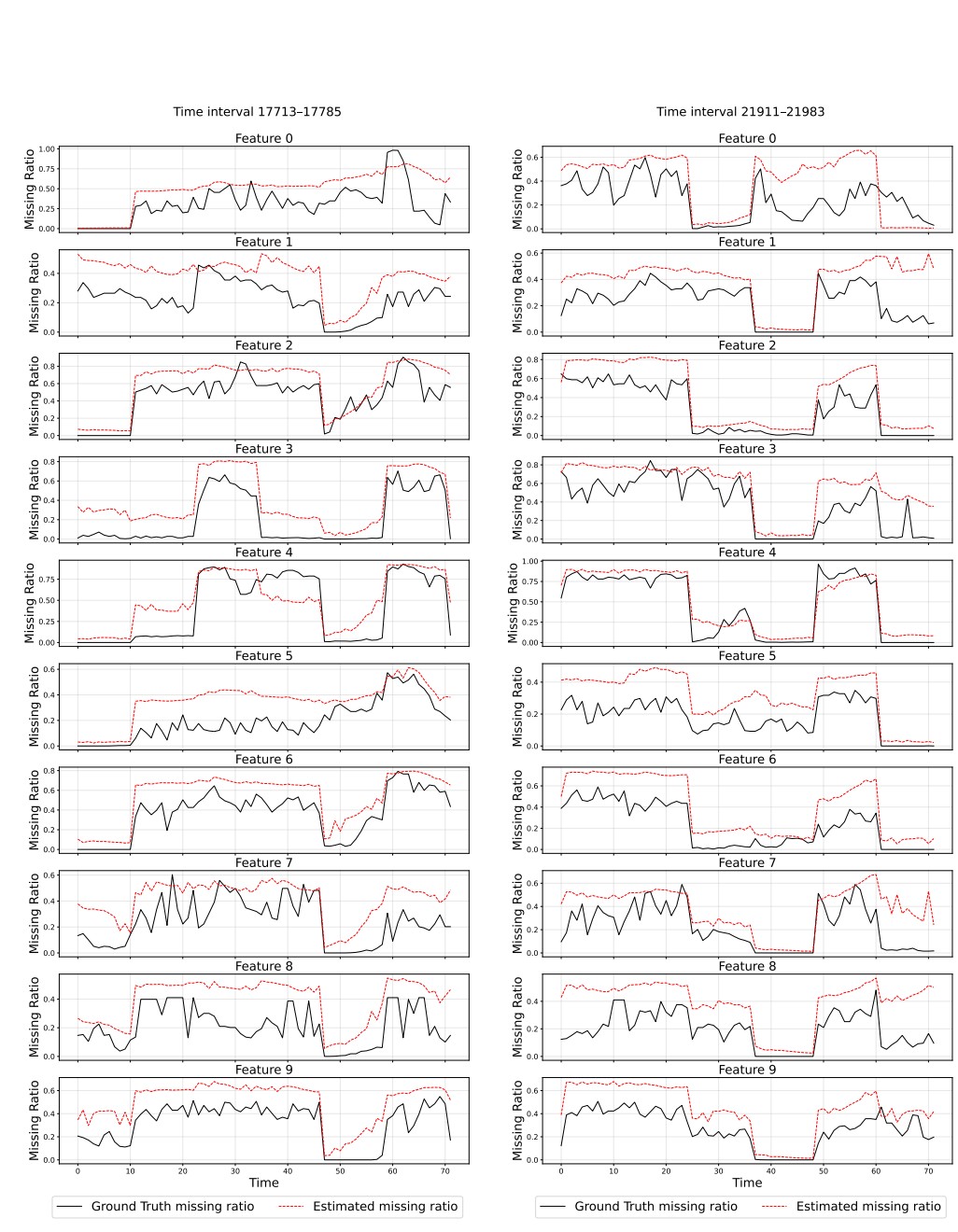

Figure 16: Additional 2 random samples of 2 PEMS-Bay segments with time length 72: Ground-truth missing ratio (black) versus Pattern Recognizer-estimated missing ratio $D_\phi(\hat{X}_0)$ across 10 of 325 features.

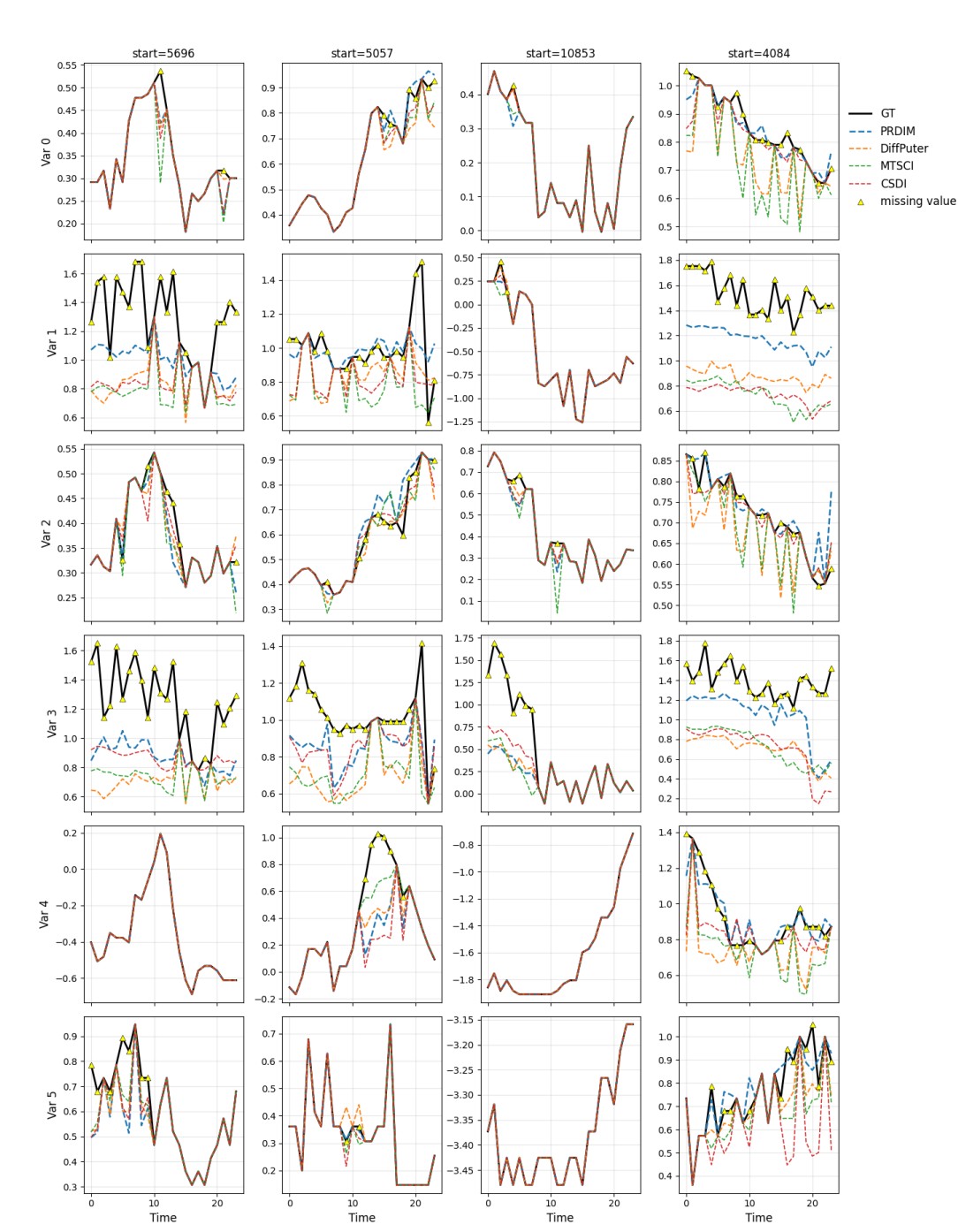

Figure 17: Qualitative results of PRDIM compared to other diffusion imputation models. 4 randomly selected imputed out-of-samples from the ETT dataset are visualized. Each panel labeled start = n corresponds to the time interval (n,n+24).

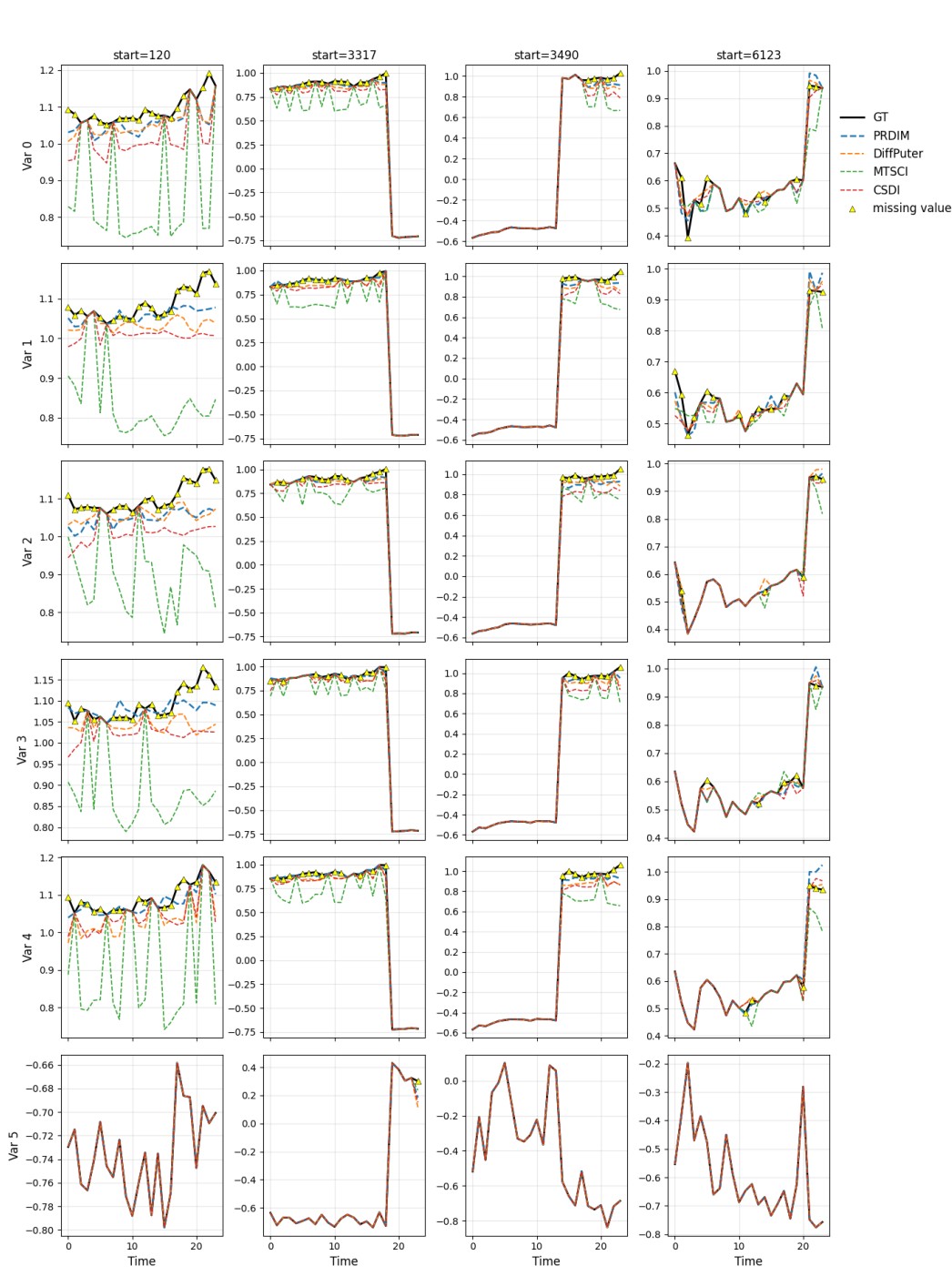

Figure 18: Qualitative results of PRDIM compared to other diffusion imputation models. 4 randomly selected imputed out-of-samples from the STOCK dataset are visualized. Each panel labeled start = n corresponds to the time interval (n,n+24).

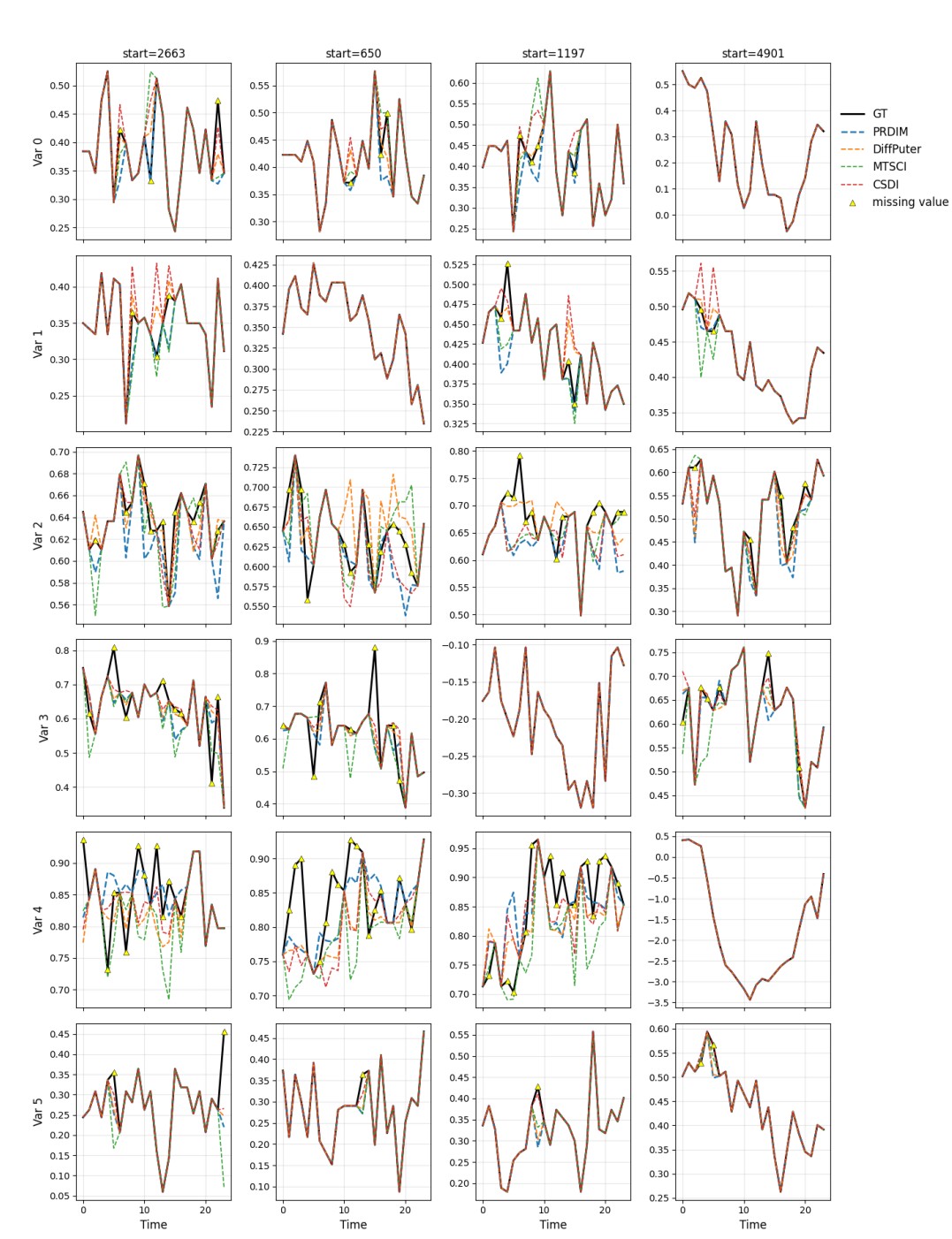

Figure 19: Qualitative results of PRDIM compared to other diffusion imputation models. 4 randomly selected imputed out-of-samples from the PEMS-Bay dataset are visualized. Each panel labeled start = n corresponds to the time interval (n,n+24).

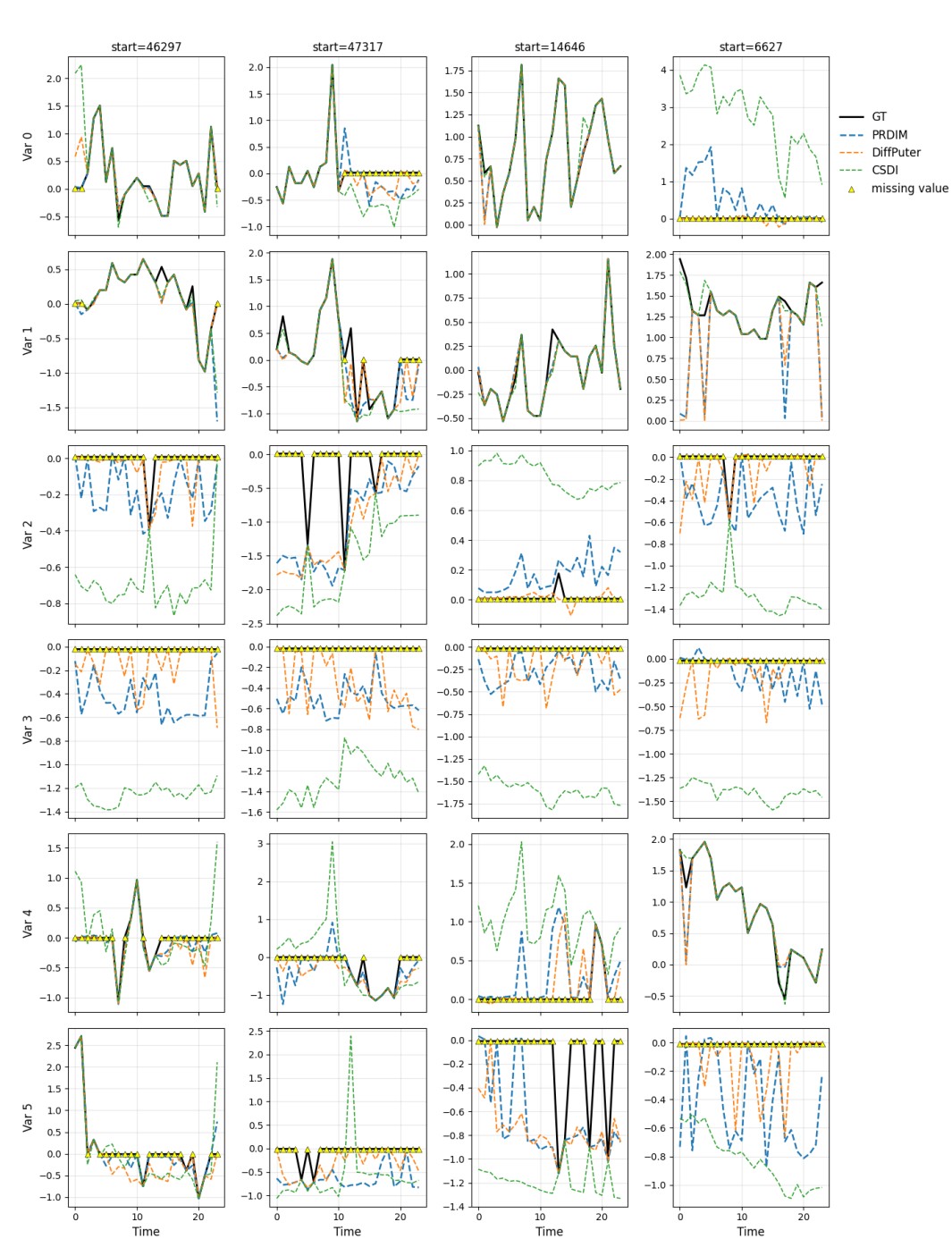

Figure 20: Qualitative results of PRDIM compared to other diffusion imputation models. 4 randomly selected imputed out-of-samples from the PhysioNet dataset are visualized. Each panel labeled start = n corresponds to the time interval (n,n+24).

