# OpenReview forum: "Missing Pattern Recognized Diffusion Imputation Model for Missing Not at Random"
_ICLR.cc/2026/Conference — Submitted to ICLR 2026_

### Official Review · Reviewer_pWjT · 2025-10-16

**Soundness:** 3
**Presentation:** 3
**Contribution:** 3
**Rating:** 6
**Confidence:** 4

**Summary:**

To address the limitations of existing diffusion-based imputation models that overlook MNAR (Missing Not At Random) scenarios and rely on strong assumptions about missing patterns, this paper proposes PRDIM. The method introduces a pattern recognizer capable of identifying missing patterns in the data and employs the Expectation-Maximization (EM) algorithm to iteratively maximize the likelihood of the joint distribution of observed values and the missing mask. Extensive experiments on multiple datasets are conducted to verify the effectiveness of the proposed model.

**Strengths:**

S1: This paper conducts extensive experiments, covering multiple baselines, diverse datasets, and various imputation scenarios.

S2: This paper uses clear notations, solid theoretical derivations, and well-crafted figures to effectively illustrate its ideas.

**Weaknesses:**

W1: The experiments are conducted by simulating MNAR missing patterns on complete real-world datasets, rather than using datasets that contain naturally occurring missing values.

W2: As the proposed method belongs to the class of diffusion-based models, it lacks evaluation metrics such as CRPS to assess the quality of imputation results.

**Questions:**

Q1: Could the authors explain why missing patterns in real-world data are considered to be more consistent with the MNAR assumption? Is it possible to conduct experiments on datasets with naturally occurring missing values to demonstrate that the simulated MNAR patterns resemble natural ones? Alternatively, could the naturally missing patterns be used to artificially simulate missing values for evaluation?

Q2: This paper focuses on the MNAR scenario and notes in Table 5 that the advantage of PRDIM diminishes under the MCAR setting. Could the authors further explore the model’s performance under the MAR scenario? In addition, it would be helpful to clarify in which types of missingness patterns or data distributions the proposed pattern recognizer is applicable, and under what circumstances it may not be effective. Please further discuss the limitations of PRDIM in such contexts.

Q3: In Table 1, how is the imputation error computed for the original missing entries, and how is the ground truth obtained? In addition, could the authors clarify how the out-of-sample and in-sample imputation settings are specifically configured in the experiments?

---

> ### Author Response · Authors · 2025-11-21
>
> We would like to thank Reviewer **pWjT** for raising questions regarding the lacking experimental results. Based on these comments, we have added more detailed explanations in both below responses and revised paper.
>
> > **W1**.  The experiments are conducted by simulating MNAR missing patterns on complete real-world datasets, rather than using datasets that contain naturally occurring missing values.
> > **Q3**. In Table 1, how is the imputation error computed for the original missing entries, and how is the ground truth obtained? In addition, could the authors clarify how the out-of-sample and in-sample imputation settings are specifically configured in the experiments?
>
> Thank you for raising these important questions regarding the core motivation of our study. We acknowledge that the distinction between _artificial missing entries_ and _original missing entries_, as well as the definitions of _in-sample_ and _out-of-sample_ imputation tasks, was not sufficiently detailed in the Introduction. Reviewer **7n6G** pointed out similar concerns, and to address this, we have added a new **Appendix D** with Figure 12 in the revised version, where we provide a more thorough explanation of these terms and the motivation behind our design choices.
>
> **Appendix D Figure 12** of the revised paper, illustrates the full pipeline of how prior works process data for imputation.  We intentionally avoided using datasets such as PhysioNet or AirQuality datasets that contain naturally occurring missing values for quantitative evaluation. Nevertheless, for completeness, we additionally provide qualitative visualizations of the imputation performance of CSDI, DiffPuter, and PRDIM in Figure 20. As shown in our response to Reviewer 7n6G (Weakness 1), the original missing probability $p(M_O|X_0)$ differs substantially from the artifical missing probability $p(M_A|X_0,M_O)$. Consequently, a diffusion imputation model conditioned on artificially missing entries satisfies $p_\theta(X_0|M_A)\neq p_\theta(X_0|M_O)$ meaning that evaluation on artificial masks does not reflect the true missing pattern, and evaluation on original missing entries lacks accessible ground truth. This is why we chose complete datasets (ETT, image datasets) where we can explicitly control the missing mechanism and ensure fair comparisons across all baselines. This same perspective is adopted in several prior works (e.g., GAIN [1], not-MIWAE [2], MCFlow [3], and DiffPuter [4]) all of which evaluate imputation using complete data with synthetic missing.
>
> In our paper, In-sample imputation means Imputation on data that the model encounters during training (i.e., the incomplete training set). Out-of-sample imputation task is the imputation on unseen data (validation and test sets) that the model never observes during training. This evaluates generalization to new samples and tests the robustness of the learned pattern recognizer. We have added a detailed explanation of these settings in the end paragraph of **Appendix C** with Figure 8 in the revised version. We appreciate the reviewer for highlighting these conceptual points.
>
> [1] Yoon, Jinsung, James Jordon, and Mihaela Schaar. "Gain: Missing data imputation using generative adversarial nets." _International conference on machine learning_. PMLR, 2018.
>
> [2] Ipsen, Niels Bruun, Pierre-Alexandre Mattei, and Jes Frellsen. "not-MIWAE: Deep generative modelling with missing not at random data." _arXiv preprint arXiv:2006.12871_ (2020).
>
> [3] Richardson, Trevor W., et al. "Mcflow: Monte carlo flow models for data imputation." _Proceedings of the IEEE/CVF conference on computer vision and pattern recognition_. 2020.
>
> [4] Zhang, Hengrui, et al. "Diffputer: Empowering diffusion models for missing data imputation." _The Thirteenth International Conference on Learning Representations_. 2025.

---

> > ### Comment · Reviewer_pWjT · 2025-11-21
> >
> > Thank you for your reply. I would like to further clarify a few points:
> >
> > According to your description in Appendix D, Figure 12, it seems that the original missing entries in your paper refer to using the missing patterns extracted from a real-world dataset (which contains original missing values) to mask a complete dataset, thereby simulating missing data for evaluation purposes. Is my understanding correct?
> >
> > A minor point:
> > In the CRPS experiment table, the SOTA on the PEMS-BAY dataset should be CSDI.

---

> > > ### Author Response · Authors · 2025-11-22
> > >
> > > Dear Reviewer **pWjT**, thank you for your fast follow-up and for carefully reading our response.
> > >
> > > Regarding your first question, Figure 12 is intended to illustrate the distinction between the datasets used in our PRDIM experiments and those we chose not to use for evaluation. It does not correspond to applying the missing pattern from PhysioNet onto the PEMS-Bay dataset (for example), nor does it simulate missingness by transplanting real-world masks across datasets.
> > >
> > > We understand that your concern is fundamentally about how reliably the missing mechanisms were constructed for complete datasets such as ETT and STOCK, and whether these mechanisms meaningfully reflect real-world MNAR behavior. To address this, we clarify in the Missing Mechanisms paragraph in Appendix C that the MNAR mechanism used in the main paper follows the same formulation as in not-MIWAE [7]. Moreover, as described in our response to Reviewer **vEpr**, we additionally evaluated PRDIM under several **MNAR subtypes**. We hope these results provide further evidence that PRDIM performs robustly across diverse MNAR settings which imitate real-world scenarios.
> > >
> > > To summarize, we agree that the visual layout of Figure 12 where the **Simulated incomplete data** and **Additionally masked data** share identical illustrations may cause unintended confusion. We will revise this figure to avoid ambiguity. Also, thank you for pointing out the minor issue in the CRPS experiment table. We have corrected that indication.
> > >
> > > [7] Ipsen, Niels Bruun, Pierre-Alexandre Mattei, and Jes Frellsen. "not-MIWAE: Deep generative modelling with missing not at random data." arXiv preprint arXiv:2006.12871 (2020).

---

> > > > ### Comment · Reviewer_pWjT · 2025-11-22
> > > >
> > > > Thank you for your responses. I understand your description of Figure 12, but I'm still confused about how the performance of the "original missing entries" in Table 1 was calculated. Could you please explain the settings in Table 1 in more detail? I think the first Figure/Table in the introduction is crucial for readers' understanding.

---

> > > > > ### Author Response · Authors · 2025-11-22
> > > > >
> > > > > Thank you very much for the follow-up question. We now understand that the term _“original missing entries”_ in our paper may have caused ambiguity, especially regarding whether it refers specifically to _naturally occurring_ missing values in real-world datasets.
> > > > >
> > > > > To clarify, in our paper we used the term originally missing entries in a broader sense which include both "naturally missing" and "simulated missing".
> > > > >
> > > > > As the reviewer pointed out, there are two distinct groups of datasets:
> > > > > - Real-world incomplete datasets such as PhysioNet, Beijing AirQuality datasets. We now denote these not-accessible missing as "naturally missing".
> > > > > - Complete datasets with simulated missing (under MCAR, MAR, MNAR) such as ETT, PEMS-Bay, Fashion-MNIST, etc. We now denote these missing as "simulated missing" which can be accessible only when evaluting imputation performance.
> > > > >
> > > > > Thus, the “artificial missing entries” MAE reported in Table 1 is designed to mimic the evaluation setting used by prior works such as CSDI [8] and SADI [9] on the PhysioNet dataset, but reproduced on the ETT dataset. Consequently we use the ground-truth values for MNAR "simulated missing entries" (what we wrote as original missing entries on Table 1).
> > > > >
> > > > > In prior works that evaluate naturally incomplete datasets, we found it problematic that artificial missing is added on top of naturally occurring missing entries, since the ground truth for the natural missing values is inherently unavailable. We believe that, to properly assess imputation capability, one should instead simulate missing entries **only for evaluation**, where the ground-truth values are known, and then measure performance based on these controlled missing regions.
> > > > >
> > > > > In perspective of the reviewer’s suggestion, we believe it would be beneficial to add a dedicated subsection in the preliminary part of the paper that clearly differentiates between **original missing**, **simulated missing**,  **naturally missing**, and **artificial missing** entries. (also modifying in Figure 12) Providing these definitions upfront will help future readers develop a clearer understanding of our experimental setup. We appreciate the reviewer for raising this question regarding terminology, which allowed us to offer a more precise explanation.
> > > > >
> > > > > [8] Tashiro, Yusuke, et al. "Csdi: Conditional score-based diffusion models for probabilistic time series imputation." _Advances in neural information processing systems_ 34 (2021): 24804-24816.
> > > > >
> > > > > [9] Dai, Zongyu, Emily Getzen, and Qi Long. "Sadi: Similarity-aware diffusion model-based imputation for incomplete temporal ehr data." _International Conference on Artificial Intelligence and Statistics_. PMLR, 2024.

---

> > > > > > ### Comment · Reviewer_pWjT · 2025-11-22
> > > > > >
> > > > > > Thank you for your responses. In Table 1, the "artificial missing entries" are missing values ​​simulated according to previous work, and the "original missing entries" are missing values ​​simulated according to the MNAR mechanism in this paper. Is my understanding correct?
> > > > > >
> > > > > > Also, regarding your statement in your reply:
> > > > > > > "In prior works that evaluate naturally incomplete datasets, we found it problematic that artificial missing is added on top of naturally occurring missing entries, since the ground truth for the natural missing values ​​is inherently unavailable,"
> > > > > >
> > > > > > I don't quite agree. Even in previous work evaluating naturally incomplete datasets, the evaluation was only performed on artificially masked positions (where ground truth values ​​exist), as long as these ground truth values ​​are not leaked during the training phase. When experimenting with datasets with naturally missing data, the missing data during the training phase includes two types (one type with ground truth values, which serve as supervisory signals, and the other type being naturally missing data). The evaluation during the inference phase is also performed on the missing data with ground truth values. I don't see any problem with this. Furthermore, further experiments can be conducted: if the trained model, after imputing the naturally missing data, can improve the performance of this type of data in downstream tasks (such as classification), it indicates that the proposed imputation model can truly be used in real-world scenarios. This is my opinion.

---

> > > > > > > ### Author Response · Authors · 2025-11-23
> > > > > > >
> > > > > > > Dear Reviewer **pWjT**, we apologize for the delay in our response, as we needed additional time to conduct further experiments. We would like to address each of the reviewer’s 3 questions individually.
> > > > > > >
> > > > > > > >In Table 1, the "artificial missing entries" are missing values ​​simulated according to previous work, and the "original missing entries" are missing values ​​simulated according to the MNAR mechanism in this paper. Is my understanding correct?
> > > > > > >
> > > > > > > Yes. the reviewer’s interpretation is fully aligned with our intended meaning.
> > > > > > >
> > > > > > > >When experimenting with datasets with naturally missing data, the missing data during the training phase includes two types (one type with ground truth values, which serve as supervisory signals, and the other type being naturally missing data). The evaluation during the inference phase is also performed on the missing data with ground truth values. I don't see any problem with this.
> > > > > > >
> > > > > > > Thank you again for your thoughtful follow-up. We realize that our use of the term “problematic” may have led to some misunderstanding. To clarify our position more precisely, we would like to restate the setting using the notation introduced in the last paragraph of **Appendix D**.
> > > > > > >
> > > > > > > - For a naturally incomplete dataset, let $(X_0^{obs}, X_0^{mis})$ denote the observed and naturally missing entries, and let $M_O$ be the corresponding natural missing mask variable.
> > > > > > > - For training (in Phase1 scheme), it is common practice to impose an additional artificial mask on $X_0^{obs}$. Let $M_A$ denote this **artificial missing mask**, and $X_0^{obs,A}$ the the remaining observed entries after applying $M_A$. These artificially masked entries provide the supervision needed for training procedure and quantitative evaluation.
> > > > > > >
> > > > > > > In works such as CSDI, the diffusion model is trained to learn $p_\theta(X_0|X_0^{obs,A},M_A)$ and the same type of evaluation is conducted at test time by inserting an artificial mask into the test data. On this point, our understand is fully same with the reviewer: this evaluation protocol is valid, and we do not disagree with the correctness of these prior work. But at the same time, since naturally incomplete datasets do not provide the ground-truth values of $M_O$​, we cannot directly measure quantitative performance on $p_\theta(X_0|X_0^{obs},M_O)$.
> > > > > > >
> > > > > > > In this sense, our research question begins from a different perspective:
> > > > > > >
> > > > > > > **If the distribution of the natural missing mask $M_O$ differs substantially from the distribution of the artificial mask $M_A$, can a model trained in the regime of $p_\theta(X_0|X_0^{obs,A},M_A)$ be expected to perform well in the regime of $p_\theta(X_0|X_0^{obs},M_O)$?**
> > > > > > >
> > > > > > > This distinction is crucial to the motivation behind PRDIM. Our methodology ultimately aims to introduce a pattern recognizer, $D_{\phi}$, in order to model the distribution $p_{\theta,\phi}(X_0 \mid X_0^{\mathrm{obs}}, M_O)$ explicitly.
> > > > > > >
> > > > > > > This is why we use complete datasets and simulate the ground-truth missing mask $M_O$ according to a well-defined MNAR mechanism, rather than using datasets such as PhysioNet for quantitative analysis.
> > > > > > >
> > > > > > > Our intuition is based on the following differences in the Bayesian structure. The artificial mask $M_A$​ is generated from $p(M_A|X_0^{obs},M_O)$ which conditions only on observable elements. In contrast, the natural missing mask $M_O$​ arises from $p(M_O|X_0^{obs},X_0^{mis})$, a fundamentally different process that depends also on unobserved variables.
> > > > > > >
> > > > > > > Thus, in general we assume $p_\theta(X_0|X_0^{obs,A},M_A)\neq p_\theta(X_0|X_0^{obs},M_O)$. This is precisely the gap that PRDIM attempts to address the underlying missing pattern. we want to explicitly model the natural missing mechanism $p(M_O|X_0^{obs},X_0^{mis})$ using the pattern recognizer $D_\phi$, which introduces a new modeling capability absent in prior diffusion-based imputation methods. For comparison, not-MIWAE addresses this problem within the VAE framework by maximizing the ELBO of $\log{p(X_0,M_O)}$ in closed form. For diffusion models, however, iterative sampling renders such computation intractable, so the EM formulation becomes necessary.
> > > > > > >
> > > > > > > In short, we agree that the objectives of prior works are valid and correctly formulated. Our difference in dataset choice arises because our focus is on evaluating whether an imputation model can truly recover missing values under original MNAR mechanisms, i.e., whether $p_{\theta,\phi}(X_0|X_0^{obs}, M_O)$ is accurately modeled. We appreciate the reviewer’s thoughtful critique.

---

> > > > > > > > ### Author Response · Authors · 2025-11-23
> > > > > > > >
> > > > > > > > > Furthermore, further experiments can be conducted: if the trained model, after imputing the naturally missing data, can improve the performance of this type of data in downstream tasks (such as classification), it indicates that the proposed imputation model can truly be used in real-world scenarios.
> > > > > > > >
> > > > > > > > Lastly, Thank you for suggesting an additional experiment involving post-imputation downstream tasks. We agree that this type of evaluation can more directly demonstrate the applicability of PRDIM in real-world scenarios.
> > > > > > > >
> > > > > > > > To enable a clear and controlled comparison, we designed a downstream classification experiment using the Fashion-MNIST (FMNIST) dataset. We imputed all 60,000 training samples using the three baseline generative imputation models compared in the main text: MCFlow, CSDI, and misGAN as well as our proposed PRDIM.
> > > > > > > >
> > > > > > > > We then trained a simple CNN classifier with an identical architecture, initialization, and training configuration across all settings, using the imputed training data only.  Finally, we evaluated all classifiers on the clean FMNIST test set and report  total classification accuracy below.
> > > > > > > >
> > > > > > > > A model whose accuracy approaches that of a classifier trained on clean data indicates that its imputed samples better preserve the semantic attributes necessary for classification. In this sense, PRDIM achieves the highest downstream performance, followed closely by MCFlow, which also employs an EM-style refinement. These results support the reviewer’s intuition that PRDIM can be effectively used in real-world applications.
> > > > > > > >
> > > > > > > > | Method (Clean test data accuracy %) | **Total** |
> > > > > > > > |-----|-------|
> > > > > > > > | **Clean Data** | 92.59 |
> > > > > > > > | **PRDIM** | **91.14** |
> > > > > > > > | **MCFlow** | 90.49 |
> > > > > > > > | **CSDI** | 87.41 |
> > > > > > > > | **misGAN** | 84.34 |

---

> > > > > > > > > ### Comment · Reviewer_pWjT · 2025-11-23
> > > > > > > > >
> > > > > > > > > I greatly appreciate the authors’ thorough response. Although some descriptions in the paper may unintentionally cause confusion for readers in the data imputation community, the rebuttal has clearly addressed most of my concerns. I will be increasing my score. Good luck!

---

> > > > > > > > > > ### Author Response · Authors · 2025-11-23
> > > > > > > > > >
> > > > > > > > > > We would like to once again express our appreciation to Reviewer **pWjT** for the thoughtful and constructive feedback, particularly for helping us clarify the core motivation of our work. We will revise the manuscript to ensure a clearer and more coherent presentation based on the reviewer’s suggestions, and we are going to highlight these updates in general response during the rebuttal stage.

---

> ### Author Response · Authors · 2025-11-21
>
> > **W2**. As the proposed method belongs to the class of diffusion-based models, it lacks evaluation metrics such as CRPS to assess the quality of imputation results.
>
> The Continuous Ranked Probability Score (CRPS) is a widely used metric for evaluating probabilistic time-series imputation. We appreciate the reviewer for requesting this additional experiment. In our study, all diffusion-based imputation models generate 10 samples, and we use the median value of these samples as the final imputed output. Following the same setting as in Table 2, we computed CRPS (lower is better) for all diffusion-based baselines and report the results in the table below.
>
> | **Dataset (CRPS $\downarrow$)**                | **CSDI** | **MTSCI** | **cDiffPuter** | **PRDIM** |
> |----------------------------|----------|-----------|----------------|-----------|
> | **ETT out-of-sample**      | 0.608    | 0.549     | 0.449          | **0.372** |
> | **ETT in-sample**          | 0.425    | 0.408     | 0.291          | **0.237** |
> | **STOCK out-of-sample**    | 0.482    | 0.476     | 0.2631         | **0.1593**|
> | **STOCK in-sample**        | 0.506    | 0.494     | 0.2752         | **0.1633**|
> | **PEMS-Bay out-of-sample** | 0.2065   | 0.2392    | 0.2475         | **0.2085**|
> | **PEMS-Bay in-sample**     | **0.1922**   | 0.2255    | 0.2342         | 0.1945|
>
>
> > **Q1**. Could the authors explain why missing patterns in real-world data are considered to be more consistent with the MNAR assumption? Is it possible to conduct experiments on datasets with naturally occurring missing values to demonstrate that the simulated MNAR patterns resemble natural ones? Alternatively, could the naturally missing patterns be used to artificially simulate missing values for evaluation?
>
>
> First, the MNAR assumption represents the most general class of missing mechanisms, encompassing both MAR and MCAR as special cases which are independent with variables $X^{\text{mis}}$ or both $X^{\text{obs}}$ and $X^{\text{mis}}$ [5]. In this sense, we view MNAR as a broad and flexible formulation capable of expressing a wide range of real-world missing behaviors.
>
> For the consecutive question, which is similar with Reviewer **vEpr**’s question, we conducted additional experiments to demonstrate the robustness of PRDIM under various forms of missing distributions. We kindly invite the reviewer to refer to these results and verify that PRDIM performs reliably across a wide range of MNAR missing scenarios which resemble real-world scenarios.
>
> Finally, as we noted in our responses to W1 and Q3, we currently do not have a principled theory that forces artificially simulated missing patterns to match naturally occurring ones. Developing such a framework would indeed be highly valuable for building more broadly applicable imputation models, and we consider this an important direction for future work.
>
> [5] Little, Roderick JA, and Donald B. Rubin. "Statistical analysis with missing data." _New York: Wiley_ (1987).
>
> > **Q2**. This paper focuses on the MNAR scenario and notes in Table 5 that the advantage of PRDIM diminishes under the MCAR setting. Could the authors further explore the model’s performance under the MAR scenario? In addition, it would be helpful to clarify in which types of missingness patterns or data distributions the proposed pattern recognizer is applicable, and under what circumstances it may not be effective. Please further discuss the limitations of PRDIM in such contexts.
>
> Thank you for raising this question regarding the MAR scenario. We agree that evaluating PRDIM beyond the MAR setting is important for demonstrating its robustness across general missing patterns.
> To this end, we conducted additional experiments using the PyGrinder repository, a public toolkit for generating missing in time-series datasets.
> Following its MAR configuration, we introduced 25% missing ratio to the ETT, STOCK, and PEMS-Bay datasets, and evaluated both out-of-sample and in-sample MAE performance.
>
> The results below show that PRDIM continues to perform consistently well under MAR, confirming that the pattern recognizer still provides useful guidance even when the missing mechanism no longer depends on unobserved values.  We are pleased to observe that PRDIM generalizes effectively to MAR, MCAR, and MNAR scenarios.
>
> | **MAR 25%** (MAE)| **ETT Out-of-sample** | **ETT In-sample** | **STOCK Out-of-sample** | **STOCK In-sample** | **PEMS-Bay Out-of-sample** | **PEMS-Bay In-sample** |
> |---------|--------|---------|-------|-----|-------|-----|
> | CSDI  | 0.1895  | 0.2428  | 0.1472      | 0.0477               | 0.2158                      | 0.2034                   |
> | cDiffPuter  | 0.1785  | 0.1853  | **0.1469**   | **0.0268**               | 0.2248                      | 0.2012   |
> | **PRDIM** | **0.1776** | **0.1699** | 0.1523  | 0.0315           | **0.2120**                  | **0.1971** |
>
> [6] https://github.com/WenjieDu/PyGrinder

---

### Official Review · Reviewer_Yosc · 2025-10-30

**Soundness:** 4
**Presentation:** 4
**Contribution:** 4
**Rating:** 6
**Confidence:** 4

**Summary:**

This paper investigates the application of diffusion model-based imputation methods under the missing not at random (MNAR) scenario. To address the challenges posed by MNAR, the authors propose the Pattern Recognized Diffusion Imputation Model (PRDIM), which incorporates a pattern recognizer to explicitly model the missingness pattern and mitigate the bias associated with MNAR. The authors further demonstrate that guidance from this pattern recognizer provides additional, approximate information for imputing missing values. Extensive experiments are conducted to validate the effectiveness of the proposed approach.

**Strengths:**

1. The topic of missing data imputation under the MNAR setting, combined with the use of diffusion models, is both timely and of significant interest to the ICLR community.
2. Theoretical derivations presented in the paper are rigorous and well-formulated.
3. The experimental results are comprehensive and adequately support the theoretical claims.

**Weaknesses:**

1. The notation should be revised for consistency and clarity. For example, $\mathcal{L}\_{diff}$ should be written as $\mathcal{L}\_{\text{diff}}$.
2. The conditions under which the assumptions in Proposition 3.1 hold should be described in greater detail. In particular, could the authors analyze the condition $q(X_{1:T}\vert X_0,M) = q(X_{1:T}\vert X_0)$ throughout the noisy process of the diffusion models?
3. According to reference [1], the diffusion process promotes diversity, which may not be ideal for imputation tasks. How does the proposed approach address this issue? Furthermore, reference [2] utilizes the median value for imputation; how does the proposed method estimate the imputed values?
4. Why did the authors choose DDPM-based diffusion models for their approach? What would be the implications of using VP-SDE-based diffusion processes, as discussed in reference [3]? Additionally, while the authors cite Theorem 1 from reference [3], it is proven in the context of the VP-SDE framework. Is the proposed method compatible with the DDPM-based diffusion models used in this work?
5. Since standard deviations for each model are reported, it would be beneficial to conduct paired-sample $t$-tests to more robustly support the theoretical claims.
6. For MNAR scenarios, reference [4] introduces MIRACLE, which post-adjusts imputed values. Could the authors compare their approach with MIRACLE?
7. The comparisons primarily focus on RMSE MAE, and MRE. Could the authors also consider metrics that evaluate distributional similarity, such as the Wasserstein distance?

---
References:
[1]. Self-Supervision Improves Diffusion Models for Tabular Data Imputation
[2]. CSDI: Conditional Score-based Diffusion Models for Probabilistic Time Series Imputation
[3]. Diffputer: Empowering diffusion models for missing data imputation.
[4]. MIRACLE: Causally-Aware Imputation via Learning Missing Data Mechanisms

**Questions:**

Please refer to the weaknesses.

---

> ### Author Response · Authors · 2025-11-21
>
> We would like to thank Reviewer **Yosc** for providing comments that helped enrich the experimental evaluation of PRDIM. Below, we address each of the identified weaknesses in detail.
>
> ---
>
> > **W1**. The notation should be revised for consistency and clarity. For example, $\mathcal{L}\_{diff}$ should be written as $\mathcal{L}\_{\text{diff}}$.
>
> Thank you for the suggestion. We changed the notations in a revised version as $\mathcal{L}\_{diff}\rightarrow\mathcal{L}\_{\text{diff}}$ and $\mathcal{L}\_{PR}\rightarrow\mathcal{L}\_{\text{PR}}$ for clear presentation.
>
> > **W2**. The conditions under which the assumptions in Proposition 3.1 hold should be described in greater detail. In particular, could the authors analyze the condition $q(X_{1:T}|X_0,M)=q(X_{1:T}|X_0)$ throughout the noisy process of the diffusion models?
>
> To prove Proposition 3.1, we adopt the assumption $q(X_{1:T}|X_0,M)=q(X_{1:T}|X_0)$ which states that the forward noising process is independent of the missing mask $M$. This assumption is fully aligned with the standard conditional diffusion literature, where the conditioning variable (e.g., a class label $C$) is also assumed to be independent of the forward process. For example, both ADM [1] and CFG [2] rely on the assumption $q(X_{1:T}|X_0,C)=q(X_{1:T}|X_0)$. The rationale is that the forward diffusion process is a fixed Markov chain designed solely to gradually transform any input data distribution into a known prior, typically a standard Gaussian.
>
> [1] Dhariwal, Prafulla, and Alexander Nichol. "Diffusion models beat gans on image synthesis." _Advances in neural information processing systems_ 34 (2021): 8780-8794.
>
> [2] Ho, Jonathan, and Tim Salimans. "Classifier-Free Diffusion Guidance." _NeurIPS 2021 Workshop on Deep Generative Models and Downstream Applications_.
>
> > **W3**. According to reference [1], the diffusion process promotes diversity, which may not be ideal for imputation tasks. How does the proposed approach address this issue? Furthermore, reference [2] utilizes the median value for imputation; how does the proposed method estimate the imputed values?
>
> As the reviewer noted, we follow the median-value imputation strategy introduced in CSDI. While CSDI reports the median over 100 independently generated samples, all diffusion-based imputation models reproduced in our paper use 10 generated samples when computing the median for test-time efficiency.
>
> > **W4**. Why did the authors choose DDPM-based diffusion models for their approach? What would be the implications of using VP-SDE-based diffusion processes, as discussed in reference [3]? Additionally, while the authors cite Theorem 1 from reference [3], it is proven in the context of the VP-SDE framework. Is the proposed method compatible with the DDPM-based diffusion models used in this work?
>
> Thank you for raising theoretical claim on our work. Theorem 1 in DiffPuter assumes dimensionality independence in the reverse process over an infinitesimal time interval. Consequently, the theorem applies to any diffusion SDE of the general form that $dx_t=f(x_t,t)dt+g(t)dW_t$ as established by Song et al. [4]. The DiffPuter paper and its official implementation adopt the diffusion design space of EDM [3], which corresponds to a VE-SDE formulation. Since DDPM is a discrete-time instance of the VP-SDE, and VP-SDE is a specific case within the same family of diffusion processes, Theorem 1 remains valid for DDPM as well. For this reason, and due to its favorable training stability, we chose to adopt the DDPM-style design in our framework.
>
> In short, the reason we adopt a DDPM parameterization for PRDIM is that it allows us to (i) freely rely on the Theorem 1 of DiffPuter derived from the SDE formulation, while simultaneously (ii) benefiting from the stable training and sampling behavior of DDPM implementations.
>
> Finally, to demonstrate the additional scalability of PRDIM across diverse datasets in VE-SDE, we replicated the MNAR tabular imputation experiments from the official DiffPuter codebase[7] under the same default settings. We incorporated our lightweight pattern recognizer (a simple MLP) and evaluated PRDIM under identical conditions. The table below (Table 11 in revised paper) reports the mean $\pm$ std of In-sample MAE over 5 runs.
>
> | Dataset (MAE)  | DiffPuter (VE-SDE) | PRDIM (VE-SDE)|
> |------|-------|------|
> | adult    | 0.497 ± 0.013 | **0.474 ± 0.012** |
> | bean     | 0.240 ± 0.069 | **0.199 ± 0.058** |
> | default  | 0.374 ± 0.092  | **0.336 ± 0.074** |
> | gesture  | **0.391 ± 0.023** | 0.394 ± 0.029     |
> | magic    | 0.539 ± 0.088     | **0.490 ± 0.083** |
>
> [3] Karras, Tero, et al. "Elucidating the design space of diffusion-based generative models." _Advances in neural information processing systems_ 35 (2022): 26565-26577.
>
> [4] Song, Yang, et al. "Score-Based Generative Modeling through Stochastic Differential Equations." _International Conference on Learning Representations_.
>
> [7] https://github.com/hengruizhang98/DiffPuter

---

> > ### Author Response · Authors · 2025-11-21
> >
> > > **W5**. Since standard deviations for each model are reported, it would be beneficial to conduct paired-sample t-tests to more robustly support the theoretical claims.
> >
> > Thank you for the valuable suggestion regarding statistical robustness.  To strengthen the empirical support for our theoretical claims, we conducted paired-sample t-tests comparing PRDIM against other generative imputation models on MAE performance.
> > The table below reports the corresponding t-statistics for both out-of-sample and in-sample settings. Across all datasets, all p-values were below 0.05, indicating that the performance improvements of PRDIM are statistically significant.
> >
> > | **Dataset** (t-statistics) | **CSDI-PRDIM (Out)** | **CSDI-PRDIM (In)** | **MTSCI-PRDIM (Out)** | **MTSCI-PRDIM (In)** | **cDiffPuter-PRDIM (Out)** | **cDiffPuter-PRDIM (In)** |
> > |-------------|------------------------:|-----------------------:|--------------------------:|-------------------------:|------------------------------:|-----------------------------:|
> > | ETT         | 1805.326               | 672.818               | 1116.906                | 2895.170                | 613.489                      | 477.692                     |
> > | STOCK       | 1792.928               | 7465.676              | 925.312                 | 1644.460                | 583.022                      | 4671.192                    |
> > | PEMS-Bay    | 111.131                | 164.932               | 284.000                 | 657.997                 | 298.500                      | 368.286                     |
> >
> >
> > > **W6**. For MNAR scenarios, reference [4] introduces MIRACLE, which post-adjusts imputed values. Could the authors compare their approach with MIRACLE?
> >
> > Thank you for introducing us to MIRACLE, an interesting and relevant approach for MNAR imputation. To the best of our understanding, MIRACLE suggests a post refinement mechanism that follows an EM-like iterative procedure. The method employs some regularization terms, including a missing model, to control the refinement dynamics. Since the refinement step is separated from the pre-imputation phase, its overall training structure is conceptually similar to PRDIM.
> > A key distinction, however, is that PRDIM adopts a likelihood-maximization perspective through the EM algorithm, and thus does not require the additional regularization hyperparameters used in MIRACLE.
> >
> > Using the official MIRACLE codebase [8], we explored some hyperparameter configurations for the ETT and STOCK datasets for stable convergence. Under (reg_lambda, reg_beta, reg_m, n_hidden)=(0.1, 1, 1, 64), the below table reports the out-of-sample and in-sample MAE.
> >
> > Interestingly, we observed that contrary to the stable convergence of the training loss the MIRACLE refinement initialized by CSDI yielded worse imputations than the original CSDI pre-imputation, which was an unexpected behavior.
> >
> > | **Method**  (MAE)                   | **ETT (Out)** | **ETT (In)** | **STOCK (Out)** | **STOCK (In)** |
> > |----------|-----:|----------:|------------:|--------:|
> > | Mean                           | 2.034          | 1.486         | 1.949            | 2.039            |
> > | MIRACLE (mean-preimputed)      | 1.596          | 0.951         | 1.919            | 1.570            |
> > | CSDI                           | 1.071          | 0.522         | 0.641            | 0.710            |
> > | MIRACLE (CSDI-preimputed)      | 1.233          | 0.613         | 0.920            | 0.763            |
> > | **PRDIM (Ours)**               | **0.663**      | **0.303**     | **0.254**        | **0.275**        |
> >
> >
> > Unfortunately, the MIRACLE implementation currently encounters a runtime failure on the PEMS-Bay dataset.  We are actively investigating this issue and will upload the results as soon as the problem is resolved.
> >
> > [8] https://github.com/vanderschaarlab/MIRACLE

---

> > > ### Author Response · Authors · 2025-11-21
> > >
> > > > **W7**. The comparisons primarily focus on RMSE MAE, and MRE. Could the authors also consider metrics that evaluate distributional similarity, such as the Wasserstein distance?
> > >
> > > Thank you for the helpful suggestion.  In addition to RMSE, MAE, and MRE, we evaluated the distributional similarity between the imputed samples and the ground-truth samples using the Wasserstein distance.
> > >
> > > We compute the Wasserstein distance between the empirical distribution of the ground-truth samples and that of the imputed samples corresponding to the results in Table 2. Each sample is treated as a single high-dimensional vector, and due to the large dimensionality, we estimate the Wasserstein distance using the Sliced Wasserstein distance. For reproducibility, we fix the projection seed to ensure a fair and consistent evaluation.
> > >
> > > There are two widely used approaches for efficiently estimating Wasserstein distance in high-dimensional settings: the **Sliced Wasserstein distance** [5] and the **Sinkhorn-regularized Wasserstein distance** [6]. Although we also experimented with the Sinkhorn-regularized formulation, we found that the relative ordering of diffusion-based imputation models varied sensitively with the choice of the regularization coefficient. This made it difficult to extract clear and stable conclusions from the results. For this reason, we opted to report the Sliced Wasserstein distance, which provides more intuitive and consistent behavior across different models.
> > >
> > > We appreciate the reviewer for suggesting this additional evaluation metric.
> > >
> > >
> > > | **Model** (Sliced WD)    | **ETT (In)** | **ETT (Out)** | **STOCK (In)** | **STOCK (Out)** | **PEMS-Bay (In)** | **PEMS-Bay (Out)** |
> > > |---------------|---------------:|----------------:|-----------------:|------------------:|---------------------:|-----------------------:|
> > > | CSDI          | 0.1019        | 0.4414         | 0.1502          | 0.1279           | 0.0139              | 0.0148                |
> > > | MTSCI         | 0.1046        | 0.4184         | 0.1707          | 0.1466           | 0.0207              | 0.0215                |
> > > | cDiffPuter    | 0.0727        | 0.4345         | 0.0946          | 0.0805           | 0.0169              | 0.0180                |
> > > | **PRDIM**     | **0.0579**    | **0.2826**     | **0.0556**      | **0.0485**       | **0.0130**          | **0.0141**            |
> > >
> > > [5] Bonnotte, Nicolas. _Unidimensional and evolution methods for optimal transportation_. Diss. Université Paris Sud-Paris XI; Scuola normale superiore (Pise, Italie), 2013.
> > >
> > > [6] Cuturi, Marco. "Sinkhorn distances: Lightspeed computation of optimal transport." _Advances in neural information processing systems_ 26 (2013).

---

> > > > ### Comment · Reviewer_Yosc · 2025-11-24
> > > > **Thank you for answering my questions**
> > > >
> > > > Thank you for addressing my previous questions. However, I still have some concerns regarding your responses.
> > > >
> > > > 1. When the DDPM process is used as the noise process, its terminal distribution approximates a standard Gaussian. Could we potentially change the mean of this distribution to improve performance? Since the OU process (the SDE form of DDPM) can be interpreted as a mean regression process [1], the vanilla DDPM implicitly assumes that the mean is fixed at $0$. Is this assumption appropriate in general, or might incorporating a data-dependent or historical mean help improve performance?
> > > > 2. My earlier question about the trade-off between sample diversity and imputation accuracy remains unanswered [2]. Does the model employ any specific technique during inference to prevent the injected noise from degrading imputation accuracy while maintaining sufficient diversity?
> > > > 3. For time-series tasks, the primary challenge compared to multi-objective optimization lies in the strong autocorrelation across time steps. I therefore suggest using a Fourier-based Wasserstein distance [3] to better account for temporal dependencies. Additionally, since the standard Wasserstein distance can be computed exactly via linear programming, for example Kuhn-Munkres algorithm [4], why did the authors choose the sliced or entropy-regularized variants instead?
> > > >
> > > > ---
> > > > References
> > > > [1]. Score-Based Generative Modeling through Stochastic Differential Equations
> > > > [2]. Self-Supervision Improves Diffusion Models for Tabular Data Imputation
> > > > [3]. The Wasserstein-Fourier Distance for Stationary Time Series
> > > > [4]. Population Matching Discrepancy and Applications in Deep Learning

---

> > > > > ### Author Response · Authors · 2025-11-25
> > > > >
> > > > > Dear Reviewer **Yosc**, Thank you for raising this insightful question regarding the terminal distribution assumption in DDPM framework, and for asking a deeper examination of its theoretical implications for imputation accuracy and model diversity. We would like to respond to your questions in several steps.
> > > > >
> > > > > > When the DDPM process is used as the noise process, its terminal distribution approximates a standard Gaussian. Could we potentially change the mean of this distribution to improve performance? Since the OU process (the SDE form of DDPM) can be interpreted as a mean regression process [1], the vanilla DDPM implicitly assumes that the mean is fixed at  . Is this assumption appropriate in general, or might incorporating a data-dependent or historical mean help improve performance?
> > > > >
> > > > > We thought that your question fundamentally concerns the possible discrepancy in the prior loss $\mathcal{L}\_T=\mathbb{E}\_{q(X_0)}[D_{KL}(q(X_T|X_0)||p(X_T))]$ and whether using a data-dependent mean for the terminal prior may improve performance.
> > > > >
> > > > > In our diffusion design, we would like to demonstrate the practical effect of this gap turns out to be minimal. Before training, all datasets undergo standard normalization, which centers the empirical mean of $X_0$ extremely close to zero. Combined with the forward diffusion coefficients adopted from the CSDI implementation: $\beta_{min}=10^{-4}, \beta_{max}=0.5, \text{diffusion steps}=50$ with a quadratic schedule. (We can obtain $\bar{\alpha}\_T\equiv\prod_{t=1}^T(1-\beta_t)=3.354\times10^{-5})$
> > > > >
> > > > > It enables us to compute the mean $\mu_1$ and variance $\sigma_1^2$ of the terminal forward process $q(X_T|X_0)$ Assuming dimensional independence, we can calculate the closed-form KL divergence between the forward terminal distribution and the standard Gaussian prior.
> > > > >
> > > > > To further examine the empirical effect of this mismatch, we conducted an additional experiment. Instead of sampling $X_T$ from the standard prior $p(X_T)=\mathcal{N}(0,I)$, we sampled from the data-induced terminal distribution $\mathbb{E}_{q(X_0)}[q(X_T|X_0)]$ and performed inference with the same pre-trained diffusion model. The results are summarized in the table below.
> > > > >
> > > > > The results show that the imputation MAE performance remains virtually unchanged across all datasets, indicating that the prior loss $\mathcal{L}_T$ has negligible impact on the diffusion inference process in our setting.
> > > > >
> > > > > | Out-of-sample                   | ETT       | STOCK     | PEMS-Bay  |
> > > > > |-------------------------------------|-----------|-----------|-----------|
> > > > > | Mean  $\mu_1$ | $-1.532\times 10^{-3}$ | $-5.95\times 10^{-4}$ | $5.32\times 10^{-4}$  |
> > > > > | Variance   $\sigma_1^2$ | $1.001236$  | $1.001055$  | $1.000088$  |
> > > > > | $D_{KL}(\mathbb{E}_{q(X_0)}[q(X_T\|X_0)]\Vert\mathcal{N}(0,I))$               | $2.61\times 10^{-4}$  | $6.60\times 10^{-5}$   | $5.59\times 10^{-4}$  |
> > > > > | MAE ($p(X_T)$ prior) | 0.663     | 0.254     | 0.170     |
> > > > > | MAE ($\mathbb{E}_{q(X_0)}[q(X_T\vert X_0)]$ prior)| 0.663     | 0.255     | 0.172 |
> > > > >
> > > > > We agree that the prior might influence the sample diversity and the inputation precision, and we may also perform prior updates as DBAE [6]. However, the diffusion prior learning will require a big changes and too-much extended scope, so we would regard them as a future extension of PRDIM. Having said that, we provide additional analyses on the diffusion prior by examing its magnitude in the loss term; and experimenting on the data induced terminal distribution as prior.
> > > > >
> > > > > [6] Kim, Yeongmin, et al. "Diffusion Bridge AutoEncoders for Unsupervised Representation Learning." _The Thirteenth International Conference on Learning Representations_.

---

> ### Author Response · Authors · 2025-11-25
>
> > My earlier question about the trade-off between sample diversity and imputation accuracy remains unanswered [2]. Does the model employ any specific technique during inference to prevent the injected noise from degrading imputation accuracy while maintaining sufficient diversity?
>
> I apologize for not addressing this part of your earlier question. As discussed in SimpDM [2], that model regulates sampling diversity by introducing a self-supervised alignment loss $\mathcal{L}_{sa}$, which effectively constrains the stochasticity of the reverse diffusion process and prevents excessive diversity during sampling.
>
> For PRDIM, we view the relationship between generation accuracy and sampling diversity through the following theoretical considerations.
>
> - In accuracy perspective: As discussed in our response to reviewer **pWjT**, the imputation objective in PRDIM fundamentally differs from that of CSDI [1]-style frameworks. While prior diffusion-based imputation models learn $p_\theta(X_0|X_0^{obs,A},M_A)$, our method is designed to model $p_{\theta,\phi}(X_0|X_0^{obs},M_O)$ under the assumption that the original missing pattern $M_O$ follows a distribution different from the artificial missing mask $M_A$. This leads PRDIM to prioritize accurate reconstruction under a more faithful missingness mechanism, rather than inducing auxiliary diversity.
> - When comparing DiffPuter and PRDIM, the key distinction is that PRDIM conditions additionally on $M_O$. From an information-theoretic perspective, for optimal parameters $\theta^\*$ and $\phi^\*$, we can show that $\mathbb{H}(p_{\theta^\*}(X_0|X_0^{obs}))$ upper-bounds $\mathbb{H}(p_{\theta^\*, \phi^\*}(X_0|X_0^{obs},M_O))$, because the conditional mutual information between the two distributions is non-negative. This implies that incorporating $M_O$ naturally reduces unnecessary sampling variability than DiffPuter.
>
> Aside from this structural effect, PRDIM does not employ any additional technique to regulate the inherent sampling diversity of the diffusion model. Motivated by your comment, we conducted an extra experiment in which CSDI, cDiffPuter, and PRDIM are each evaluated under two distinct prior samples $X_T^1$ and $X_T^2$. For each method, we measured both (i) the MAE under the two priors and (ii) the Mean Absolute Difference (Why we call it Difference, not an Error, is that it is not comparison with ground-truth value.)  between the corresponding reconstructions $X_0|X_T^1$ and $X_0|X_T^2$. The results are summarized in the table below. (**Bold** is the best and _italic_ is the second best)
>
> While PRDIM indeed exhibits lower sampling diversity than cDiffPuter, we also observed that CSDI can produce even lower diversity depending on the dataset. This suggests that methods such as the self-supervised alignment loss in [2] or, specifically for time-series data, the latent contrastive loss from MTSCI [4] could be incorporated orthogonally to improve diversity control in PRDIM. We appreciate your suggestion, as it highlights a promising direction for future work.
>
> | out-of-sample MAE| ETT | STOCK| PEMS-Bay|
> |----|----|-----|------|
> | CSDI (seed 0, GT)          | 1.0627    | 0.6407    | 0.1849    |
> | CSDI (seed 1, GT)          | 1.0699    | 0.6400    | 0.1841    |
> | DiffPuter (seed 0, GT)     | 0.7872    | 0.4035    | 0.1875    |
> | DiffPuter (seed 1, GT)     | 0.7981    | 0.4029    | 0.1867    |
> | PRDIM (seed 0, GT)         | 0.6698    | 0.2531    | 0.1831    |
> | PRDIM (seed 1, GT)         | 0.6688    | 0.2523    | 0.1819    |
>
>
> | Mean Absolute Difference| ETT | STOCK| PEMS-Bay|
> |----|----|-----|------|
> | CSDI (seed 0, seed1)       | 0.1071   | **0.0035**| **0.0452**|
> | DiffPuter (seed 0, seed1)  | _0.0877_      | 0.0170    | 0.0984    |
> | PRDIM (seed 0, seed1)      | **0.0827**    | _0.0132_    | _0.0705_    |
>
> [1] Tashiro, Yusuke, et al. "Csdi: Conditional score-based diffusion models for probabilistic time series imputation." _Advances in neural information processing systems_ 34 (2021): 24804-24816.
>
> [2] Liu, Yixin, et al. "Self-supervision improves diffusion models for tabular data imputation." _Proceedings of the 33rd ACM International Conference on Information and Knowledge Management_. 2024.

---

> > ### Author Response · Authors · 2025-11-25
> >
> > > For time-series tasks, the primary challenge compared to multi-objective optimization lies in the strong autocorrelation across time steps. I therefore suggest using a Fourier-based Wasserstein distance [3] to better account for temporal dependencies. Additionally, since the standard Wasserstein distance can be computed exactly via linear programming, for example Kuhn-Munkres algorithm [4], why did the authors choose the sliced or entropy-regularized variants instead?
> >
> > Thank you for your thoughtful suggestion regarding the use of Fourier-based Wasserstein distance. We address two points of your question in detail below.
> >
> > Our initial approach to use the Sliced Wasserstein and Sinkhorn-regularized Wasserstein distances was motivated primarily by computational considerations. For example, in the PEMS-Bay dataset, the training set has shape (5788,12,325), which becomes (5788,3900) after flattening each sample. Computing the exact 2-Wasserstein distance in this space requires evaluating the squared $\mathcal{l}_2$​ distance in 3900 dimensions between all 5788×5788 sample pairs, followed by solving an optimal transport problem over a 5788×5788 cost matrix.
> >
> > After reviewing your suggestion with Kuhn–Munkres algorithm and re-examining our earlier implementation, we realized that the previous code of calculating exact 2-Wasserstein distance did not adequately constructed. We therefore recomputed the exact 2-Wasserstein distance ($W_2$) using a corrected formulation and report the updated results in the table below.  The experimental results show that the $W_2$ discrepancy between diffusion-based imputation models is smallest for our method PRDIM.
> >
> > | Method     | ETT (In) | ETT (Out) | STOCK (In) | STOCK (Out) | PEMS-Bay (In) | PEMS-Bay (Out)|
> > |---|----|-----|----|----|-----|---|
> > | CSDI | 4.2535 | 11.7118 | 5.3450 | 4.8446 | 6.9161 | 7.5871 |
> > | MTSCI| 4.3723| 11.4063| 5.6456| 5.1334| 7.4099| 7.8256|
> > | cDiffPuter|3.6684 |10.3398| 4.1093| 3.8013| 7.5825|7.8158|
> > | PRDIM | **3.2449** | **9.0755** | **3.3878** | **3.1422**| **6.6534**|**7.5517**|
> >
> > Regarding the Fourier-based Wasserstein (WF) distance, our understanding is that the WF distance evaluates the Wasserstein discrepancy between the normalized power spectral densities (NPSDs) of two time-series distributions. As shown in [3], the WF metric provides an interpretable measure for temporal misalignment while benefiting from the stability and geometric grounding of optimal transport in the spectral domain.
> >
> > Following your recommendation, we extended the publicly available univariate WF implementation [5] to the multivariate setting and computed the WF distances for all diffusion-based imputation methods. These results are also summarized in the following table below. While PRDIM achieves the enough small WF distance on most datasets across other diffusion imputation models, we also observe that CSDI on STOCK (and MTSCI on PEMS-Bay) attains the minimum WF distance in some cases. One possible explanation is that methods trained without the EM framework tend to produce less stochastic variability across diffusion trajectories. As discussed in our response to the following question 2, this extra sampling diversity, although beneficial for modeling MNAR mechanisms, may slightly increase the WF distance. (**Bold** is the best, _italic_ is the second best)
> >
> > | Method| ETT (In)| ETT (Out)| STOCK (In)| STOCK (Out)| PEMS-Bay (In)| PEMS-Bay (Out)|
> > |--|------|----|----|-----|----|----|
> > | CSDI | 0.0988| 0.3825| **0.0532**| **0.0480**| 0.0230| 0.0304|
> > | MTSCI | 0.1072| 0.3489| 0.0883| 0.0794|**0.0224**| **0.0290**|
> > | cDiffPuter| _0.0856_| _0.3233_| 0.0927| 0.0841| 0.0244| 0.0313|
> > | PRDIM| **0.0783**| **0.2213**| _0.0760_| _0.0694_| **0.0224**| _0.0294_|
> >
> > [3] Cazelles, Elsa, Arnaud Robert, and Felipe Tobar. "The Wasserstein-Fourier distance for stationary time series." _IEEE Transactions on Signal Processing_ 69 (2020): 709-721.
> >
> > [4] Zhou, Jianping, et al. "Mtsci: A conditional diffusion model for multivariate time series consistent imputation." _Proceedings of the 33rd ACM International Conference on Information and Knowledge Management_. 2024.
> >
> > [5] https://github.com/GAMES-UChile/Wasserstein-Fourier

---

### Official Review · Reviewer_i3qS · 2025-10-31

**Soundness:** 3
**Presentation:** 2
**Contribution:** 2
**Rating:** 6
**Confidence:** 3

**Summary:**

The paper proposes PRDIM (Pattern Recognized Diffusion Imputation Model), a diffusion-based imputation framework designed to handle the Missing Not At Random (MNAR) problem. PRDIM integrates an Expectation-Maximization (EM) framework with a pattern recognizer and provides gradient-based guidance during the diffusion denoising process. Experiments show that PRDIM
achieves the state-of-the-art performance compared to previous diffusion imputation approaches under the MNAR setting.

**Strengths:**

1. The paper focuses on the Missing Not At Random (MNAR) setting, which reflects real-world missing mechanisms but has been largely neglected in prior diffusion-based imputation studies.

2. The method jointly models the data distribution and the missing pattern within an EM-based diffusion framework, offering a coherent and theoretically grounded solution to MNAR imputation.

3. Extensive experiments across four benchmark datasets show that the proposed method achieves consistent performance improvements over strong baselines, validating the effectiveness of the approach.

**Weaknesses:**

1.  Figure 1 lacks sufficient guidance on how to interpret the plots. The authors should provide a more detailed explanation of the axes, the meaning of color codings, and the overall message being conveyed by the figure.

2. From my understanding, Phase 1 mainly serves as an initialization for the diffusion model before the EM iterations begin. An ablation study examining the necessity and impact of this pretraining phase would strengthen the paper.

3. The derivation of equation 15 from proposition 3.2 seems not clear. Specifcially,  where does the $\frac{1-\alpha_t}{\sqrt{\alpha_t}}$ come from?

4. Since both the proposed method and DiffPuter are based on the EM framework, it would be better to include a more thorough discussion on a head-to-head comparison of design choices, etc. My understanding is that DiffPuter's diffusion model directly learns the joint distribution of X_mis and X_obs, while this work's diffusion model learns the conditional distribution p(X_mis|X_obs) like CSDI. The authors should clarify the rationale and potential advantages of this design choice.

**Questions:**

1. Can the author give more explanation on what Soft and hard EM are?

2. Why modify DiffPuter into a conditional diffusion framework instead of using the original form?

---

> ### Author Response · Authors · 2025-11-21
>
> We thank Reviewer **i3qS** for providing valuable guidance on the presentation and for suggesting new ways to further utilize PRDIM. We hope that the responses below adequately address the reviewer’s questions.
>
> ---
>
> > **W1**. Figure 1 lacks sufficient guidance on how to interpret the plots. The authors should provide a more detailed explanation of the axes, the meaning of color codings, and the overall message being conveyed by the figure.
>
> Thank you for pointing out that Figure 1 did not provide sufficient guidance for interpretation.  We agree that we attempted to convey too much information within a limited figure space.  To improve clarity, we have added a more detailed description of the missing data processing pipeline in **Appendix D** (and Figure 12) of the revised version.  This supplementary section explains how complete benchmark datasets are transformed into incomplete versions, and distinguishes between (i) artificially masked entries used by several diffusion-based imputation papers, and (ii) the truly valuable original missing entries.
>
> The top graph in Figure 1 visualizes a segment of length 120 (x-axis) from the normalized ETT test set (obtained by concatenating five consecutive samples).
> - Orange regions and blue points indicate entries that are _not observed_ in the model input.
> - Blue points represent _artificially missing_ entries introduced in Phase 1 training.
> - Orange regions correspond to _original missing entries_ present in the real data. Its missing pattern distribution is differ to the artificial one.
> - Pink and red curves denote the corresponding imputation results of DiffPuter and PRDIM respectively.
>
> Numerically, this distinction aligns with Table 1 in the paper. The left block reports imputation errors on the _artificially missing_ (blue-point entries) while the right block reports errors on the _original missing_ (orange-region entries).
>
> The bottom graph evaluates whether the trained pattern recognizer which aims to converge to the optimal missing pattern probability described in Section 2.1. The black curve is the true missing probability over time. The pink curve displays the average missing ratio of the segment. The red line represents the **estimated missing probability** for each entry, obtained by injecting the imputed sample into the trained pattern recognizer. We introduce more detailed visualization figures (Figure 13-16) and present clear explanation in Appendix C.2.4.
>
> The closer the red curve is to the black curve, the better the recognizer approximates the true missing probability. In this sense, our pattern recognizer successfully captures the temporal variation of real missing, supporting the claim that it provides meaningful guidance during imputation.
>
>
> > **W2**. From my understanding, Phase 1 mainly serves as an initialization for the diffusion model before the EM iterations begin. An ablation study examining the necessity and impact of this pretraining phase would strengthen the paper.
>
> Thank you for raising this insightful question regarding the role of Phase 1 in PRDIM.  We agree that this is a core aspect of understanding how our method operates. Our choice to adopt the CSDI objective in Phase 1 was motivated by two practical advantages. (i) Leveraging a pretrained diffusion backbone provides a more stable initialization for Phase 2, which is beneficial when configuring the EM procedure. (ii) Ensuring the generative capability of the diffusion model prior to EM iterations improves convergence stability and prevents degenerate imputations.
> However, as the reviewer correctly points out, from a theoretical standpoint, Phase 1 could be replaced with any imputation method, and PRDIM could in principle be trained fully from scratch. To validate this, we conducted an ablation study using three non-diffusion based pre-imputation baselines (MEAN, BRITS, SAITS) to initialize Phase 2. To compensate for the absence of a pretrained diffusion model, we increased the number of maximization epochs within each EM iteration.
>
> | **Initialization Method** (MAE) | **ETT Out-of-sample** | **ETT In-sample** | **STOCK Out-of-sample** | **STOCK In-sample** | **PEMS-Bay Out-of-sample** | **PEMS-Bay In-sample** |
> |--------|------|--------|----|---|------|------|
> | PRDIM (MEAN) (close to w/o Phase 1)      | 0.766| 0.407| 0.326 | 0.334| 0.207| 0.165 |
> | PRDIM (BRITS)      | 0.824 | 0.430 | 0.300  | 0.332| 0.188 | 0.168 |
> | PRDIM (SAITS)      | 0.774| 0.418| 0.307 | 0.341 | 0.180| 0.166 |
> | **PRDIM** (CSDI) | **0.663** | **0.303** | **0.254**  | **0.275** | **0.170**  | **0.154** |
>
> The table below reports MAE on ETT, STOCK, and PEMS-Bay under these different Phase 1 configurations.  Across all settings, PRDIM consistently delivers the most accurate imputations. We sincerely appreciate the reviewer’s comment, which has provided valuable intuition that PRDIM can serve as an iterative refinement module on top of any imputation model, particularly under MNAR settings.

---

> ### Author Response · Authors · 2025-11-21
>
> > **W3**. The derivation of equation 15 from proposition 3.2 seems not clear. Specifcially, where does the $\frac{1-\alpha_t}{\sqrt{\alpha_t}}$ come from?
>
> Start from Equation (13) which indicates 1-step diffusion denoising,
>
> $$\tilde{X}\_{t-1}=\frac{1}{\sqrt{\alpha\_t}}(X\_t+(1-\alpha\_t)\nabla\_{X\_t}\log{p\_\theta(X\_t|X\_0^{\text{obs}})})
> $$
>
> This equation represents a reverse step of a vanilla diffusion imputation model when no guidance for the missing mask is provided. The mask-unconditional score function $\nabla\_{X\_t}\log{p\_\theta(X\_t|X\_0^{\text{obs}})}$ should be modified to conditional score function $\nabla_{X_t}\log{p_\theta(X_t|X_0^{\text{obs}},M)}$ . From Proposition 3.2, $\nabla\_{X\_t}\log{p\_\theta(X\_t|X\_0^{\text{obs}},M)}$ could be estimated with adding $-\nabla\_{X\_t}\mathcal{L}\_{PR}(M,\hat{X}\_0,D\_{\phi^*})$ to the mask-unconditional score function term.
> Therefore, the coefficient of score function in Equation (13) is naturally adjusted to Proposition 3.2.
>
> $$
> \tilde{X}\_{t-1}=\frac{1}{\sqrt{\alpha\_t}}(X\_t+(1-\alpha\_t)\{\nabla\_{X\_t}\log{p\_\theta(X\_t|X\_0^{\text{obs}})}-\nabla\_{X\_t}\mathcal{L}\_{PR}(M,\hat{X}\_0,D\_{\phi^*})\})
> $$
> where the above equation equals to (15).
>
> > **W4**. Since both the proposed method and DiffPuter are based on the EM framework, it would be better to include a more thorough discussion on a head-to-head comparison of design choices, etc. My understanding is that DiffPuter's diffusion model directly learns the joint distribution of $X\_{mis}$ and $X\_{obs}$, while this work's diffusion model learns the conditional distribution $p(X\_{mis}|X\_{obs})$ like CSDI. The authors should clarify the rationale and potential advantages of this design choice.
> > **Q2**. Why modify DiffPuter into a conditional diffusion framework instead of using the original form?
>
> Table 2 of CSDI [1] empirically demonstrates that providing the observed variables $X^{\text{obs}}$​ as conditioning inputs results in more precise and stable imputation compared to unconditional diffusion models. Although PRDIM can also be implemented under an unconditional diffusion framework, we found that unconditional training (i) requires significantly more epochs to converge, (ii) exhibits lower training stability, and (iii) yields substantially worse imputation performance. This motivated our decision to adapt DiffPuter into a conditional formulation by supplying the additional conditioning input $X^{\text{obs}}$.
>
> To validate this design decision, we conducted additional experiments under the same settings as Table 2 in the main paper.  We report the MAE evaluation results that confirm that both PRDIM and DiffPuter perform markedly worse when trained unconditionally, reinforcing the practical advantage of conditional diffusion.
>
> | Dataset (MAE) | Conditional | Conditional | Unconditional | Unconditional                      |
> |----------|-------------|----------|---------------|-----------------------|
> |       | cDiffPuter  | PRDIM    | DiffPuter     | PRDIM  |
> | ETT      | 0.782       | **0.663**    | 1.466         | 0.972                 |
> | STOCK    | 0.406       | **0.254**    | 1.175         | 0.676                 |
> | PEMS-Bay | 0.182       | **0.170**    | 0.202         | 0.178                 |
>
>
> [1] Tashiro, Yusuke, et al. "Csdi: Conditional score-based diffusion models for probabilistic time series imputation." _Advances in neural information processing systems_ 34 (2021): 24804-24816.
>
> >**Q1**. Can the author give more explanation on what Soft and hard EM are?
>
> Soft EM refers to an approach in which multiple samples are generated using Monte Carlo sampling, and the model parameters are updated based on the expected imputed value. In this case, the imputed sample corresponds to the posterior expectation
>
> $$
> \hat{X}\_0=\mathbb{E}\_{p\_\theta(X\_0|X\_0^\text{obs}, M)}\[X\_0\]
> $$
>
> In contrast, Hard EM can be viewed as producing a _single_ imputed output obtained by the posterior mode:
>
> $$
> \hat{X\}_0=\arg\max\_{X\_0} p\_\theta(X\_0|X\_0^\text{obs}, M)
> $$
>
> We adopt Hard EM for two main reasons. (i) It significantly reduces the sampling time because only one imputed sample is required at each iteration. (ii) In our experiments, Hard EM yields slightly better imputation performance than Soft EM.
> This observation is consistent with the findings of Samdani et al. [2], who report that Hard EM performs better when the initialization is strong, whereas Soft EM is preferable under uninformed initialization. Since PRDIM benefits from Phase 1, where the diffusion model is well initialized by CSDI, the Hard EM variant is naturally more effective in our setting.
>
> [2] Samdani, Rajhans, Ming-Wei Chang, and Dan Roth. "Unified expectation maximization." _Proceedings of the 2012 Conference of the North American Chapter of the Association for Computational Linguistics: Human Language Technologies_. 2012.

---

### Official Review · Reviewer_vEpr · 2025-11-01

**Soundness:** 2
**Presentation:** 3
**Contribution:** 3
**Rating:** 6
**Confidence:** 2

**Summary:**

This paper focuses on addressing the Missing Not at Random (MNAR) problem, a common issue in missing data across time-series and image domains where missing occurrences depend on unobservable values. It points out that while existing diffusion models excel in out-of-sample imputation, most overlook the MNAR setting and rely on restrictive assumptions about the missing process, limiting practical applicability. To solve this, the paper proposes the Missing Pattern Recognized Diffusion Imputation Model (PRDIM), a novel framework that explicitly captures missing patterns and accurately imputes unobserved values. PRDIM leverages the Expectation-Maximization (EM) algorithm to iteratively maximize the likelihood of the joint distribution of observed values and missing masks, and incorporates a pattern recognizer to approximate the underlying missing pattern and guide inference for more plausible imputations. Experimental results show that under the MNAR setting, PRDIM outperforms previous diffusion-based imputation approaches.

**Strengths:**

The paper directly addresses the understudied challenge of Missing Not at Random (MNAR) imputation, a prevalent but overlooked setting in real-world data where missingness depends on unobserved values. By contrast, most existing diffusion-based imputation methods rely on restrictive assumptions that fail to reflect practical scenarios—this focus on MNAR fills a meaningful niche in the literature.
The paper demonstrates state-of-the-art (SOTA) performance of PRDIM against existing diffusion-based imputation methods under MNAR settings. This empirical rigor is critical: it validates that the model’s theoretical focus on MNAR translates to tangible improvements in imputation quality, making the work credible for researchers and practitioners working with incomplete data.

**Weaknesses:**

MNAR encompasses diverse subtypes, but the paper does not clarify which MNAR scenarios PRDIM excels at. For example, it does not test PRDIM on data with varying degrees of MNAR severity or different missingness triggers—limiting conclusions about its applicability beyond the specific experimental settings studied.
While the pattern recognizer is framed as a core component of PRDIM, the paper provides insufficient detail on its internal design  and how it specifically learns to model MNAR mechanisms. Without ablation studies or qualitative examples of its outputs, readers cannot fully assess whether this component is driving the model’s SOTA performance—or if it is redundant with other parts of the framework.

**Questions:**

Did you test PRDIM on these distinct MNAR subtypes, and if so, could you share results on which subtypes PRDIM performs best/worst? If not, do you have hypotheses about PRDIM’s limitations when facing MNAR mechanisms different from those in your current experiments?
Could you clarify whether the MNAR mechanisms in your test data align with common real-world scenarios?

---

> ### Author Response · Authors · 2025-11-21
>
> We thank the reviewer **vEpr** for the valuable comments. We address the issues below.
>
> ---
>
> > **W1-1**. It does not test PRDIM on data with varying degrees of MNAR severity or different missingness triggers—limiting conclusions about its applicability beyond the specific experimental settings studied.
> >  **Q1**. Did you test PRDIM on these distinct MNAR subtypes, and if so, could you share results on which subtypes PRDIM performs best/worst? If not, do you have hypotheses about PRDIM’s limitations when facing MNAR mechanisms different from those in your current experiments? Could you clarify whether the MNAR mechanisms in your test data align with common real-world scenarios?
>
> Thank you for encouraging us to further investigate the robustness of PRDIM across different MNAR subtypes. In the main text, our experiments focus on the self-censoring MNAR mechanism introduced in [1], and the details regarding missing ratios and missing patterns are provided in Appendix C (**Missing Mechanisms** and **Dataset Configuration**).
>
> Based on the reviewer’s suggestion, we additionally examined several real-world motivated MNAR subtypes and conducted further experiments on the ETT dataset. These subtypes were selected because they (i) appear in real-world domains and (ii) can be implemented without domain-specific knowledge, enabling reproducible evaluation:
>
> -   **Self-censoring [1]**: The true observed value directly determines whether the entry becomes missing.
>
> -   **Latent-trait MNAR [2]**: Missing depends on an unobserved latent effect or individual-specific factor.
>
> -   **Censoring / Truncation MNAR [3]**: Values outside a certain interval are unobserved or collapsed, simulating realistic censoring processes.
>
> To demonstrate the variation in MNAR severity across these subtypes, we also provide the train/val/test missing ratios produced during data processing. Across all MNAR subtypes, PRDIM consistently outperforms prior diffusion-based imputation methods.  However, we observe that the performance gap between PRDIM and DiffPuter is smallest under the latent-trait mechanism, suggesting that this subtype may be the most challenging among the ones tested. This is intuitive, as missing driven by latent attributes is more difficult to approximate only with simple CNN (or MLP) structure.
>
> | **MNAR Subtype** (missing ratio %)  | Train | Valid | Test |
> |--------------------|-------------------------|------------------------|-------------------------|
> | Self-censoring     | 19.75                  | 13.52                 | 37.16                  |
> | Latent-trait       | 30.05                  | 30.09                 | 29.95                  |
> | Censoring          | 27.76                  | 27.75                 | 28.05                  |
>
> | Method (MAE)   | **Self-censoring (In-sample)** | **Self-censoring (Out-of-sample)** | **Latent traits (In-sample)** | **Latent traits (Out-of-sample)** | **Truncation (In-sample)** | **Truncation (Out-of-sample)** |
> |--------------|--------------------------------|-------------------------------------|--------------------------------|-----------------------------------|------------------------------|--------------------------------|
> | CSDI         | 0.3073                         | 0.5303                              | 0.2338                         | 0.2621                            | 0.3682                       | 0.5780                         |
> | MTSCI        | 0.3296                         | 0.4848                              | **0.1629**                         | 0.1968                            | 0.4368                       | 0.5353                         |
> | cDiffPuter   | 0.3845                         | **0.3819**                              | 0.1973                         | 0.2074                            | 0.5471                       | 0.5709                     |
> | **PRDIM**    | **0.2738**                     | **0.3819**                           | 0.1711                     | **0.1930**                        | **0.3570**                   | **0.4833**                     |
>
> [1] Miao, Wang, et al. "Identification, doubly robust estimation, and semiparametric efficiency theory of nonignorable missing data with a shadow variable." _arXiv preprint arXiv:1509.02556_ (2015).
>
> [2] Cursio, John F., Robin J. Mermelstein, and Donald Hedeker. "Latent trait shared‐parameter mixed models for missing ecological momentary assessment data." _Statistics in Medicine_ 38.4 (2019): 660-673.
>
> [3] Shah, Jasmit S., et al. "Distribution based nearest neighbor imputation for truncated high dimensional data with applications to pre-clinical and clinical metabolomics studies." _BMC bioinformatics_ 18.1 (2017): 114.

---

> > ### Author Response · Authors · 2025-11-21
> >
> > > **W1-2**. While the pattern recognizer is framed as a core component of PRDIM, the paper provides insufficient detail on its internal design and how it specifically learns to model MNAR mechanisms.
> >
> > Thank you for pointing out the need for a clearer description of the internal architecture of the pattern recognizer. The pattern recognizer is intentionally designed to be lightweight. As shown in below pseudo code, it is implemented as a simple MLP module with a single residual block and a hidden channel size of 64. Overall, its parameter count corresponds to only 10.61% of the diffusion backbone, indicating that the recognizer is not a large or complex addition to the model. We provide the pseudo code of pattern recognizer architecture below. Furthermore, the pattern recognizer is optimized separately from the diffusion model during the M-step based on the imputed samples. This procedure is described in Algorithm 2 of the main text, and we also provide the corresponding pseudo-code below for clarity.
> >
> > As shown in the bottom of Figure 1, the recognizer’s estimated missing ratios closely match the true missing ratios over time on a randomly sampled ETT test segment. This provides experimental evidence that the learned $D_\phi$​ behaves consistently with the theoretical optimal recognizer $D_\phi^*$​ described in Proposition 3.2.
> >
> > In addition, we have added a case study section in Appendix C.2.4 of the revised paper to verify whether the trained pattern recognizer $D_\phi$​ ​ accurately estimates the simulated MNAR missing distribution. In Figures 13–16, the **black curve** represents the true missing ratio, while the **red curve** denotes the estimated missing ratio produced by the pattern recognizer when applied to the imputed samples of PRDIM. The closer these two curves are, the better the pattern recognizer has been trained.
> >
> > 	class PatternRecognizer:
> > 	    def __init__(self, channels=64):
> > 	        self.in_proj  = Conv1d(in_dim=1, out_dim=channels, kernel_size=1)
> > 	        self.res = ResidualBlock(channels)
> > 	        self.out_proj = Conv1d(in_dim=channels, out_dim=1, kernel_size=1)
> >
> > 	    def forward(self, x):
> > 	        h = relu(self.in_proj(x))
> > 	        h, s = self.res(h)
> > 	        return sigmoid(self.out_proj(h))
> >
> > 	class ResidualBlock:
> > 	    def __init__(self, channels):
> > 	        self.time_mlp = MLP(channels → channels/8 → channels)
> > 	        self.feat_mlp = MLP(channels → channels/8 → channels)
> > 	        self.mid_proj = Conv1d(channels, 2*channels, 1)
> > 	        self.out_proj = Conv1d(channels, 2*channels, 1)
> >
> > 	    def forward(self, x):
> > 	        h = self.time_mlp(x)
> > 	        h = self.feat_mlp(h)
> > 	        gate, filt = chunk(self.mid_proj(h), 2)
> > 	        y = sigmoid(gate) * tanh(filt)
> > 	        res, skip = chunk(self.out_proj(y), 2)
> > 	        return (x + res)/sqrt(2), skip
> >
> > ---
> >
> >     initialize model and pattern_recognizer
> > 	for each EM iteration (em_iter = 1 ... E):
> > 	    ### E-step ###
> > 	    set model, pattern_recognizer to eval mode
> > 	    pre_batched = empty list
> >
> > 	    for each batch in train_loader:
> > 	        # reverse process
> > 	        samples = model.evaluate(batch, number_of_sample=1, pattern_recognizer)
> >
> > 	        # construct imputed batch
> > 	        imputed_batch = copy(batch)
> > 	        imputed_batch["observed_data"] = samples
> >
> > 	        pre_batched.append(imputed_batch)
> >
> > 	        if itr_per_epoch limit reached:
> > 	            break
> >
> > 	    shuffle(pre_batched)
> > 	    imputed_train_loader = DataLoader(pre_batched)
> >
> > 	    ### M-step ###
> > 	    reinitialize Adam optimizer + MultiStep LR scheduler
> > 	    set model, pattern_recognizer to train mode
> >
> > 	    for mstep_epoch = 1 ... M:
> > 	        for imputed_batch in imputed_train_loader:
> > 	            move imputed_batch to device
> >
> > 	            optimizer.zero_grad()
> > 	            loss_model = model(imputed_batch)
> > 	            loss_pr = pattern_recognizer(imputed_batch)
> > 	            loss = loss_model + loss_pr
> > 	            loss.backward()
> > 	            optimizer.step()
> >
> > 	        scheduler.step()

---

### Official Review · Reviewer_7n6G · 2025-11-01

**Soundness:** 3
**Presentation:** 2
**Contribution:** 2
**Rating:** 4
**Confidence:** 4

**Summary:**

This paper introduces PRDIM (Missing Pattern Recognized Diffusion Imputation Model), a diffusion-based framework for imputing missing values under the Missing Not at Random (MNAR) setting. Unlike most existing imputation models that assume missingness is random or only dependent on observed values, PRDIM explicitly models the missing pattern using a pattern recognizer within an Expectation-Maximization (EM) framework.

**Strengths:**

1. The paper targets the Missing Not at Random (MNAR) scenario, which is more realistic yet underexplored in diffusion-based imputation research, addressing a clear research gap.

2. The proposed PRDIM model a pattern recognizer within an EM-based diffusion framework, providing a principled way to model and leverage the missing pattern during inference.

**Weaknesses:**

1. One of the strong motivations of the paper is that missing patterns, such as Missing Not at Random (MNAR), are more common in real-world scenarios. However, the data used in the subsequent imputation tasks are artificially split from complete datasets such as ETT, PEMS, and Stock, and do not actually use real-world missing data.

2. The proposed model is intended to be a time-series imputation model, but the validation is done on static image data from Fashion MNIST. It is recommended to use a time-series dataset with temporal variations, such as Moving-MNIST, for validation.

3. PRDIM is an imputation model for time-series data. For time-series datasets, there is a significant issue of temporal shift, i.e., the distribution of data in the training set differs greatly from that in the test set. However, PRDIM does not provide a specific solution to address this issue.

4. The terms "out-of-sample imputation tasks" and "in-sample imputation tasks" are not clearly defined in the main text.

5. PRDIM's performance heavily relies on the accuracy of the pattern recognizer. If the missing patterns are estimated inaccurately, it could affect the final imputation accuracy. This could be especially problematic for time-series datasets with significant distribution shifts, as it may lead to overfitting on the training data.

6. Although the pattern recognizer provides an estimate of the missing pattern, there is a lack of visual interpretability and analysis of how the generated imputations are guided by the pattern. This was one of the motivations of the paper, but it was not well validated in the experimental section.

7. The diffusion backbone and missing pattern recognizer require joint optimization, which may lead to high training costs. The paper lacks an analysis of the model's efficiency.

**Questions:**

1. Is the ETT dataset using ETTH1, ETTH2, or ETTM1, ETTM2?

2. In addition, there are many more complete and standardized forecasting datasets, such as Traffic or Electricity. Why use such a small ETT dataset?

---

> ### Author Response · Authors · 2025-11-21
>
> We sincerely appreciate Reviewer **7n6G** for the valuable comments and constructive feedback on our proposed method. We hope that the following clarifications and additional results help address the reviewer’s concerns and provide a deeper understanding of our experimental design and motivations.
>
> ---
>
> > **W1**. One of the strong motivations of the paper is that missing patterns, such as Missing Not at Random (MNAR), are more common in real-world scenarios. However, the data used in the subsequent imputation tasks are artificially split from complete datasets such as ETT, PEMS, and Stock, and do not actually use real-world missing data.
>
> We agree with the reviewer’s observation that the datasets used in our experiments are originally complete without any missing values. However, this experimental design was made to ensure fair and controlled comparisons of imputation performances because true values are needed to compute the accuracy performance between imputation models. We also add description about this question in **Appendix D** of the revised paper.
> The rationale behind this choice is as follows:
>
> - Real-world incomplete time-series benchmarks (e.g., PhysioNet, AirQuality) indeed contain naturally missing data.  However, the ground-truth values for these missing entries are **not accessible for validation**.
> Several prior works (CSDI [1], SSSD [2], Diffusion-TS [3]) include this limitation since they introduce _additional artificially missing entries_ to perform evaluation.  This approach, however, breaks the MNAR  missing pattern of the original datasets.  Empirically, this is confirmed by our finding that imputation performance on _artificially missing entries_ and _original missing entries_ differs significantly (see Table 1).
>
> - To quantitatively evaluate imputation accuracy, it is essential to have ground-truth values corresponding to the missing in the input.  Therefore, complete dataset must be required, and the missing mechanism generated for experiments is _unknown (not accessible)_ to all models, including PRDIM and the baselines, ensuring a fair comparison.
>
> Although our main imputation target scope differs from artificial missing entries, we further conducted experiments on the PhysioNet dataset, which contains approximately 80% real missing ratios. Therefore we add the qualitative imputation results on PhysioNet dataset. Although we could not evaluate with the ground-truth value which cannot accessible, Figure 20 on **Appendix C.2.5** (of the revised paper) shows that how does PRDIM works on real-world missing dataset.
>
>
> [1] Tashiro, Yusuke, et al. "Csdi: Conditional score-based diffusion models for probabilistic time series imputation." _Advances in neural information processing systems_ 34 (2021): 24804-24816.
>
> [2] Alcaraz, Juan Lopez, and Nils Strodthoff. "Diffusion-based Time Series Imputation and Forecasting with Structured State Space Models." _Transactions on Machine Learning Research_.
>
> [3] Yuan, Xinyu, and Yan Qiao. "Diffusion-TS: Interpretable Diffusion for General Time Series Generation." _The Twelfth International Conference on Learning Representations_.

---

> ### Author Response · Authors · 2025-11-21
>
> > **Q1**. Is the ETT dataset using ETTH1, ETTH2, or ETTM1, ETTM2?
>
> We conducted all our experiments using the **ETTm1** dataset. Our choice was based on the fact that ETTm1 is the most frequently used benchmark in prior time-series imputation studies [4]. Before introducing PRDIM, we reproduced the performance of existing imputation models on ETTm1 to ensure that our implementation aligns with previously reported and widely accepted results, for example, those reported in SAITS and MTSCI.
>
> [4] Zhou, Haoyi, et al. "Informer: Beyond efficient transformer for long sequence time-series forecasting." _Proceedings of the AAAI conference on artificial intelligence_. Vol. 35. No. 12. 2021.
>
> > **Q2**. In addition, there are many more complete and standardized forecasting datasets, such as Traffic or Electricity. Why use such a small ETT dataset?
>
> While the ETTm1 dataset indeed has a relatively small dimensionality (time length 24 with 7 features) compared to some larger forecasting benchmarks, our experiments are not limited to small-scale time-series data. For example, PEMS-Bay has a time length of 12 with 325 features, Fashion-MNIST consists of 28×28 image inputs, and the additional CelebA-HQ experiments involve inputs of size 64×64×3. These datasets span a wide range of dimensionalities, demonstrating that our experimental evaluation is not constrained to small datasets such as ETT.
>
> In short, our dataset selection was guided by two criteria, (i) the datasets must be complete, allowing us to accurately simulate missing mechanisms and reliably evaluate against ground-truth values; and (ii) we avoided datasets such as small-scale UCI tabular benchmarks, whose low input dimensionality does not adequately showcase the advantages of diffusion-based imputation models.
>
> | **Dataset** | **ETT** | **STOCK** | **PEMS-Bay** | **PhysioNet**| **FMNIST** | **CelebA-HQ (downsizing)** |
> |-------------|---------|---------|-----------|--------------|------------|-------------------------|
> | **Data size** | 24 × 7 | 24 × 6 | 12 × 325 |48 x 35 | 28 × 28 | 64 × 64 × 3 |
> | **# of Samples (train / val / test)** | 3861 / 983 / 959 | 2418 / 622 / 622 | 5788 / 1448 / 1448 | 8000 / 2000 / 2000 | 60000 / 5000 / 5000 | 30000 / 5000 / 5000 |
> | **Missing ratio (%) (train / val / test)** | 21.4 / 43.9 / 14.0 | 21.2 / 20.0 / 20.9 | 13.5 / 13.0 / 14.1 | 80.5 / 80.5 / 80.6 | 25.8 / 25.8 / 25.8 | 2.73 / 2.74 / 2.73 |
>
> Regarding the **Electricity** dataset, we decided not to include it because its characteristics with long temporal length (370), the relatively large feature dimension (100), and the limited number of training samples (approximately 1,000). It makes difficulty for diffusion-based imputation models to reliably capture the underlying data distribution since its underdetermined system. Instead, we used **PEMS-Bay**, which provides a traffic dataset with similar characteristics but a more suitable dataset scale and distribution for stable model training.
>
> >  **W3**. PRDIM is an imputation model for time-series data. For time-series datasets, there is a significant issue of temporal shift, i.e., the distribution of data in the training set differs greatly from that in the test set. However, PRDIM does not provide a specific solution to address this issue.
>
> We appreciate the reviewer’s suggestion to explore datasets that integrate both image and time-series attributes. However, we would like to clarify that PRDIM is not a time-series model. Instead, it is a likelihood-based framework that is applicable to any data modality as long as a score-based diffusion process can be defined. Figure 1 may lead Reviewer to misunderstand that PRDIM is limited to a time-series imputation model, but it is not. One example is the image imputation experiments that we present in Figure 3 and Figure 9-11, which is not time-series imputation problems.
>
> Therefore, temporal shift is not within the research scope of our paper. As an example, although the ETT dataset indeed exhibits temporal shift between the in-sample and out-of-sample partitions, as the reviewer pointed out, PRDIM achieves strong performance on both, which we demonstrate in Table 2. In the table, the out-of-sample partitions could incur the temporal distribution shift, and PRDIM shows 15.22% of performance gain in MAE compared to cDiffPuter on ETT dataset.
>
> We attribute this to the fact that the missing distribution $p(M|X_0)$ is consistent across the two partitions, despite differences in their marginal data distributions. Addressing temporal shift would require integrating ideas from continual learning or domain adaptation. However, the present paper focuses solely on the imputation problem. We appreciate the reviewer for providing valuable supervision for future work.

---

> > ### Author Response · Authors · 2025-11-21
> >
> > > **W2**. The proposed model is intended to be a time-series imputation model, but the validation is done on static image data from Fashion MNIST. It is recommended to use a time-series dataset with temporal variations, such as Moving-MNIST, for validation.
> >
> > Furthermore, to illustrate PRDIM’s scalability across different data modalities, we reported imputation performance on self-censoring and block-missing settings of Fashion-MNIST in Figure 3 of the main text and Appendix C.2.1. These experiments demonstrate that PRDIM generalizes effectively to non-temporal, high-dimensional data as well. To provide clearer visualization and complement the Moving-MNIST, we added qualitative imputation experiments on CelebA-HQ (width × height × RGB) in Figure 10 of Appendix C.2.2 (revised version).  To implement a clear MNAR missing mechanism, we applied 80% structured masking using eyes, nose, and mouth annotation masks.
> >
> > Because the Moving-MNIST dataset has substantially larger temporal dimension than the downsized CelebA-HQ we used, conducting the full set of experiments requires additional computation. We will design and conduct the Moving-MNIST experiments as quickly as possible during the rebuttal period, and we kindly ask for your understanding regarding this issue.
> >
> > >  **W5**. PRDIM's performance heavily relies on the accuracy of the pattern recognizer. If the missing patterns are estimated inaccurately, it could affect the final imputation accuracy. This could be especially problematic for time-series datasets with significant distribution shifts, as it may lead to overfitting on the training data.
> >
> > Thank you for raising this important question regarding the sensitivity of PRDIM to the accuracy of the pattern recognizer. In Section 4.3, we empirically showed that the recognizer converges toward a stable estimate of the missing pattern. However, we did not directly evaluate how PRDIM behaves when the pattern recognizer is intentionally weakened. We therefore conducted an additional study to examine this behavior.
> >
> > To systematically control the quality of the recognizer, we reduced its parameter size, thereby restricting the solution space and forcing it to converge to less accurate estimates. We then measured he average cross-entropy on observed entries ($M_{i,j}=1$) and on missing entries ($M_{i,j}=0$). A larger average cross-entropy indicates a poorer ability to discriminate missing patterns. For example, when the value approaches $-\ln{0.5}\simeq0.693$, the recognizer effectively collapses to random guessing and fails to capture any meaningful missing pattern.
> >
> > Using this setup, we evaluated PRDIM on the ETT and STOCK datasets while keeping all other conditions identical and varying only the parameter number of pattern recognizer. The tables below summarize the results. As expected, higher cross-entropy corresponds to degraded imputation accuracy. However, it is noteworthy that PRDIM still outperforms cDiffPuter even with significantly weakened pattern recognizers. We can demonstrate that the pattern recognizer provides meaningful guidance to the diffusion process when properly learned.
> >
> > A fully non-trained recognizer provides no gain to the score function, consistent with Table 1 of Kim et al. [5], which shows that discriminator (or guidance) collapse recovers the unguided diffusion model. Thereby explaining why the theoretical gap between PRDIM and DiffPuter naturally diminishes under MCAR settings.
> >
> > | **ETT PR Parameter Number** | **Avg. CE (Observed)** | **Avg. CE (Missing)** | **Out-of-sample MAE** | **In-sample MAE** |
> > |------------------------|-------------------------|------------------------|-------------------------|--------------------|
> > | 17376                  | **0.323**                  | **0.283**                 | **0.663**                  | **0.303**              |
> > | 4608                   | 0.339                  | 0.311                 | 0.744                  | 0.354              |
> > | 1296                   | 0.480                  | 0.456                 | 0.763                  | 0.360              |
> >
> > | **STOCK PR Parameter Count** | **Avg. CE (Observed)** | **Avg. CE (Missing)** | **Out-of-sample MAE** | **In-sample MAE** |
> > |------------------------|-------------------------|------------------------|-------------------------|--------------------|
> > | 17359                  | **0.165**                  | **0.151**                 | **0.254**                  | **0.275**              |
> > | 4599                   | 0.168                  | 0.155                 | 0.271                  | 0.301              |
> > | 1291                   | 0.169                  | 0.162                 | 0.277                  | 0.303              |
> >
> > [5] Kim, Dongjun, et al. "Refining Generative Process with Discriminator Guidance in Score-based Diffusion Models." 40th International Conference on Machine Learning, ICML 2023. ML Research Press, 2023.

---

> > > ### Author Response · Authors · 2025-11-21
> > >
> > > > **W6**. Although the pattern recognizer provides an estimate of the missing pattern, there is a lack of visual interpretability and analysis of how the generated imputations are guided by the pattern. This was one of the motivations of the paper, but it was not well validated in the experimental section.
> > >
> > > We agree with the reviewer’s consecutive observation with W5. In our framework, the pattern recognizer indeed plays a crucial role in effectively estimating MNAR missing. In **Proposition 3.2**, we theoretically demonstrate that an optimal pattern recognizer $D_\phi^*$​ can provide meaningful guidance to diffusion-based imputation.
> > >
> > > The bottom part of Figure 1 presents a plot of the _true missing ratio_ and the _estimated missing ratio_ produced by $D_\phi$​​ over the time axis for a randomly sampled segment from the ETT test set. This visualization empirically confirms that the pattern recognizer$D_\phi$​​, learned through the EM iterations, closely approximates the true missing pattern. And as discussed in our response to W5, the closer $D_\phi$ is to the true missing ratio, the smaller the cross-entropy value becomes.
> > >
> > > To demonstrate this phenomenon, in the revised version, we include additional case studies in Figure 13-16 of **Appendix C.2.4**, where we illustrate that $D_\phi$​ is indeed well trained by visualizing its predictions on randomly sampled time-series segments from ETT, STOCK, and PEMS-Bay. Please refer the reviewer to this updated section for further details.
> > >
> > > We have also added **Appendix C.2.5**, where Figures 16–18 present the qualitative out-of-sample imputation results for the ETT, STOCK, and PEMS-Bay datasets, respectively. For clearer visualization and fair comparison across datasets, we uniformly cropped each feature dimension in the same size.
> > >
> > > > **W4**. The terms "out-of-sample imputation tasks" and "in-sample imputation tasks" are not clearly defined in the main text.
> > >
> > > We notice that we missed to summarize the explanation of _in-sample_ and _out-of-sample_ imputation in the paper. In our setting, _in-sample imputation_ refers to data that are directly used during model training, which corresponds to the incomplete training set in our experiments.  In contrast, _out-of-sample imputation_ refers to unseen data that the imputation model does not observe during training, which includes the validation and test sets. We updated the explanation with **blue text** to provide a clear and more detailed description of these concepts in the last paragraph of **Appendix C** section of the revised paper. Figure 8 of revised paper could help the reviewer to better understand the data flow.

---

> > > > ### Author Response · Authors · 2025-11-21
> > > >
> > > > > **W7**. The diffusion backbone and missing pattern recognizer require joint optimization, which may lead to high training costs. The paper lacks an analysis of the model's efficiency.
> > > >
> > > > Thank you for pointing out the need for a more thorough analysis of the model’s efficiency. In our experiments, we design pattern recognizer as simple CNN structure being significantly smaller than the diffusion model. In spite of our effort to reduce recognizer size, there are two primary reasons why PRDIM requires longer training time compared to prior diffusion-based imputation methods:
> > > >
> > > > - In expectation step, computing approximate guidance requires torch.autograd operations on both the diffusion backbone and the pattern recognizer, increasing computational overhead.
> > > > - In maximization step, the pattern recognizer must be updated via gradient descent at each EM iteration, adding an additional optimization component on top of the diffusion model.
> > > >
> > > > In Appendix C, Table 6, we report the wall-clock training time on ETT as well as the out-of-sample imputation time across diffusion-based imputation baselines. Notably, the parameter count of the pattern recognizer corresponds to only 10.61% of the diffusion model, meaning that the additional parameters themselves do not significantly increase backpropagation cost.  However, due to the guidance computation in both the expectation step and the imputation process, training _wall-clock time_ increases by 54.14% relative to DiffPuter, and inference time increases by 59.57%.
> > > > We emphasize that these measurements correspond to applying approximate guidance at every diffusion timestep. As explored in DPS [6], guidance can be applied only from intermediate timesteps onward without compromising performance. We believe such a strategy can substantially improve time efficiency, and we consider this an important direction for future work.
> > > >
> > > > [6] Chung, Hyungjin, et al. "Diffusion Posterior Sampling for General Noisy Inverse Problems." _The Eleventh International Conference on Learning Representations_.
> > > >
> > > > | **Method**    | **Model Size (ETT)** | **Training Time (s)** | **Inference Time (s)** | **Model Size (STOCK)** | **Training Time (s)** | **Inference Time (s)** |
> > > > |---------------|----------------------:|-----------------------:|------------------------:|------------------------:|-----------------------:|------------------------:|
> > > > | CSDI          | 164025               | 366                    | 39                     | 164017                 | 138                    | 11                     |
> > > > | MTSCI         | 162321               | 585                    | 21                     | 146969                 | 354                    | 12                     |
> > > > | cDiffPuter    | 163769               | 981                    | 28                     | 163761                 | 516                    | 11                     |
> > > > | PRDIM (Ours)  | 163769 + 17376       | 1812                   | 47                     | 163761 + 17359         | 1052                   | 28                     |

---

### Author Response · Authors · 2025-12-02

We sincerely thank all reviewers for their valuable questions and insightful comments! We carefully organized the questions raised by each reviewer, and a summary of our responses is provided below. Based on these discussions, we are going to upload a revised version of the paper.

**Reviewer 7n6G**

-   In Appendix C.2.2, we evaluate PRDIM on the CelebA-HQ dataset to demonstrate its imputation performance on high-dimensional image data, which is similar in nature to Moving-MNIST.

-   Moreover, PRDIM is not intended solely for time-series imputation; rather, it is a general framework applicable across diverse data modalities. To support this, we report additional results on image and tabular datasets in Appendices C.2.1, C.2.2, and C.2.3.

**Reviewer vEpr**

-   We appreciate the reviewer’s suggestion regarding different MNAR subtypes. We interpreted this as a question of whether PRDIM remains robust under various missing distributions. To address this, we investigated three practically implementable MNAR subtypes and compared PRDIM with existing diffusion-based imputation models under each setting.

**Reviewer i3qS**

-   We analyzed the effect of Phase 1 initialization strategies on PRDIM’s performance, confirming both the benefits of using diffusion-based initialization and the validity of our experimental configuration. We consider this question to be an important experiment in perspective of applications in our research, and we added this experiment results in Section 4.5 of the revised paper.

-   In addition, we carefully revised the presentation of key concepts in the paper based on the reviewer’s feedback, adding clarifications to ensure smooth and unambiguous logical flow for readers.

**Reviewer Yosc**

-   We thank the reviewer for raising additional thoughtful questions. We view these questions as theoretically meaningful and potentially conducive to advancing diffusion imputation research. We planned and conducted new experiments accordingly and reported their results in the discussion.

-   The suggested work on regularizing generation diversity in diffusion imputation model [1] presents a promising orthogonal direction, and we acknowledge it as a valuable avenue for future extensions of PRDIM. We appreciate the reviewer’s constructive insight.

[1] Self-Supervision Improves Diffusion Models for Tabular Data Imputation

**Reviewer pWjT**

-   We agree that the reviewer’s recurring concern touches on a key conceptual difference between PRDIM and existing diffusion imputation models regarding their target objectives. To address this clearly, we added a new paragraph titled **Revisiting the Objective of Diffusion Imputation Models** in the Preliminaries section, providing a precise explanation of our research question and methodological perspective. We are pleased that the clarification resolved the reviewer’s concerns and contributed to their improved evaluation.

To all reviewers, we sincerely appreciate your engagement with our work and the constructive feedback that has strengthened both the robustness of the current paper and the direction of future research.

---

### Meta-Review · Area_Chair_baZ8 · 2026-01-05

**Summary:**

The authors provided a careful and detailed rebuttal and addressed several reviewer questions. Nevertheless, despite these efforts, several important concerns remain insufficiently addressed, which collectively lead me to recommend rejection.

First, although the paper targets MNAR settings and includes several simulated MNAR scenarios, the experimental evaluation relies predominantly on synthetic missingness. While the authors provide some additional visualizations related to the empirical validation on real-world datasets with their original missing data remains limited. This substantially constrains the practical relevance of the proposed method, particularly given the complexity and heterogeneity of MNAR mechanisms encountered in real applications.

Second, the core pattern recognition component—consisting of two 1D CNNs with residual connections—plays a central role in the proposed framework. While the paper provides some discussion of this module, the analysis and interpretation of its learned representations and outputs remain limited. As a result, it is still difficult to assess whether and how the model genuinely captures missingness patterns, beyond observed empirical performance gains.

Third, based on both the paper title and the authors’ clarification in the rebuttal, PRDIM is not a time-series model and does not explicitly model temporal shifts; instead, it is positioned as a likelihood-based framework applicable to any data formats. This raises an important conceptual gap. If the framework is indeed modality-agnostic, then for domains such as images, relevant domain-specific baselines (e.g., standard inpainting methods) should be considered and discussed. The current manuscript lacks sufficient discussion and comparison along this dimension.

Additionally, several reviewers noted that certain descriptions in the paper may unintentionally cause confusion for readers in the data imputation community, particularly regarding the scope of the method and its relationship to existing imputation paradigms.

Taken together, these concerns, spanning experimental validation, model interpretability, conceptual positioning, evaluation methodology, and theoretical clarity, remain substantial and prevent a confident endorsement of the paper for acceptance.

**Reviewer Concerns:**

**Concerns partially or fully addressed**

Across reviewers, the rebuttal and revision addressed a number of clarification- and presentation-level concerns:

•	Two reviewers ( Yosc, pWjT) acknowledged that the authors added additional explanations and figures (e.g., Figure 12) to distinguish original missing data from artificially generated missingness, which helped clarify parts of the experimental setup.

•	Some questions regarding experimental protocol and evaluation (e.g., Q1/Q2 for multiple reviewers, W4 for 7n6G, W2/W4 for i3qS, W1–W7 for Yosc, W1/W2 for pWjT) were considered settled after the rebuttal.

•	Additional experiments were provided to demonstrate improvements in imputation performance under simulated MNAR settings, which partially addressed concerns about empirical validation (7n6G, pWjT).

•	Clarifications were given regarding the intended scope of the method, in particular that PRDIM is not a time-series model but a likelihood-based framework applicable to general data modalities.


**Outstanding concerns**

Despite these improvements, several **substantive concerns remain unresolved**, and were repeatedly raised by multiple reviewers:

1.	**Limited real-world MNAR validation**

Multiple reviewers (7n6G, vEpr, pWjT) remained concerned that the experimental evaluation relies heavily on simulated MNAR settings. While a visualization on real-world data was added, quantitative evaluation and deeper analysis on datasets with real original missingness are still lacking, limiting the practical credibility of the claims.

2.	**Insufficient analysis of the pattern recognition module**

Reviewers (7n6G, vEpr) noted that the core pattern recognition component, implemented as two 1D CNNs with residual connections, remains insufficiently analyzed. There is no detailed investigation of what the module learns or how its outputs correspond to meaningful missing patterns beyond aggregate metrics.

3.	**Conceptual gap regarding modality generality**

As highlighted by reviewer 7n6G, the clarification that PRDIM is modality-agnostic introduces a new concern: in domains such as images, standard inpainting methods and established baselines should be considered. The paper currently lacks sufficient discussion and comparison in this respect.

4.	**Computational cost**

Reviewer 7n6G noted that the additional pattern recognition component increases computational time, which remains a disadvantage of the proposed method.

**Reviewer Scores:**

**Reviewer 7n6G**

**Expected score change: No change**

While some clarifications and additional experiments were acknowledged, major concerns regarding real-world MNAR validation, pattern recognition analysis, evaluation methodology, and computational cost remain heavy. These outstanding issues are likely to prevent an upward revision of the score.


**Reviewer vEpr**

**Expected score change: No change**

The rebuttal addressed some questions and added simulated MNAR experiments, but key weaknesses—including lack of real-world MNAR analysis, insufficient examination of the pattern recognition module, and unresolved theoretical concerns—persist. A score increase is therefore unlikely.


**Reviewer i3qS**

**Expected score change: Slight improvement in confidence, but no score change**

The added figures and clarifications addressed several questions, and some concerns were settled. However, doubts regarding the interpretation of missing ratio alignment remain, making a formal score increase unlikely.


**Reviewer Yosc**

**Expected score change: No change (remains strongly positive)**

This reviewer indicated that all listed concerns were settled in the rebuttal and already assigned high scores. No further score change is expected.


**Reviewer pWjT**

**Expected score change: Slight positive increase**

Most concerns were addressed in the rebuttal, and the additional clarifications and experiments improved the overall presentation and empirical support. However, the lack of detailed quantitative experiments on real-world datasets with naturally occurring missingness (Q1) remains a notable limitation. As a result, while a modest score increase is plausible, the remaining gap prevents a substantial revision of the reviewer’s assessment.

---

### Decision · Program_Chairs · 2026-01-26

Reject